# From Flat to Hierarchical: Extracting Sparse Representations with Matching Pursuit

Valérie Costa[1*]    Thomas Fel[2*]    Ekdeep Singh Lubana[3,4*]
Bahareh Tolooshams[5,6†]    Demba Ba[2,7†]

[1]EPFL    [2]Kempner Institute, Harvard University
[3]CBS-NTT Program in Physics of Intelligence, Harvard University
[4]Physics of Artificial Intelligence Group, NTT Research, Inc., Sunnyvale, CA, USA
[5]University of Alberta    [6]Alberta Machine Intelligence Institute (Amii)    [7]SEAS, Harvard University

## Abstract

Motivated by the hypothesis that neural network representations encode abstract, interpretable features as linearly accessible, approximately orthogonal directions, sparse autoencoders (SAEs) have become a popular tool in interpretability literature. However, recent work has demonstrated phenomenology of model representations that lies outside the scope of this hypothesis, showing signatures of hierarchical, nonlinear, and multi-dimensional features. This raises the question: do SAEs represent features that possess structure at odds with their motivating hypothesis? If not, does avoiding this mismatch help identify said features and gain further insights into neural network representations? To answer these questions, we take a construction-based approach and re-contextualize the popular matching pursuit (MP) algorithm from sparse coding to design MP-SAE—an SAE that unrolls its encoder into a sequence of residual-guided steps, allowing it to capture hierarchical and nonlinearly accessible features. Comparing this architecture with existing SAEs on a mixture of synthetic and natural data settings, we show: (i) hierarchical concepts induce conditionally orthogonal features, which existing SAEs are unable to faithfully capture, and (ii) the nonlinear encoding step of MP-SAE recovers highly meaningful features, helping us unravel shared structure in the seemingly dichotomous representation spaces of different modalities in a vision-language model, hence demonstrating the assumption that useful features are solely linearly accessible is insufficient. We also show that the sequential encoder principle of MP-SAE affords an additional benefit of adaptive sparsity at inference time, which may be of independent interest. Overall, we argue our results provide credence to the idea that interpretability should begin with the phenomenology of representations, with methods emerging from assumptions that fit it.

## 1  Introduction

Modern neural networks trained on large-scale datasets have achieved unprecedented performance on several practical tasks [1–10], prompting efforts towards understanding how these abilities are implemented within a model [11–13]. To this end, Sparse Autoencoders (SAEs) [14–23], motivated by the Linear Representation Hypothesis (LRH) [24, 25], have become a popular tool for interpreting neural networks. Specifically, LRH, a phenomenological model of organization and computation of neural network representations [24–28], posits that neural network representations of dimension $m$ can be decomposed along a basis of $p \gg m$ approximately orthogonal directions that reflect abstract, interpretable concepts underlying the data distribution. Here, approximate orthogonality is argued to be necessary for circumventing the problem of packing more unique vectors (concepts) than the

---

[*]Equal contribution. [†]Equal advising. Emails: `btolooshams@ualberta.ca`, `demba@seas.harvard.edu`.

dimensionality of the space allows (a.k.a. the superposition problem [24]), hence enabling reliable estimation of a concept's presence by a linear operator (e.g., the linear map in an MLP) [29, 30]. Grounded in this idea, and inspired by its relation to the well-known problem of sparse coding [31, 32], SAEs have been proposed to learn, in an unsupervised manner, a sparse, overcomplete dictionary of directions that (ideally) map onto abstract, interpretable concepts encoded in a neural network, hence enabling its interpretability [33, 14].

Despite the fact that substantial empirical support has been shown in favor of LRH—e.g., representations capturing concepts like geometry and lighting [34–38], facial characteristics [39, 40], broader scene organizations [41–43], and instance structures [44–51] have been shown in vision models, while ones capturing concepts like basic semantics [52–57], character roles and theory-of-mind properties [58, 59], arithmetic concepts [60], refusal to unsafe queries [61, 62], and subject–object relations [63–66] have been shown in language models—our goal in this work is to better contextualize the assumptions underlying the design of SAEs in light of recent evidence *against* the validity of LRH [67, 68]. For example, Park et al. [69] recently analyzed the organization of hierarchical and categorical concepts in neural network representations, showing that cross-entropy loss tends to encourage orthogonal structure for hierarchical relations, while categorical concepts are arranged as polytopes with overlapping support. Meanwhile, Csordas et al. [70] demonstrated the existence of "onion-like" representations in a simple copying task that cannot be linearly accessed; the existence of such "nonlinear" representations in larger-scale models was also emphasized by Engels et al. [71]. Finally, the existence of multi-dimensional features for concepts with temporal relations, e.g., days-of-the-week, has been used to argue against the use of directions as a basis for decomposing neural network representations [72, 21]; this is also supported by experiments showing concepts' representations and dimensionality can be flexibly manipulated via the inputted context [73].

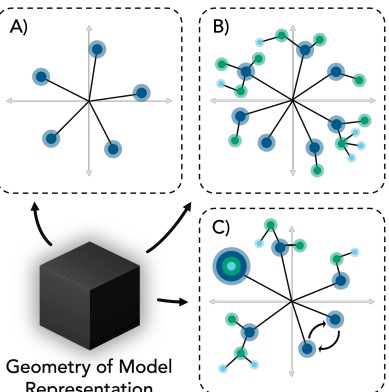

Figure 1: **Conceptual organization in neural representations. A)** *Linearly accessible concepts*: abstract directions that are approximately orthogonal and independently interpretable, as assumed by the Linear Representation Hypothesis (LRH). **B)** *Hierarchical concepts*: representations structured in parent–child relations. **C)** *Nonlinear, multidimensional, and temporally structured concepts*: features that cannot be accessed via a single direction.

The disparity highlighted above between phenomenology of neural network representations identified in recent work versus one hypothesized by LRH raises the question as to whether SAEs, which were motivated by LRH, are able to capture and explain representations that lie outside the scope of LRH (Fig. 1). Since the SAE architecture is not explicitly biased against such representations, is it possible SAEs remain performant towards expressing them, or do these disparities manifest as limitations on what concepts SAEs can explain? To address these questions, we make the following contributions.

- **Contextualizing SAEs beyond the linear representation hypothesis.** Via a mixture of synthetic and natural data, we analyze whether popular SAE architectures are able to capture concepts encoded in a hierarchical, nonlinearly accessible manner (see Hindupur et al. [21] for an analysis of multi-dimensional concepts). To this end, we introduce the notion of *Conditional Orthogonality* (Def. 2.3), under which, in a hierarchy, orthogonality is expected only between concepts within across different levels. Our experiments yield consistent evidence that existing SAEs, including those trained with hierarchical objectives [20], are unable to capture such concepts faithfully.

- **Introducing MP-SAE for phenomena beyond linear assumptions.** To contextualize the results above and understand if capturing hierarchical structures yields meaningful features in larger-scale models, we design an SAE architecture that can capture (certain kinds of) hierarchical and nonlinear structures. Specifically, we extend the standard SAE framework by unrolling the Matching Pursuit algorithm [74] into a residual-guided, sequential encoder—called the *MP-SAE*. Each step of MP-SAE selects a feature (approximately) orthogonal to what has already been explained, naturally promoting a conditionally orthogonal structure. Across experiments on both synthetic benchmarks and real-world vision-language models, MP-SAE proves more expressive and uncovers richer structural features than existing SAEs. These results question the assumption that meaningful features must be linearly and independently accessible.

- **Additional advantages of MP-SAE.** We find MP-SAE recovers richer structure than standard SAEs across diverse settings: on synthetic trees with controlled interference, it uniquely preserves both within- and across-level organization; on large-scale models (e.g., CLIP [75], DINOv2 [76]), it identifies nonlinear and multimodal features that elude linear encoders. Moreover, we show MP-SAE offers two practical benefits: (i) it enables adaptive sparsity without retraining—features can be incrementally added at test time—and (ii) it ensures monotonic improvement in reconstruction error with each inference step. These properties especially contrast with fixed-$k$ SAEs, which often degrade when sparsity levels shift at test time. Thus, though our goal for analyzing MP-SAE was to understand whether useful features can be elicited by actively modeling phenomenology outside the scope of LRH, we believe MP-SAE can be of independent interest to the community.

## 2    Formalizing Linear Representation Hypothesis and Sparse Autoencoders

We begin by formally describing two objects central to our study: linear representation hypothesis and sparse autoencoders. We note that though the idea that concepts are "linearly encoded" in neural networks has found significant attention in literature (see App. A), a formal definition has rarely been offered (see [25, 77] for notable exceptions). Consequently, several related observations on neural network representations have come to be associated with LRH, including linear accessibility of interpretable concepts [78, 59], linear algebraic manipulability of model behavior [79, 66, 80], and decomposability into a linear mixture of directions [14, 23, 22]. Hence, to be precise about what we mean by the term, we distill the core claims posited by Elhage et al. [24] on LRH and formally state (our interpretation of) the hypothesis. This definition then directly motivates SAEs.

In what follows, we use plain lowercase letters $a$ to denote scalars, bold lowercase letters $\boldsymbol{a}$ to denote vectors, and uppercase letters $\boldsymbol{A}$ to denote matrices. The $k^{\text{th}}$ column of a matrix $\boldsymbol{A}$ is written as $\boldsymbol{A}_k$. The norms $\|\cdot\|_2$ and $\|\cdot\|_1$ denote the standard $\ell_2$ and $\ell_1$ norms, while $\|\cdot\|_0$ refers to the $\ell_0$ *pseudo*-norm, i.e., the number of nonzero entries. The Frobenius norm of a matrix is denoted by $\|\cdot\|_F$. For any vector $\boldsymbol{z}$, its support is defined as $\operatorname{supp}(\boldsymbol{z}) := \{i : z_i \neq 0\}$, and $\boldsymbol{z}$ is said to be $k$-sparse if $|\operatorname{supp}(\boldsymbol{z})| = k$. Hereafter, we use "features" and "concepts" as interchangeable terms, distinguishing them from "representations" a model derives for an input.

**Definition 2.1** (**Linear Representation Hypothesis (LRH)**). *A representation $\boldsymbol{x} \in \mathbb{R}^m$ is said to satisfy the* linear representation hypothesis *(LRH) if there exists a dictionary $\boldsymbol{D} = [\boldsymbol{D}_1, \ldots, \boldsymbol{D}_p] \in \mathbb{R}^{m \times p}$ and a coefficient vector $\boldsymbol{z} \in \mathbb{R}^p$ such that $\boldsymbol{x} = \boldsymbol{D}\boldsymbol{z}$, under the following conditions:*

$$
\begin{cases}
\textbf{(i) Overcompleteness:} & p \gg m; \\
\textbf{(ii) Quasi-orthogonality:} & \max_{i \neq j}\left|\boldsymbol{D}_i^\top \boldsymbol{D}_j\right| \leq \varepsilon, \quad \text{where } \forall i \ \|\boldsymbol{D}_i\|_2 = 1 \ ; \text{ and} \\
\textbf{(iii) Sparsity:} & |\operatorname{supp}(\boldsymbol{z})| \leq k \ll p.
\end{cases}
$$

We emphasize the constraints above are deeply interdependent. In particular, if a model is to represent a large number of distinct concepts ($p \gg m$) within a lower-dimensional space $\mathbb{R}^m$, while ensuring that a linear readout can reliably recover any active concept, the directions associated with these concepts must be approximately orthogonal. This requirement is substantiated by a reinterpretation of the Johnson–Lindenstrauss (JL) lemma [29, 30][2]. While the lemma is typically used to show that high-dimensional data can be compressed into lower dimensions while preserving pairwise distances, Elhage et al. [24] apply a flipped version: one can embed exponentially many quasi-orthogonal directions *within* $\mathbb{R}^m$ such that sparse linear combinations remain distinguishable. This perspective justifies the feasibility of constructing overcomplete, low-coherence dictionaries in LRH. Rather than compressing structure, the goal is to expand representational capacity: to pack many interpretable directions into a shared space while preserving linear accessibility.

Overall, if the conditions above hold, sparse linear decompositions become not only possible, but a natural mechanism for expressing concept-aligned features—this leads to the idea of SAEs. Specifically, noting that the model described in Def. 2.1 closely aligns with the generative model assumed in classical work on sparse coding [32, 31, 81], wherein one seeks to express data as sparse linear combinations over an overcomplete dictionary, SAEs were recently proposed to re-contextualize that literature's tools for identifying concepts encoded in a neural network's representations [14–23].

---

[2]JL-lemma [29] states that $p = e^{\mathcal{O}(m)}$ vectors can be embedded in $\mathbb{R}^m$ such that all pairwise inner products are bounded by a small $\varepsilon > 0$, thereby justifying the feasibility of near-orthogonality even when $p \gg m$.

**Definition 2.2** (**Sparse Autoencoders**). *Given model representations* $\boldsymbol{x} \in \mathbb{R}^m$, *the goal of an SAE is to compute a sparse code* $\boldsymbol{z}$ *that reconstructs* $\boldsymbol{x}$ *as a linear combination of a learned dictionary* $\boldsymbol{D}$:

$$\begin{aligned} \boldsymbol{z} &= \boldsymbol{\Pi}\{\boldsymbol{W}^\top(\boldsymbol{x} - \boldsymbol{b}_{pre}) + \boldsymbol{b}\}, \quad and \\ \hat{\boldsymbol{x}} &= \boldsymbol{D}\boldsymbol{z} + \boldsymbol{b}_{pre}, \end{aligned} \tag{1}$$

*where* $\boldsymbol{W} \in \mathbb{R}^{m \times p}$ *and* $\boldsymbol{b} \in \mathbb{R}^p$ *denote the encoder weights and bias,* $\boldsymbol{b}_{pre} \in \mathbb{R}^m$ *is a pre-decoding bias, and* $\boldsymbol{D} \in \mathbb{R}^{m \times p}$ *is the learned dictionary of concept vectors.*

In the above, the nonlinearity projection $\boldsymbol{\Pi}\{\cdot\}$ projects the pre-activation to a sparse support [21]; common choices for activation maps include ReLU [23, 14], TopK [82, 15, 17], and JumpReLU [18, 19]. Training proceeds by minimizing a reconstruction loss along with sparsity and auxiliary penalties:

$$\mathcal{L} = \|\boldsymbol{x} - \hat{\boldsymbol{x}}\|_2^2 + \lambda\,\mathcal{R}(\boldsymbol{z}) + \alpha\,\mathcal{L}_{\text{aux}},$$

where $\mathcal{R}(\boldsymbol{z})$ promotes sparsity, via $\ell_1$ or $\ell_0$ mechanisms, and $\mathcal{L}_{\text{aux}}$ may be used to minimize number of inactive units [15, 19].

## 2.1 Stress-Testing SAEs via Conditional Orthogonality—a Structure Outside LRH's Scope

While SAEs provide a natural operationalization of LRH, their effectiveness hinges on structural assumptions that may not hold in practice. In particular, Eq. 1 presumes that concepts correspond to approximately orthogonal directions and can be recovered via a linear projection. However, as several recent works show, these assumptions can prove overly rigid [72, 68, 71, 21, 69]. For instance, neural network representations are known to encode hierarchically structured concepts [83–88]. Park et al. [69] show that for such hierarchical concepts, while concepts at the same level of hierarchy frequently form polytopes with overlapping support, parent and child concepts tend to span orthogonal subspaces. In what follows, we formalize this idea under the term 'conditional orthogonality' and use it as a structure to stress-test whether SAEs can identify concepts that encoded in a manner outside the scope of LRH. We note this definition is merely a paraphrased version of the one offered by Park et al. [69].

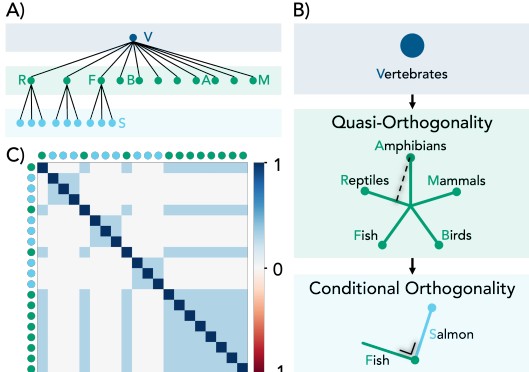

Figure 2: **Illustrative Example of Conditional vs. Quasi-Orthogonality. A)** Example of a hierarchical concept tree. **B)** Comparison of quasi-orthogonality (interference within levels) vs. conditional orthogonality (orthogonality across levels). **C)** Correlation matrix of features sampled from A, showing conditional orthogonality (white, $= 0$) across levels and quasi-orthogonality (light blue, $= \varepsilon$) within levels.

**Definition 2.3** (**Conditional Orthogonality**). *Let* $\boldsymbol{D} \in \mathbb{R}^{m \times p}$ *be a dictionary whose columns* $\boldsymbol{D}_1, \ldots, \boldsymbol{D}_p$ *denote concept directions, and let* $\ell : [p] \to \mathbb{N}$ *assign each concept to a discrete level in a hierarchy. We say that* $\boldsymbol{D}$ *is* conditionally orthogonal *with respect to* $\ell$ *if*

$$\boldsymbol{D}_i^\top \boldsymbol{D}_j = 0 \quad \text{whenever } \ell(i) \neq \ell(j).$$

Relating back to LRH (Def. 2.1), we note Def. 2.3 relaxes the global quasi-orthogonality constraint posited by LRH, requiring instead orthogonality only between concept vectors drawn from different hierarchical levels: specifically, note that no constraint is imposed on inner products $\boldsymbol{D}_i^\top \boldsymbol{D}_j$ for pairs $i, j \in [p]$ with $\ell(i) = \ell(j)$. This permits controlled interference within a given hierarchical level, while preserving separation across levels. Fig. 2, adapted from Bussmann et al. [20], depicts an instance of such structure, where hierarchical organization gives rise to distinct patterns of inter- and intra-level alignment. To evaluate whether conventional SAEs (implicitly aligned with LRH) are capable of capturing the interference structure induced by conditional orthogonality, we will analyze a synthetic generative process grounded in the taxonomy of Fig. 2 in Sec. 4.1. This will allow us to probe the inductive biases of different SAE variants under controlled structural constraints. However, to contextualize these results, we next develop an SAE that is specifically motivated to capture the ability to model features that are conditionally orthogonal in nature.

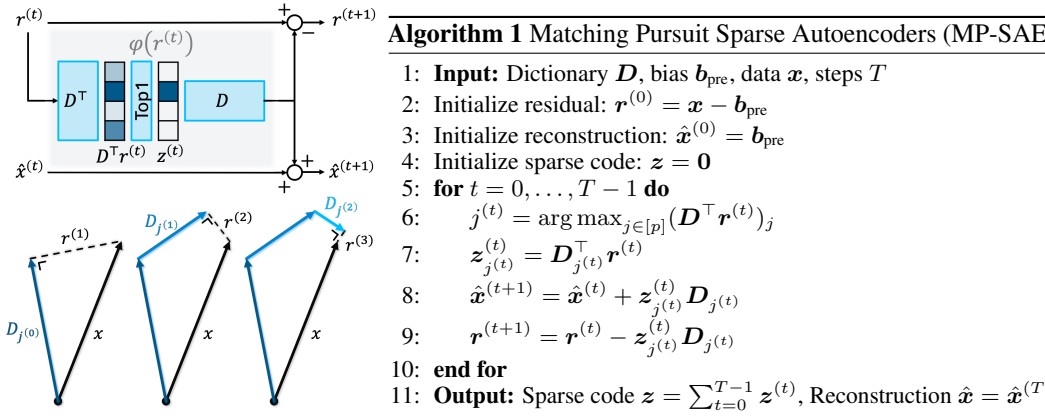

**Algorithm 1** Matching Pursuit Sparse Autoencoders (MP-SAE)

1: **Input:** Dictionary $\boldsymbol{D}$, bias $\boldsymbol{b}_{\text{pre}}$, data $\boldsymbol{x}$, steps $T$
2: Initialize residual: $\boldsymbol{r}^{(0)} = \boldsymbol{x} - \boldsymbol{b}_{\text{pre}}$
3: Initialize reconstruction: $\hat{\boldsymbol{x}}^{(0)} = \boldsymbol{b}_{\text{pre}}$
4: Initialize sparse code: $\boldsymbol{z} = \boldsymbol{0}$
5: **for** $t = 0, \dots, T - 1$ **do**
6: $\quad j^{(t)} = \arg\max_{j \in [p]} (\boldsymbol{D}^\top \boldsymbol{r}^{(t)})_j$
7: $\quad \boldsymbol{z}^{(t)}_{j^{(t)}} = \boldsymbol{D}^\top_{j^{(t)}} \boldsymbol{r}^{(t)}$
8: $\quad \hat{\boldsymbol{x}}^{(t+1)} = \hat{\boldsymbol{x}}^{(t)} + \boldsymbol{z}^{(t)}_{j^{(t)}} \boldsymbol{D}_{j^{(t)}}$
9: $\quad \boldsymbol{r}^{(t+1)} = \boldsymbol{r}^{(t)} - \boldsymbol{z}^{(t)}_{j^{(t)}} \boldsymbol{D}_{j^{(t)}}$
10: **end for**
11: **Output:** Sparse code $\boldsymbol{z} = \sum_{t=0}^{T-1} \boldsymbol{z}^{(t)}$, Reconstruction $\hat{\boldsymbol{x}} = \hat{\boldsymbol{x}}^{(T)}$

Figure 3: **(Top Left) One iteration of the MP-SAE encoder:** The residual $\boldsymbol{r}^{(t)}$ is projected onto the dictionary $\boldsymbol{D}$, the most correlated feature $\boldsymbol{D}_{j^{(t)}}$ is selected, and its contribution is subtracted from the residual and added to the reconstruction. **(Bottom Left) MP sequentially reconstructs** $\boldsymbol{x}$ by greedily selecting features that best explain the residual, promoting orthogonal selection and enabling access to higher-order features that are nonlinearly accessible from $\boldsymbol{x}$. **(Right)** MP-SAE embeds the MP algorithm into a sparse autoencoder, where the dictionary $\boldsymbol{D}$ is learned through backpropagation.

## 3 Operationalizing Conditional Orthogonality via MP-SAE

To enable the ability to identification of features encoded in a *conditionally orthogonal* manner (Def. 2.3), we construct a sparse autoencoder whose inference reflects the structure of hierarchical representations—minimizing interference across levels while tolerating it within [69]. Our design draws on the Matching Pursuit (MP) algorithm [74] from sparse coding [89–92], which has been shown to recover features from the appropriate hierarchical level in conditionally orthogonal dictionaries [93]. MP performs inference greedily: at each step, the feature most correlated with the residual is selected, its contribution subtracted, and the process repeated. This iterative residual decomposition promotes conditionally orthogonal feature selection, as each new feature explains variance orthogonal to what was captured by the previous feature (Fig. 3). Embedding this mechanism into a sparse autoencoder yields the *Matching Pursuit Sparse Autoencoder (MP-SAE)*, formalized in Algorithm 1. Below, we unpack the structure and implications of MP-SAE, examining: (*i*) the mechanics of its inference procedure; (*ii*) how this procedure promotes conditional orthogonality; and (*iii*) its capacity to extract higher-order, nonlinearly accessible features.

(*i*) **Mechanics of MP-SAE.** MP-SAE uses a shared learned dictionary for encoding and decoding. The encoder embeds the classical Matching Pursuit algorithm [74] by unrolling its greedy inference procedure into a fixed number of steps. At inference, the model starts with an initial estimate and residual (Algorithm 1, lines 3 and 2). At each iteration, MP-SAE greedily selects the feature that best aligns with the current residual by computing the inner product between the residual and each feature, and choosing the one with the highest projection (line 6). Once the best-matching feature is identified, the algorithm determines its contribution by projecting the residual onto that feature (line 7), adds this contribution to the current approximation of the input (line 8), and updates the residual accordingly (line 9). This procedure is repeated for $T$ steps, producing a sparse code with $\|\boldsymbol{z}\|_0 \leq T$. The resulting encoding represents a sequential approximation of the input, constructed through greedy selection of locally optimal features that progressively refine the reconstruction of $\boldsymbol{x}$.

(*ii*) **Conditional Orthogonality via Sequential Inference.** A defining property of MP-SAE is that its greedy inference procedure promotes conditional orthogonality across selected features. This emerges from the residual update rule: once a concept $\boldsymbol{D}_{j^{(t-1)}}$ has been selected from the dictionary at iteration $t-1$, the updated residual $\boldsymbol{r}^{(t)}$ is orthogonal to it, as stated below formally.

**Proposition 3.1** (Stepwise Orthogonality of MP Residuals). *Let $\boldsymbol{r}^{(t)}$ be the residual at iteration $t$ of MP-SAE inference, and let $\boldsymbol{D}_{j^{(t-1)}}$ be the feature selected at step $t-1$. Then:*

$$\boldsymbol{D}^\top_{j^{(t-1)}} \boldsymbol{r}^{(t)} = 0.$$

Each selected concept is removed from the residual subspace before the next selection, and subsequent concepts are chosen to explain what remains in the residual, orthogonal to the last selected feature. When trained, this yields a dictionary of features that are not globally orthogonal but are selectively chosen to be conditionally orthogonal. Although MP-SAE only enforces orthogonality to the most recently selected concept—unlike Orthogonal Matching Pursuit [89], which re-orthogonalizes against all prior selections—we observe empirically that residuals are often nearly orthogonal to all previously selected directions. This emergent behavior suggests the model implicitly promotes hierarchical separation in the learned dictionary, minimizing interference across levels. It aligns with the inductive bias of conditional orthogonality (Def. 2.3) and contrasts with standard SAEs, which tend to promote global quasi-orthogonality regardless of feature structure.

(***iii***) **Access nonlinearly accessible features.** Beyond promoting conditional orthogonality, the residual-based inference structure of MP-SAE enables access to features that are nonlinearly embedded in the representation space. While each iteration applies a linear projection to the current residual, the residual itself evolves nonlinearly as a function of the input, due to its recursive dependence on previous selections. This results in a structured approximation of $\boldsymbol{x}$ that can be decomposed as:

$$\boldsymbol{x} = \boldsymbol{r}^{(0)} = \boldsymbol{\varphi}(\boldsymbol{x}) + \boldsymbol{r}^{(1)} = \underbrace{\boldsymbol{\varphi}(\boldsymbol{x})}_{\textit{linearly accessible}} + \underbrace{\sum_{t=1}^{T} \boldsymbol{\varphi}(\boldsymbol{r}^{(t)})}_{\textit{nonlinearly accessible}} + \underbrace{\boldsymbol{r}^{(T+1)}}_{\textit{residual error}},$$

where $\boldsymbol{\varphi}(\cdot)$ denotes the linear projection onto the selected feature at each step (See Fig. 3). The crucial insight is that, although each $\boldsymbol{\varphi}(\boldsymbol{r}^{(t)})$ is linear in its argument, the argument itself, $\boldsymbol{r}^{(t)}$, is a nonlinear function of $\boldsymbol{x}$. As a result, the composition $\boldsymbol{\varphi}(\boldsymbol{r}^{(t)})$ defines a feature that cannot be obtained from $\boldsymbol{x}$ via a single linear map. MP-SAE can thus potentially uncover higher-order concepts that are conditionally dependent on previously explained structure. This mechanism may be particularly relevant in settings where important features are entangled or nonlinearly composed such as hierarchies, temporal dependencies, or multimodal correlations. Moreover, it provides a constructive hypothesis for the phenomenon of "dark matter" in neural representations [71]: features that evade standard SAEs because they are not linearly *accessible* from the raw representation; that is, one can still have the LRH assumption of a linear mixing process hold, i.e., $x = Dz$, but other constraints start to relax (Def. 2.1). We return to this phenomenon in our empirical analysis.

## 4 Empirical Results

We now take a step towards uncovering challenges in SAEs emergent from assuming a partially correct model of neural network representations, i.e., LRH. Specifically, in Sec. 4.1, we analyze a synthetic domain that highlight the inability of existing SAEs to identify hierarchically structured concepts in a clearly defined domain. Building on these results, in Sec. 4.2, we analyze a natural vision–language domain to assess how features extracted under an inductive bias that accommodates hierarchical, nonlinearly accessible concepts—as in MP-SAE—differ from those identified by existing SAEs. In particular, we show that MP-SAE uniquely recovers cross-modal structure.

### 4.1 A Synthetic Generative Model of Hierarchical Features

We begin by evaluating whether SAEs, including our proposed MP-SAE, can recover conditionally orthogonal features using a synthetic hierarchy-based setup adapted from Bussmann et al. [20]. To formalize this setting, we introduce the following definition of a hierarchical generative process.

**Definition 4.1** (**Hierarchical Generative Process**). *A generative process over $\boldsymbol{p}$ (parent concept) and $\boldsymbol{c}$ (child concept) is said to be* hierarchical *if their activations $z_p$ and $z_c$ satisfy*

$$\mathbb{P}(z_p > 0 \mid z_c > 0) = 1 \quad \textit{and} \quad \mathbb{P}(z_c > 0 \mid z_p > 0) < 1.$$

*That is, a child's activation requires its parent's, whereas a parent may activate independently.*

Based on this definition, we implemented the following hierarchical generative process. The ground truth $\boldsymbol{D}$ consists of 20 unit-norm concepts organized into a two-level tree: 11 disjoint parent concepts $\boldsymbol{D}_p$, 3 of which each have 3 disjoint children $\boldsymbol{D}_c$ as depicted in Fig. 2. Each input is generated by sampling a parent and, if present, a child, resulting in one or two active concepts.

$$\boldsymbol{x} = z_p \cdot \boldsymbol{D}_p + z_c \cdot \boldsymbol{D}_c$$

The activations $z_p$ and $z_c$ follow the activation pattern defined in Definition 4.1, taking values from $\mathcal{N}(1.5, 1/4^2)$ (ensuring positive values) when active and 0 otherwise.[3]

We analyze Vanilla (ReLU) [23, 14], BatchTopK [17], and Matryoshka [20, 94] all trained with fixed $\ell_0$ sparsity targets. In a fixed low intra-level interference regime (Fig. 4A), Vanilla and BatchTopK suffer from *feature absorption* [95], aligning child concepts with their parent and collapsing hierarchy—though they retain within-level corelation. Matryoshka avoids absorption and preserves hierarchy but introduces *negative interference* between siblings, distorting flat structure. Only MP-SAE recovers both intra and inter level structure. To further stress-test the different SAEs, we vary within-level correlation in the ground truth and evaluate learned dictionaries using "Flat MSE" (for intra-level alignment; see Fig. 4B) and "Hierarchical MSE" (for inter-level separation; see Fig. 4C). We observe that Matryoshka tends to

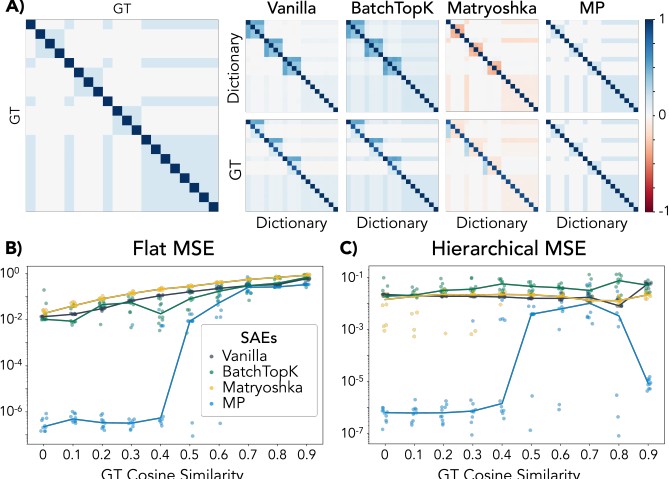

Figure 4: **Evaluating SAE on a hierarchical tree with controlled within-level similarity. A)** Correlation matrices for one similarity setting. Left shows the ground-truth matrix; the top row displays $D^\top D$ (self-similarity of learned features) and bottom row shows $D_{GT}^\top D$ (alignment with ground truth). **Bottom:** Quantitative evaluation across varying levels of within-group correlation, median over 10 runs is reported. **B)** Flat MSE captures the deviation from the ground-truth intra-level correlation. **C)** Hierarchical MSE quantifies unintended correlations across levels.

preserves hierarchy but loses flat structure; meanwhile, Vanilla and BatchTopK behave oppositely. MP-SAE exhibits three regimes: low to moderate interference yields recovery of both structures; under high interference, both degrade, reaching eventually a point where we see MP-SAE sacrifice flatness to maintain hierarchy, highlighting its inductive bias.

Overall, this synthetic benchmark elicits challenges towards capturing hierarchical features via existing SAEs, including ones trained with explicit hierarchy-promoting objectives. Instead, capturing the relevant inductive bias via MP-SAE works really well: MP-SAE is uniquely capable of recovering hierarchically structured features when such property statistically exists. We now turn to pretrained representations to assess whether such structure arises in practice and how SAEs respond to it.

## 4.2 Representations from Pretrained Models

We analyze representations from pretrained vision-language models in the following experiments. This helps us avoid the challenge of flexibility of model representations in solely language-driven domains [73] (though see Sec. B.5 for preliminary results) Our evaluation is organized in four parts. We begin by (*i*) assessing MP-SAE's expressivity relative to existing SAEs. We then (*ii*) analyze the structure of its learned representations through effective rank and coherence metrics. Next, (*iii*) we test its robustness to inference-time sparsity variation. Finally, (*iv*) we investigate its ability to uncover multimodal features in vision-language models that remain inaccessible to existing SAEs.

**Expressivity.** We begin by evaluating the expressivity of different SAEs, including MP-SAE. Training was performed for 50 epochs with Adam, using a learning rate of $5 \cdot 10^{-4}$ and cosine decay to $10^{-6}$ with warmup. Models were trained on IN1K [96] train set, using frozen representations from the final layer of each backbone. For ViT-style models (e.g., DINOv2), all spatial tokens and the CLS token were included ($\sim$261 tokens per image for DinoV2, which results in approximately 25 billion training tokens for a training).

---

[3]We generalize the generative process from Bussmann et al. [20], where $z_p$ and $z_c$ are fixed across all inputs, making their firing magnitudes perfectly correlated ($z_p = \lambda z_c$). See Appendix B.1.4 for an extended discussion.

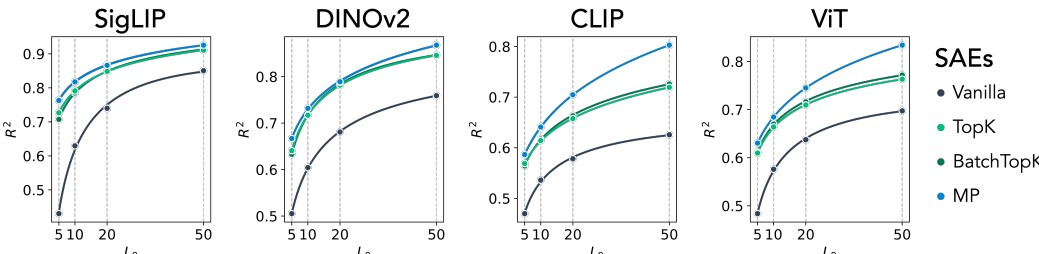

Figure 5: **MP-SAE recovers more expressive features than standard SAEs.** Reconstruction performance ($R^2$) as a function of sparsity level across four pretrained vision models: SigLIP, DINOv2, CLIP, and ViT. MP-SAE consistently achieves higher $R^2$ at comparable sparsity, indicating more efficient and informative decompositions.

Results are shown in Fig. 5, where we plot the Pareto frontier obtained by varying the sparsity level, using an expansion factor of 25 ($p = 25m$) for all SAEs. Across all tested models: SigLIP [97], DINOv2 [76], CLIP [75], and ViT [98], MP-SAE consistently achieves higher $R^2$ at comparable sparsity levels, indicating more efficient reconstruction. Our results above suggest that MP-SAE can explain a larger fraction of the representation space using fewer active features. Equivalently, its selected features are more informative per unit of sparsity. This aligns with the hypothesis that MP-SAE, through its iterative and residual-guided inference, can recover features that remain inaccessible to conventional SAEs. We note that, Engels et al. [71] identify a class of non-linearly accessible features that can't be recovered by linear sparse encoders. The improved expressivity observed here provides indirect evidence that MP-SAE may capture some of these otherwise hidden components. In the sections that follow, we examine more precisely the structure and semantics of the features recovered by MP-SAE to further probe this possibility.

**Emergence of Rank Structure.** To investigate how MP-SAE organizes features across varying sparsity levels, we analyze the effective rank of the co-activation matrix $\boldsymbol{Z}^\top \boldsymbol{Z}$, where $\boldsymbol{Z} \in \mathbb{R}^{n \times p}$ is the matrix of sparse codes across $n$ inputs (see Fig. 6). Each entry of $\boldsymbol{Z}^\top \boldsymbol{Z}$ captures how frequently two concepts are co-activated. The effective rank, defined as the exponential of the entropy of the normalized eigenvalues of $\boldsymbol{Z}^\top \boldsymbol{Z}$ (Eq.2, Appendix B.3), measures the diversity of feature co-activation patterns across inputs. For standard SAEs, increasing the sparsity level $k$ typically results in limited structural change: the encoder reuses similar subsets of features, leading to saturated rank and strong diagonal blocks in $\boldsymbol{Z}^\top \boldsymbol{Z}$. In contrast, MP-SAE shows a markedly different trend. As $k$ increases, the effective rank grows steadily, reflecting a continual diversification of active feature sets. The rank growth induced by MP-SAE is thus **not an artifact of increased capacity, but a structural signal**: it reflects the model's ability to discover and disentangle latent modularity within representations.

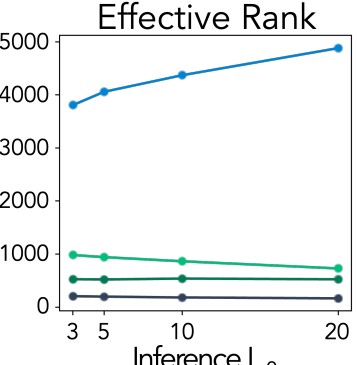

Figure 6: **Growth of effective rank as a function of sparsity** $k$. MP-SAE exhibits increasing combinatorial diversity, unlike standard SAEs whose co-activation structure quickly saturates.

**Coherence and Conditional Orthogonality.** We now examine the internal organization of the learned dictionaries by quantifying their coherence. Specifically, we use the Babel function [99], a standard metric in sparse approximation that captures cumulative interference between features. Given a dictionary $\boldsymbol{D} = [\boldsymbol{D}_1, \dots, \boldsymbol{D}_p] \in \mathbb{R}^{m \times p}$ with unit-norm columns, the Babel function of order $r$ is defined as:

$$\mu_1(r) = \max_{\substack{S \subset [p] \\ |S| = r}} \left( \max_{j \notin S} \sum_{i \in S} \left| \boldsymbol{D}_i^\top \boldsymbol{D}_j \right| \right)$$

Intuitively, $\mu_1(r)$ reflects how well a single concept can be approximated by a group of $r$ others; lower values indicate better separability. Fig. 7 reports $\mu_1(r)$ for the learned dictionary, as well as the average over multiple representations for subsets of co-selected concepts at inference time. Interestingly, MP-SAE learns dictionaries with higher overall Babel scores than standard SAEs,

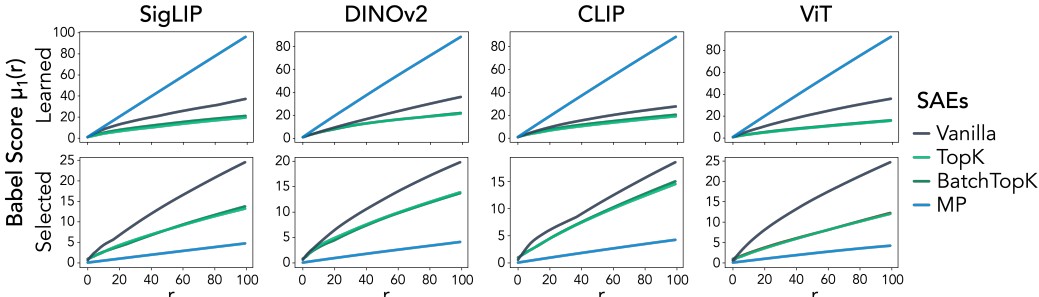

Figure 7: **MP-SAE promotes conditional orthogonality at inference.** Babel scores for full dictionaries (top) and co-activated subsets (bottom). MP-SAE dictionaries exhibit higher global coherence than standard SAEs, but select more separated features at inference.

indicating greater global interference. However, the concepts it selects during inference exhibit lower Babel scores, reflecting MP-SAE's tendency to construct conditionally orthogonal representations at inference (even when the full dictionary is correlated). By contrast, linear SAEs enforce global quasi-orthogonality in the dictionary but do not control which features co-activate at inference. As a result, inference often selects interfering directions despite a well-structured dictionary.

**Adaptive Inference-Time Sparsity.** A key property of MP-SAE is its ability to adaptively adjust the number of selected features $k$ at inference time without re-training. Because inference proceeds via residual-based greedy selection, each additional step guarantees non-increasing reconstruction error. As shown in Fig. 8, MP-SAE is the only architecture for which reconstruction fidelity improves monotonically with $k$ across all tested architectures and representation types. This stands in contrast to TopK SAEs, which degrade under sparsity mismatch: when trained with fixed $k$, the decoder implicitly specializes to superpositions of *exactly* $k$ features, resulting in instability when $k$ changes.

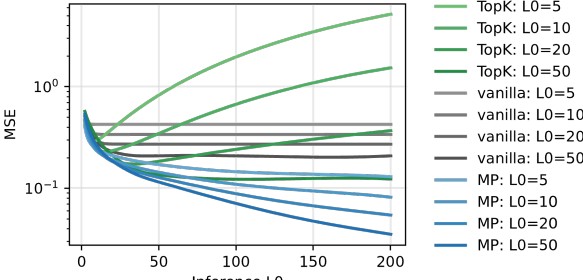

Figure 8: **Reconstruction error as a function of inference-time sparsity** $k$**.** When increasing SAEs' sparsity at inference on DINOv2 representations, we see MP-SAE exhibits monotonic improvement, while TopK SAEs may degrade due to sparsity mismatch. Similar results emerge in other settings (see Appendix B.5).

ReLU-based SAEs, by contrast, cannot extend their support beyond what was active during training and exhibit flat or plateaued performance as $k$ increases. This robustness is particularly salient given the epistemic uncertainty surrounding the "true sparsity" of neural representations. This property allows adjusting $k$ at inference time without retraining, enabling controlled trade-offs between sparsity and reconstruction quality. Rather than fixing $k$, one can target a desired reconstruction level and let the model incrementally reveal the relevant features by adapting $k$.

**Recovering Multimodal Concepts.** We evaluate whether MP-SAE can recover shared concepts from vision-language models (VLMs). Our analysis focuses on representations from CLIP [75], with consistent results observed for AIMv2 [100], SigLIP [97], and SigLIP2[101] (See Appendix B.4). All models are evaluated on the COCO dataset [102], using both image and caption embeddings, balanced to ensure equal sampling across modalities. When trained on these joint embedding spaces, classical SAEs frequently learn *split dictionaries* [103, 104], where distinct features respond exclusively to either visual or textual inputs. This occurs despite the alignment of the underlying representation space and reflects a structural limitation in how these existing SAEs extract shared features. To quantify modality selectivity, we use the Modality Score from [105], defined for concept $i$ as:

$$\text{ModalityScore}_i = \frac{\mathbb{E}_{\boldsymbol{z} \sim \iota}(z_i)}{\mathbb{E}_{\boldsymbol{z} \sim \iota}(z_i) + \mathbb{E}_{\boldsymbol{z} \sim \tau}(z_i)},$$

where $\iota$ and $\tau$ denote activations over image and text inputs, respectively. Scores near 1 indicate image specificity; near 0, text specificity; and intermediate values reflect balanced, multimodal activation.

Consistent with prior work [105], we find that standard SAEs yield sharply bimodal modality score distributions, confirming their tendency to separate modalities. In contrast, MP-SAE yields a significantly flatter distribution with substantial mass in the midrange (see Fig. 9), suggesting that it recovers genuinely multimodal units responsive to both modalities. This ability to extract multimodal concepts appears unique to MP-SAE. We hypothesize that MP-SAE's residual based inference plays a central role: once modality-specific information is explained, the residual shifts toward shared structure, allowing subsequent steps to capture cross-modal features (See Fig. 18). Prior work [104] has attempted to bridge the modality gap by applying a learned translation from one modality to the other. This can be seen as equivalent to a single inference step in MP-SAE. However, MP-SAE continues this process iteratively, progressively refining the residual and revealing joint semantic structure. This iterative mechanism enables MP-SAE to extract hierarchical and non-linearly accessible multimodal features beyond the scope of conventional SAEs.

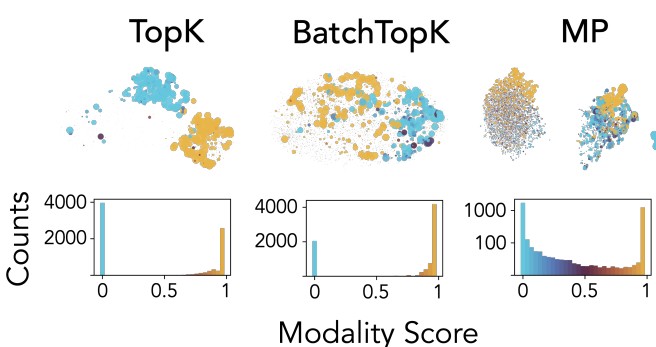

Figure 9: **MP-SAE identifies shared structures across modalities. Top:** UMAP of sparse codes (yellow denotes image representations; blue denotes text). **Bottom:** Distribution of modality scores. We see existing SAEs yield features skewed towards a specific modality, suggesting bimodal structure in model representations. However, MP-SAE uniquely recovers multimodal units with shared activation across text and vision inputs.

## 5    Conclusion

In this work, we revisit SAEs to understand their ability to identify concepts encoded in a manner outside the scope of their motivating hypothesis, i.e., LRH. Our results show that while standard SAEs are indeed effective under LRH, they struggle to capture more complex representational structures—such as hierarchical, nonlinearly accessible, and multimodal features—that have emerged in recent studies of large neural networks. To contextualize these results with respect to an SAE that is motivated to (partially) accommodate such phenomenology outside LRH's scope, we introduced MP-SAE. Specifically, an extension of the classical Matching Pursuit algorithm, MP-SAE promotes conditional orthogonality through residual-guided inference. Our results show that this design enables MP-SAE to recover rich, hierarchically organized, and nonlinearly accessible features that standard SAEs missed. Beyond interpretability, we also find that residual-based, sequential encoders like MP-SAE offer practical advantages, such as adaptive inference-time sparsity and progressive feature discovery, which may be of independent interest to the community. *Overall, we argue our results highlight the limitations of assuming purely linear structure in neural representations.*

**Limitations**    We emphasize that our goal with proposing MP-SAE was not to propose a "superior" SAE architecture, but to use it as a tool whose inductive bias aligns with a particular class of concept structures, namely, hierarchically and nonlinearly accessible features. Under this hypothesis, MP-SAE proves more expressive, more modular, and more robust than standard SAEs. However, we note MP-SAE assumes that representations can be incrementally explained by conditionally orthogonal features. This inductive bias may be poorly matched in flat or entangled regimes, or in settings with categorical structure. Matching Pursuit is also a greedy algorithm, which lacks global optimality and may be brittle under extreme noise (though we do find existing SAEs fail in this regime as well).

## Acknowledgements

Authors thank the CRISP Group at Harvard SEAS for insightful conversations, and the Kempner Institute and CBS-NTT program in Physics of Intelligence at Harvard University for access to compute resources used for performing experiments reported in this paper. Valérie Costa thanks the Bertarelli Foundation for supporting her work as a fellow.

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

# Appendix

## Table of Contents

# A  Related Work

**Sparse coding**  Sparse linear generative models aim to find a sparse representations $z \in \mathbb{R}^p$ that explains the data $x \in \mathbb{R}^m$ via a collection of atoms, forming an overcomplete dictionary $D \in \mathbb{R}^{m \times p}$ ($p \gg m$.) [106, 32]. Sparse representations are ubiquitous in science and engineering, with origins from computational neuroscience [106, 32] and applications in medical imaging [107–109], image restorations [110–112], radar sensing [113], transcriptomics [114, 115], and genomics [116, 117].

When the dictionary is fixed, finding the sparsest $z$ decomposition of the input $x$ is known as a sparse approximation problem (i.e., minimizing the $\ell_0$ norm of the representation) [118, 99, 119]. This combinatorial sparse problem is NP-hard and has been the focus of extensive research in compressed sensing [120–123] and signal processing [31, 124]. Classical approaches include greedy $\ell_0$-based algorithms such as Matching Pursuit [74] and Orthogonal Matching Pursuit [89], as well as the convex relaxation $\ell_1$-based methods [90, 125–127] such as Iterative Soft-Thresholding Algorithm (ISTA) [91, 92] and Iterative Hard-Thresholding (IHT) [128]. Since sparse recovery lacks a closed-form solution [118], the sparse approximation step typically requires multiple residual-based iterations to converge.

When the dictionary is learned jointly with the sparse codes, the problem is referred to as sparse coding [32] or dictionary learning [129, 31, 130]. Classical approaches such as MOD [131] and K-SVD [124] solve the problem using alternating minimization [132]. The bottleneck of sparse coding is in the convergence rate of the inner problem, which relates to the properties on dictionary atoms; while the slow sublinear convergence of inner iterative algorithms such as ISTA [91, 92] can be accelerated via optimization techniques (e.g., adding momentum [133]), there exist no optimization algorithm that can guarantee linear convergence for general sparse coding problem.

**Unrolling for sparse coding**  To address the slow convergence of sparse coding, in 2010, learned ISTA (LISTA) [134] sparked a new approach, which is now known as unrolling, to turn and relax the iterations of ISTA into layers of a neural network to accelerate the inner problem. Since then, numerous works have theoretically studied the linear convergence of unrolled ISTA [135, 136], and some others have highlighted the implicit acceleration that this approach may bring in learning the dictionary [137, 138]. Moreover, a parallel literature has theoretically shown that approximate sparse coding, such as a shallow ReLU network [139, 140], or shallow hard-thresholding (recently named as JumpReLU [18]) [141, 140] is practically enough to perform dictionary learning.

**Interpretability of Neural Networks**  Recently, [51] has shown that many interpretability methods used to extract concepts were in fact sparse coding methods. The field has re-emerged as a compelling framework for interpreting the internal mechanisms of high-performing, large-scale models [142, 1–10], particularly in light of persistent challenges surrounding safety and interpretability [11–13]. Within this renewed interest, Sparse Autoencoders (SAEs) have gained prominence as a principled method for exposing the latent structure of neural representations [14–23].

A significant portion of recent interpretability research is underpinned by the Linear Representation Hypothesis (LRH) [24, 25, 25], which posits that high-dimensional neural representations can be decomposed as superpositions over a large set of approximately orthogonal directions, each aligned with human-interpretable concepts. This hypothesis has been theoretically substantiated through formal complexity bounds and constructive frameworks for linearly decodable computation in superposition [26–28].

Empirical support for LRH spans both vision and language domains. In vision models, internal representations have been shown to encode factors such as geometry, lighting, facial structure, and scene composition [34–50]. In language models, analogous patterns emerge, with latent directions capturing syntactic and semantic content [52–57], character roles, arithmetic concepts [58–60], relational structures [63–66], and even safety-related behaviors such as refusal to generate harmful outputs [61, 62].

However, recent work has challenged the universality of LRH, arguing that the geometric assumptions behind SAEs may not hold in practice [67, 68]. Several studies reveal nonlinear and non-directional structures: hierarchical concepts may align with orthogonal axes, but categorical ones form overlapping polytopes [69]; RNNs exhibit "onion-like" layers [70]; and temporal concepts like weekdays often require multi-dimensional, context-sensitive representations [71–73, 21]. These findings highlight the need for improved SAE architectures to capture interpretable concepts that lie beyond the assumptions of the LRH.

# B Extended Experimental Details and Further Results

## B.1 Synthetic Experiment

The synthetic experiments were conducted using the codebase provided by Bussmann et al. [20]. Specifically, we introduced the following modifications:

- Extended the framework to support both MP-SAE and BatchTopK variants.
- Enforced mutual exclusivity among child features, yielding a sparsity of 1 or 2.
- Introduced variance in the firing magnitudes of the features (see Section B.1.4).
- Added controlled intra-level correlations between features.

Before detailing our experimental setup, we first introduce the concept of Matryoshka Sparse Autoencoders (SAEs).

**Matryoshka Sparse Autoencoders (SAEs)** Matryoshka SAEs [20, 94] extend traditional sparse autoencoders by enforcing accurate reconstruction from multiple nested prefixes of the latent vector. Given an input $x \in \mathbb{R}^m$, the encoder computes $z = \Pi\{W^\top(x - b_{\text{pre}}) + b\} \in \mathbb{R}^p$, and each prefix $m_i \in \mathcal{M}$ is used to reconstruct $x$ via the first $m_i$ latents:

$$\hat{x}^{(m_i)} = D_{0:m_i,:} z_{0:m_i} + b_{\text{pre}}.$$

The set $\mathcal{M} = m_1, m_2, \ldots, m_n$ defines a hierarchy of prefix lengths such that $m_1 < m_2 < \cdots < m_n = p$, where each $m_i$ corresponds to a sub-SAE tasked with reconstructing the input from a progressively longer portion of the latent code. In practice, $\mathcal{M}$ can be fixed before training or sampled stochastically per batch to encourage robustness across multiple levels of abstraction.

The total loss encourages reconstructions at multiple levels:

$$\mathcal{L}(x) = \sum_{m_i \in \mathcal{M}} \|x - \hat{x}^{(m_i)}\|_2^2 + \alpha \mathcal{L}_{\text{aux}}.$$

This design encourages early latents to encode general features and later ones to specialize, improving disentanglement and reducing feature absorption [95] in hierarchical settings.

### B.1.1 Dataset Generation

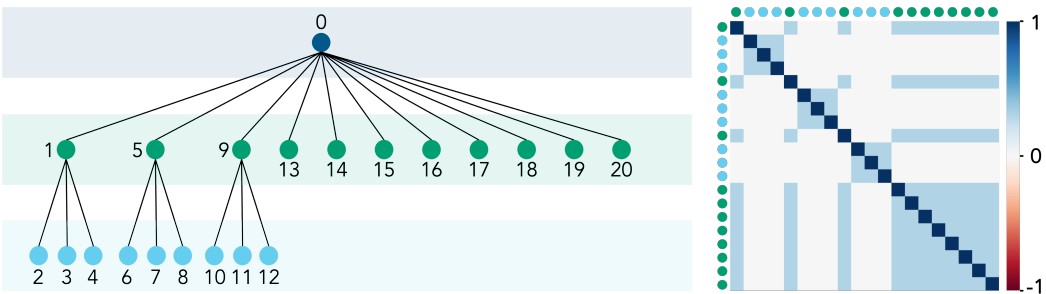

Figure 10: Hierarchical tree structure (from [20], also shown in Figure 4) with node indices. Indices are omitted in subsequent figures to avoid visual clutter.

The tree structure contains 20 nodes, each corresponding to a unit-norm vector in $\mathbb{R}^{20}$, as illustrated in Figure 2 and detailed in Figure 10. The exact tree configuration used in our experiments is available on the project's GitHub repository[4].

**Feature Activations** The root node (index 0) is fixed to the zero vector, i.e., $0 \in \mathbb{R}^{20}$. All child nodes are mutually exclusive, leading to input sparsity levels of either 1 or 2. Leaf nodes—green parents without children (indices 13–20)—are independently activated with probability 0.05 ($\mathbb{P}[z_p >$

---

[4] https://github.com/mpsae/MP-SAE

$0] = 0.05$). Internal green parent nodes (1, 5, and 9) are activated with probability 0.2 ($\mathbb{P}[z_p > 0] = 0.2$). Blue child nodes (2–4, 6–8, 10–12) are activated with probability 0.2 ($\mathbb{P}[z_c > 0 \mid z_p > 0] = 0.2$), conditional on their corresponding parent (1, 5, or 9) being active, in line with Definition 4.1. This results in an expected target sparsity of $\ell_0 = 1.36$.

**Activation Magnitudes**   If a node is active, its sparse code is drawn from a Gaussian distribution $z \sim \mathcal{N}(1.5, 1/4^2)$. This differs from the generative process of Bussmann et al. [20], where $z_p$ and $z_c$ are fixed across all inputs, making their firing magnitudes perfectly correlated ($z_p = \lambda z_c$). We provide a justification for this design choice in Section B.1.4. The input $x$ is then constructed as the sum of the active node vectors scaled by their corresponding code values:

$$x = z_p \cdot D_p + z_c \cdot D_c.$$

**Introducing Intra-Level Correlations.**   In the original codebase, all dictionary directions are orthogonal, obtained via QR decomposition. To introduce correlations specifically between nodes at the same hierarchical level, we perturb each dictionary element as

$$D_i \leftarrow (1 - \varepsilon)D_i + \varepsilon \sum_{j \neq i} D_j,$$

where the sum is restricted to elements sharing the same parent, followed by renormalization. The parameter $\varepsilon$ is tuned empirically to achieve the desired degree of intra-level correlation.

### B.1.2   Training

Our training pipeline follows the same procedure as that used by Bussmann et al. [20]. Samples are drawn from the tree structure in batches of 200 over 15,000 training steps, using the Adam optimizer with $\beta = (0.5, 0.9375)$ and a learning rate of $3 \times 10^{-2}$. To stabilize optimization, gradient norms are clipped at 1.

For sparsity control, all SAEs, except MP-SAE, are trained to match a target $\ell_0$ sparsity of 1.36.

- Vanilla and Matryoshka SAEs begin with a low $\ell_1$ regularization during the first 3,000 steps, which is subsequently increased or decreased based on whether the observed sparsity is below or above the target.
- BatchTopK SAE is trained with a fixed sparsity of 3 for the initial 1,000 steps, after which the sparsity level is reduced to 1.36 for the remainder of training.
- MP-SAE, in contrast, does not need to rely on a fixed sparsity target. (make it sound more like an advantage)Instead, it unrolls the encoder until either the residual norm drops below 0.05 or the support of the sparse code $z$ stabilizes—avoiding infinite loops when the residual cannot be sufficiently reduced. Unlike the other SAEs, MP-SAE uses tied weights.

### B.1.3   Detailed Results on Synthetic Data

In this section, we provide detailed results from the synthetic experiments (Figure 4). Figure 11a displays the ground truth correlation matrices used to construct dictionaries with varying levels of intra-level correlation; we focus here on four representative settings: 0, 0.3, 0.6, and 0.9.

All models are evaluated using a fixed sparse code $z$, shown in Figure 11b, which depicts the ground truth activation matrix $z \in \mathbb{R}^{N \times p}$. When varying the degree of intra-level correlation, Vanilla SAE and BatchTopK consistently exhibit feature absorption. Matryoshka SAE sometimes alleviates this issue but does so by suppressing intra-level correlations entirely. In contrast, MP-SAE reliably recovers both the hierarchical structure and intra-level correlations at low to moderate levels (0, 0.3) and at high correlation levels (0.9), it prioritizes preserving the hierarchy over matching correlation. We summarize our detailed observations for each correlation level below:

- **Figure 12 (Correlation = 0):** We observe feature absorption in both the Vanilla SAE and BatchTopK models. Specifically, features corresponding to child nodes are often aligned with those of their parent nodes, as evidenced by prominent horizontal structures in the second row of Figure 12. This indicates that the recovered child features (2–4,6–8,10–11) are not disentangled from their respective parents (1,5,9). While the Matryoshka SAE occasionally mitigates this issue (see rightmost runs in the Matryoshka column of Figure 12), it does so by introducing

negative correlations between features at the same level, which can result in partial misalignment. In contrast, MP-SAE perfectly recovers the ground truth dictionary without exhibiting feature absorption. In terms of support recovery and sparse code estimation, all methods (excluding the failed runs of Matryoshka in rows 2 and 4) correctly identify the active set. However, in Vanilla SAE and BatchTopK, feature absorption leads to underestimation of parent activation values when a child is active. When successful, Matryoshka mitigates this issue. MP-SAE achieves exact recovery, with the inferred sparse codes $\hat{z}$ matching the ground truth $z$ both in support and magnitude.

- **Figure 13 (Correlation = 0.3):** Feature absorption remains a consistent issue in both Vanilla SAE and BatchTopK. Nonetheless, these models partially reflect the ground truth intra-level correlations among the green parent nodes without children (13–20). Matryoshka SAE continues to exhibit inconsistent behavior: while it occasionally reduces feature absorption, it fails to consistently capture intra-level dependencies. In contrast, MP-SAE maintains perfect recovery, accurately reconstructing the sparse codes with $\hat{z} = z$ and showing no evidence of feature absorption.

- **Figure 14 (Correlation = 0.6):** Vanilla SAE and BatchTopK partially capture intra-level correlations, with correlation values closest to the ground truth among the blue child nodes. However, these correlations remain weaker or less accurate for other node types. Matryoshka SAE continues to struggle, failing to recover meaningful intra-level dependencies. In contrast, MP-SAE achieves partial recovery of intra-level correlations—particularly among parent nodes without children—and more faithfully preserves the hierarchical organization of the dictionary through its inductive bias toward conditional orthogonality.

- **Figure 15 Correlation = 0.9):** At high levels of intra-level correlation, all SAEs exhibit degraded performance in recovering features. Vanilla SAE and BatchTopK still capture some similarity among the last-layer children but tend to flatten the hierarchy by focusing on correlated low-level features while neglecting higher-level ones. MP-SAE, interestingly, separates each hierarchical level into two, resulting in four orthogonal sublevels composed of 1, 10, 3, and 6 features, respectively. Within each ground-truth level of correlated features, MP-SAE learns one anchor direction capturing the shared structure, and additional orthogonal directions that describe deviations from this anchor to reconstruct the remaining features, as observed in the correlation matrices $D^\top D$. Since it is not trained with a fixed target sparsity, this separation is achieved by increasing the support: most inputs now activate 2–4 components compared to the ground-truth sparsity of 1–2. This behavior preserves conditional structure even under strong entanglement—at the cost of reduced correlation—illustrating MP-SAE's bias toward maintaining hierarchical organization when representations are highly coupled.

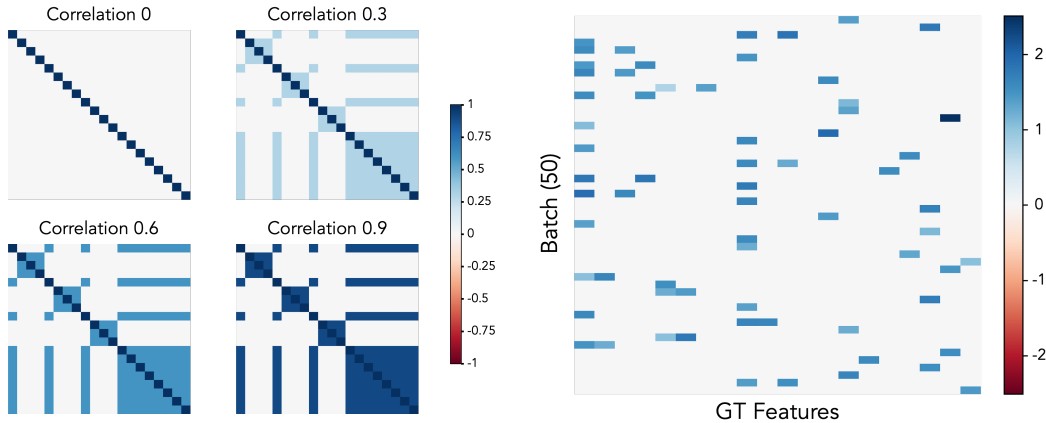

(a) Ground truth correlation matrices used in the experiments, corresponding to four levels of intra-level correlation: 0, 0.3, 0.6, and 0.9.

(b) Ground truth reference activation matrix $z$ used for generating synthetic inputs in the following experiments.

Figure 11: **Ground truth structures used for the synthetic experiments.** (a) shows the correlation levels introduced among dictionary elements at each setting. (b) shows the corresponding ground truth activation patterns used to construct the inputs.

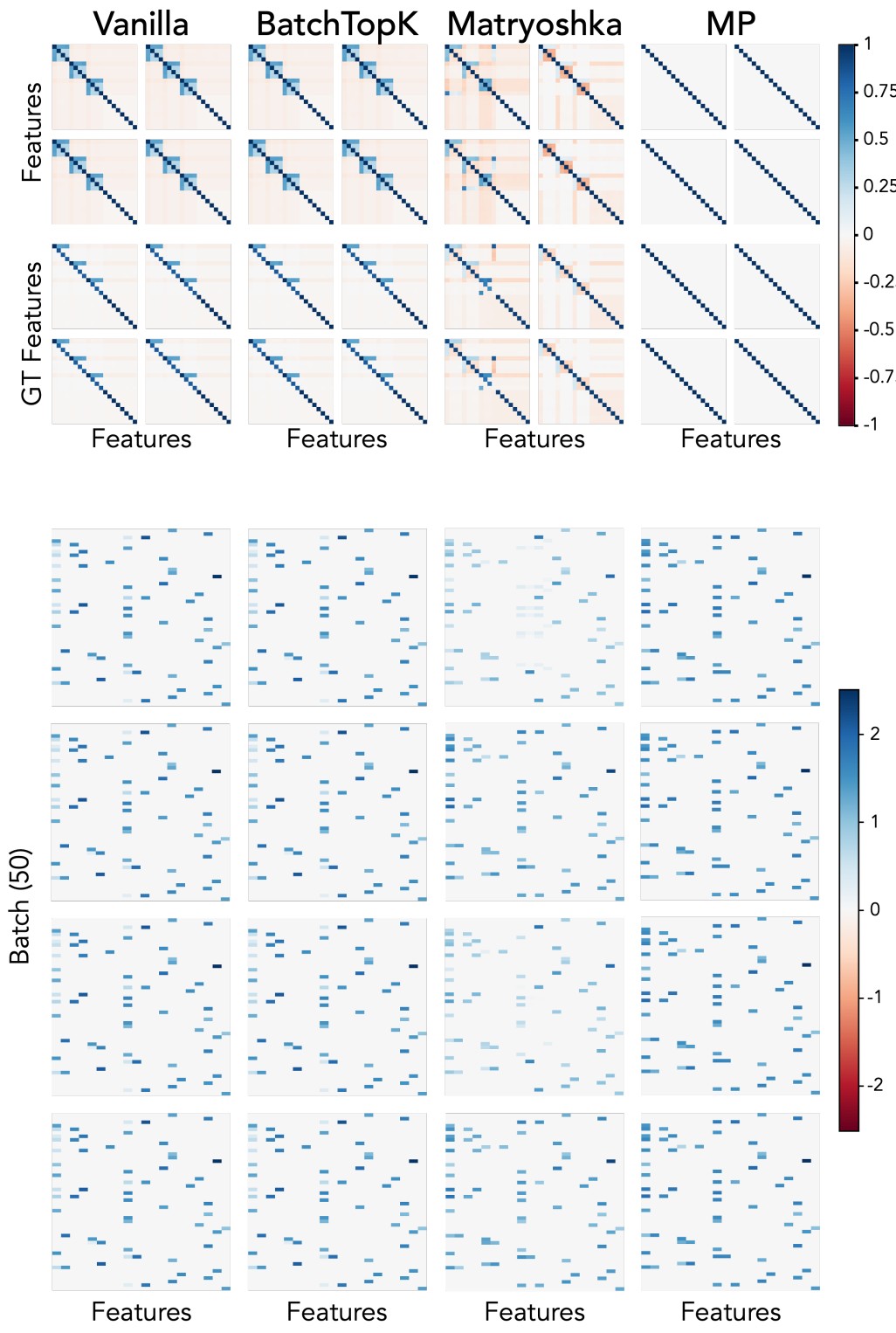

Figure 12: **4 runs for correlation level 0. Top:** Correlation matrices — the top row shows $D^\top D$ (self-similarity of learned features), and the bottom row shows $D_{\mathrm{GT}}^\top D$ (alignment with ground truth). **Bottom:** Recovered sparse codes $\mathbf{z}$.

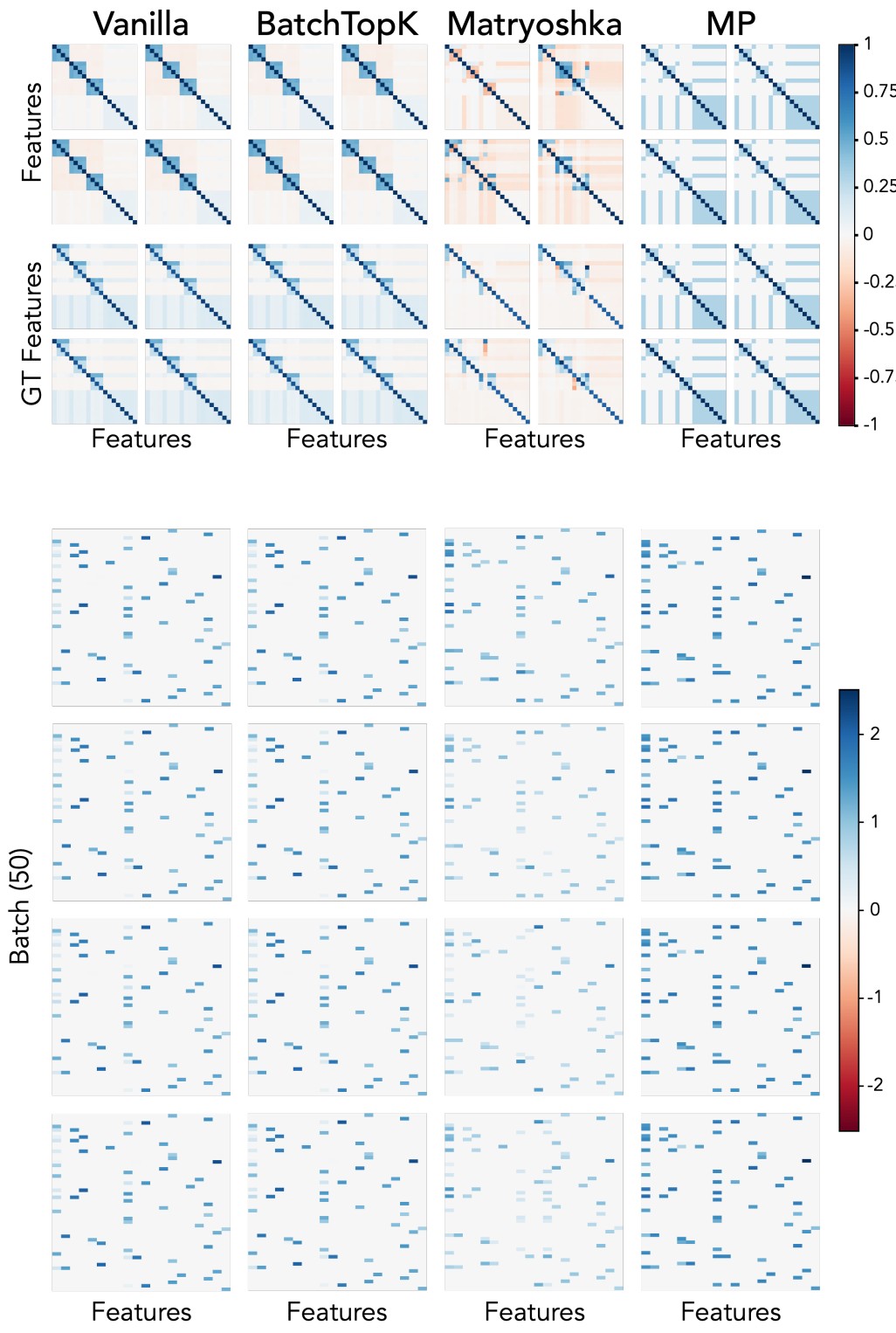

Figure 13: **4 runs for correlation level 0.3. Top:** Correlation matrices — the top row shows $D^\top D$ (self-similarity of learned features), and the bottom row shows $D_{\mathrm{GT}}^\top D$ (alignment with ground truth). **Bottom:** Recovered sparse codes $\mathbf{z}$.

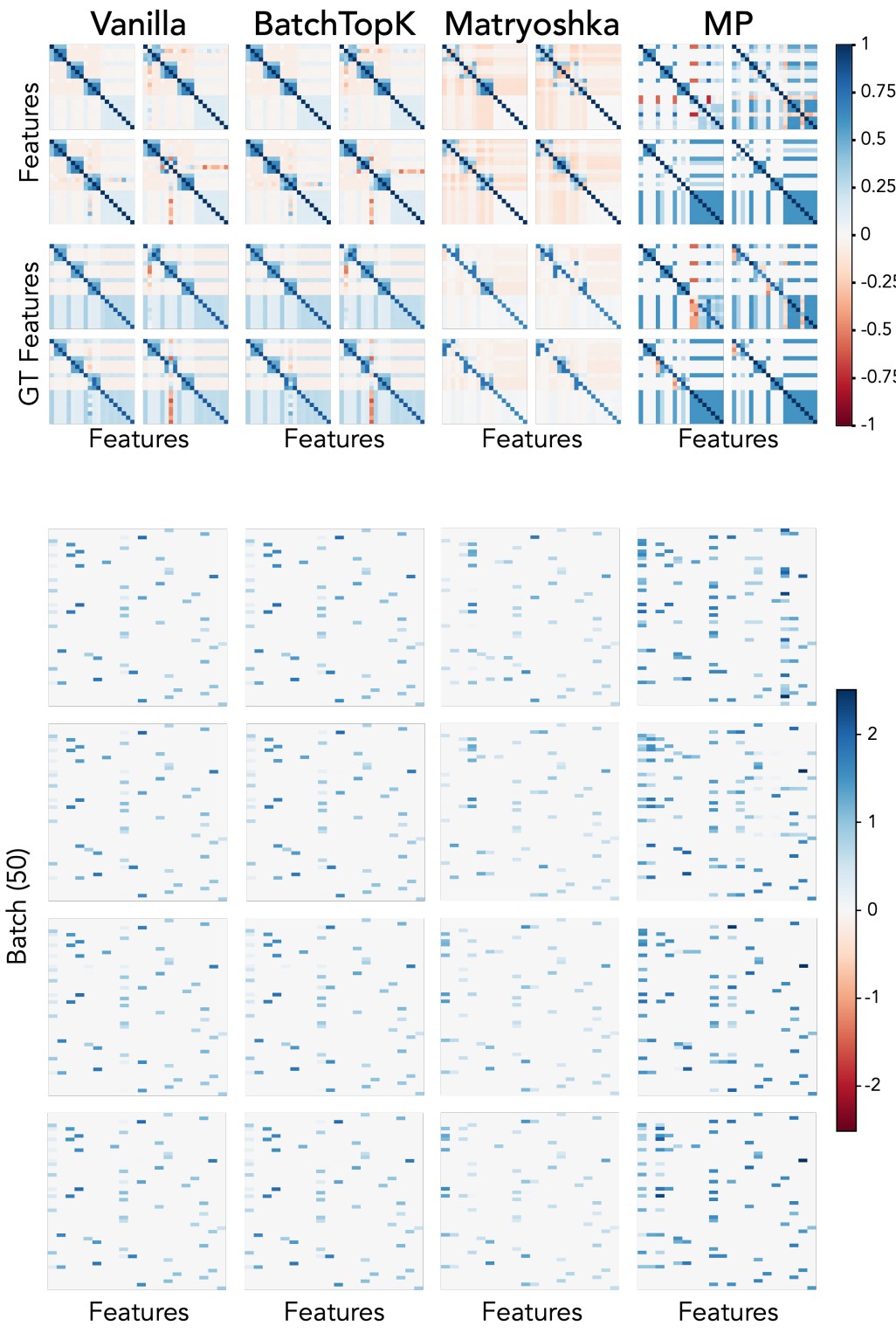

Figure 14: **4 runs for correlation level 0.6. Top:** Correlation matrices — the top row shows $D^\top D$ (self-similarity of learned features), and the bottom row shows $D_{\mathrm{GT}}^\top D$ (alignment with ground truth). **Bottom:** Recovered sparse codes **z**.

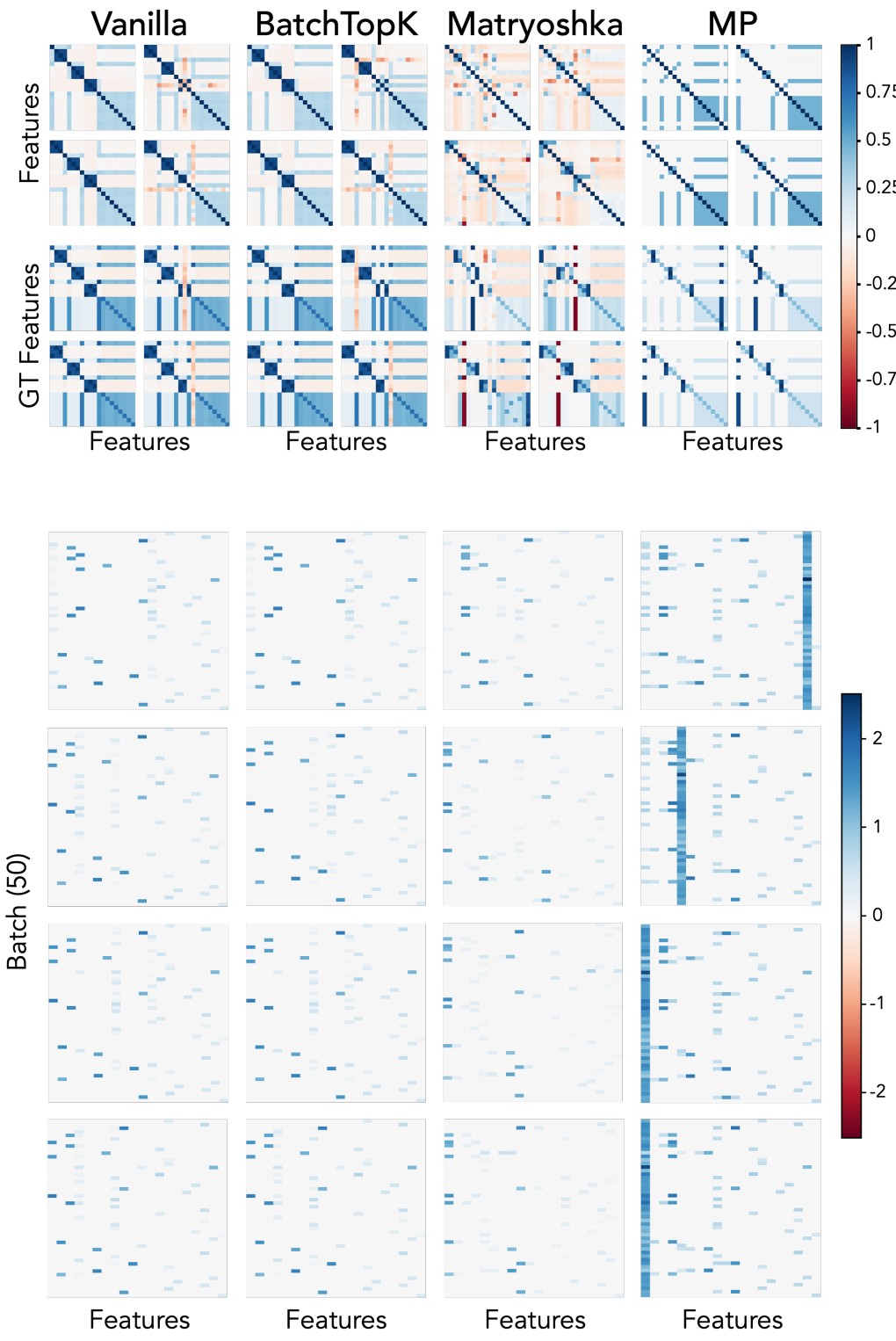

Figure 15: **4 runs for correlation level 0.9. Top:** Correlation matrices — the top row shows $D^\top D$ (self-similarity of learned features), and the bottom row shows $D_{\mathrm{GT}}^\top D$ (alignment with ground truth). **Bottom:** Recovered sparse codes $\mathbf{z}$.

### B.1.4 Introducing Variance in Firings

In the original codebase of Bussmann et al. [20], activation magnitudes are fixed across all inputs. This implicitly assumes that parent and child firings are perfectly correlated:

$$z_p = \lambda z_c.$$

Such a configuration defines a strict hierarchical setting that inherently favors *feature absorption*. In contrast, our formulation does not impose any constraint on the correlation between firing magnitudes across concepts. Our setup generalizes the original formulation by removing the perfect correlation assumption. This is achieved by introducing variance in the activation magnitudes while maintaining the sole structural assumption that a child feature can only be active if its parent is active (Definition 4.1).

**Rationale.** To illustrate the consequences of assuming perfect correlation in activation magnitudes, consider two concepts: *"red object"* (parent) and *"apple"* (child). The child can only activate when the parent does (assuming all apples are red). If the parent's firing magnitude $z_p$ represents color intensity and the child's $z_c$ represents size, assuming $z_p$ and $z_c$ are perfectly correlated is equivalent to stating that larger apples are always redder. This perfectly merges the two informational components—color intensity ($z_p \boldsymbol{D}_p$) and apple size ($z_c \boldsymbol{D}_c$)—into a single absorbed direction $z_p \boldsymbol{D}_p + z_c \boldsymbol{D}_c$, thereby eliminating their hierarchical distinction. In this extreme case, it becomes unclear whether the absorbed feature should still be interpreted as a child, since its activation is perfectly predictable from the parent. We argue that imperfect correlation between $z_p$ and $z_c$ is a necessary condition for preserving hierarchical interpretability.

**Implications for Sparse Methods.** Following Occam's razor, an $\ell_0$-penalized method such as MP-SAE will optimally encode both parent and child with a single feature when their activations are nearly identical, since the absorbed feature forms a highly compressible direction [143]. Encoding the absorbed feature $z_p \boldsymbol{D}_p + z_c \boldsymbol{D}_c$ yields a sparsity of 1, whereas disentangling parent and child requires a sparsity of 2. Therefore, it is optimal for MP-SAE to represent the parent and child through a single absorbed feature when it can perfectly reconstruct $\boldsymbol{x}$ with lower sparsity, which is precisely what occurs when no firing variance is introduced, as in the original benchmark. However, because of the greedy nature of Matching Pursuit, MP-SAE may still learn to disentangle the parent and child, depending on initialization. Figure 16 illustrates this phenomenon: it shows the correlation matrices and ground-truth alignment for the unmodified Matryoshka SAEs benchmark, trained for 15,000 iterations with a threshold of $T = 0.6$ for MP-SAE.

**Additional results with different child firing distributions.** In our generative data process, both parent and child activations initially follow the same distribution, $z_p \sim \mathcal{N}(1.5, 1/4^2)$ and $z_c \sim \mathcal{N}(1.5, 1/4^2)$. However, high variance combined with equal means ($\mu_p = \mu_c$) can produce edge cases where a strong child and a weak parent yield $\boldsymbol{x} = z_p \boldsymbol{D}_p + z_c \boldsymbol{D}_c \approx z_c \boldsymbol{D}_c$, effectively making the parent contribution negligible and thus deviating from Definition 4.1.

To assess the robustness of MP-SAE under varying child activation statistics, we evaluated its performance across a grid of child means and variances, while keeping the parent distribution fixed as $z_p \sim \mathcal{N}(1, 1/4^2)$. For each configuration, we computed an *absorption score*, defined as the average cosine similarity between the learned child features and their corresponding ground-truth parents over 15 independent runs. Lower similarity indicates successful disentanglement, whereas higher similarity reflects feature absorption. NaN values correspond to cases where the model failed to recover the child feature due to its very low firing amplitude. All activations were constrained to remain positive when active (entries marked "–" in Table 1 denote untested configurations).

Across all evaluated conditions, MP-SAE consistently achieves low absorption scores, confirming its robustness and its ability to recover hierarchical structure even under challenging firing regimes. Indeed, when the child average firing magnitude is particularly low (0.1), MP-SAE remains the only method capable of recovering the hierarchical structure, owing to its iterative greedy encoder. Interestingly, when the variance of the child distribution decreases for a mean of 0.5—a regime where overall performance becomes comparable to Matryoshka—the two methods exhibit opposite trends: Matryoshka performs better at low variance, whereas MP-SAE improves as variance increases. This observation suggests that the two approaches embody complementary inductive biases: Matryoshka is

particularly effective in scenarios dominated by strong feature absorption and near-perfect correlation, while MP-SAE excels when moderate variance is present in the firing magnitudes.

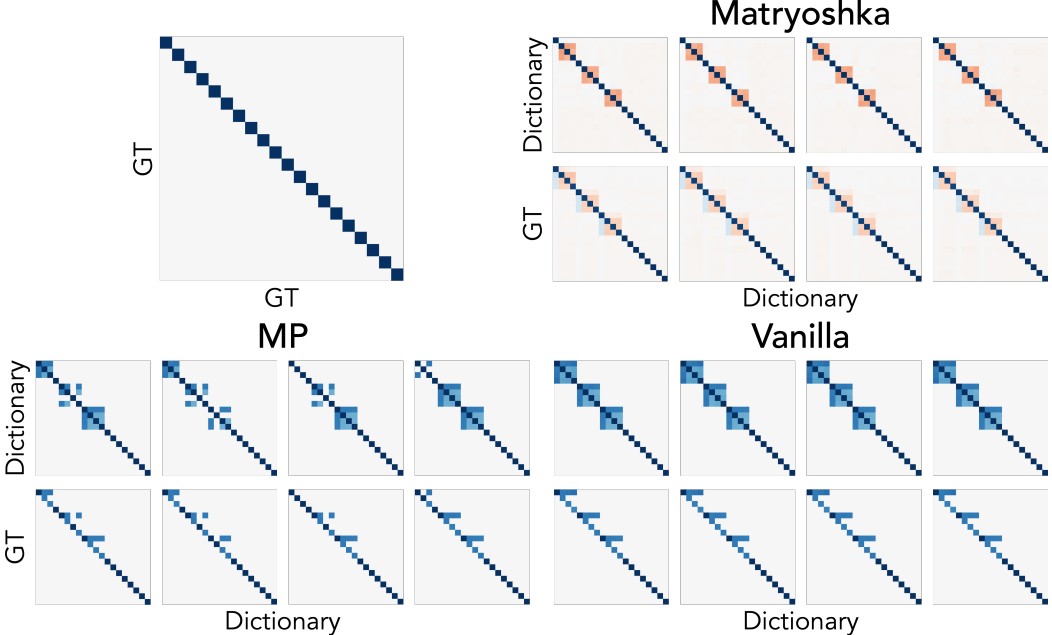

Figure 16: **Evaluating MP-SAE on the original Matryoshka benchmark[5], where firings are perfectly correlated.** For each SAE, four independent runs are shown. The top row displays $D^\top D$ (self-similarity of learned features), and the bottom row shows $D_{\text{GT}}^\top D$ (alignment with ground truth). **Top left:** ground-truth correlation matrix, where features are perfectly orthogonal. **Matryoshka** overcomes feature absorption at the cost of negative interference. **Vanilla** consistently learns absorbed features. **MP-SAE** occasionally recovers parent–child pairs thanks to its inductive bias, but under perfect correlation, its behavior is unstable.

Table 1: **Comparison of absorption scores across sparse autoencoders for different child activation distributions.** Child activations follow normal distributions with varying means (rows) and standard deviations (columns), while the parent distribution is fixed as $z_p \sim \mathcal{N}(1, 1/4^2)$. Lower absorption scores indicate better disentanglement between parent and child features. MP-SAE maintains low absorption across settings and remains the only method robust to low mean child firings.

| MP | 0.1 | 0.5 | 1.0 | Matryoshka | 0.1 | 0.5 | 1.0 |
|---|---|---|---|---|---|---|---|
| 1/400 | 3.6e-03 | 7.3e-02 | 1.3e-03 | 1/400 | nan | 4.8e-02 | 2.1e-01 |
| 1/40 | 5.0e-03 | 6.2e-02 | 1.2e-03 | 1/40 | nan | 9.0e-02 | 1.5e-01 |
| 1/8 | – | 4.3e-02 | 1.1e-03 | 1/8 | – | 9.6e-02 | 2.1e-01 |
| 1/4 | – | – | 1.1e-03 | 1/4 | – | – | 2.2e-01 |

| Vanilla | 0.1 | 0.5 | 1.0 | BatchTopK | 0.1 | 0.5 | 1.0 |
|---|---|---|---|---|---|---|---|
| 1/400 | nan | 7.0e-01 | 4.5e-01 | 1/400 | nan | 5.0e-01 | 4.8e-01 |
| 1/40 | nan | 7.0e-01 | 4.5e-01 | 1/40 | nan | 5.4e-01 | 4.5e-01 |
| 1/8 | – | 7.0e-01 | 4.7e-01 | 1/8 | – | 4.9e-01 | 4.6e-01 |
| 1/4 | – | – | 4.7e-01 | 1/4 | – | – | 4.2e-01 |

---

[5]https://github.com/noanabeshima/matryoshka-saes

## B.2 Large Vision Experiment

We train our vision SAEs on activations from four pretrained vision and VLM models: CLIP, DINOv2, SigLIP, and ViT. Representations are extracted from the final layer of each model, utilizing all spatial tokens (e.g., DINOv2 yields approximately 261 tokens per image). Each SAE processes these token-wise activations independently. Training is conducted on the ImageNet-1K training set, comprising around 1,3 Millions images. Over 50 epochs, this results in approximately $1.3 \times 10^6 \times 50 \times 261 \approx 16.9$ billion input tokens (for DINOv2). We employ a batch size of 8,000 tokens per step and train all models using the AdamW optimizer with a cosine learning rate schedule: the learning rate warms up from $10^{-6}$ to $5 \times 10^{-4}$ and decays back to $10^{-6}$ by the final epoch. A fixed weight decay of $10^{-5}$ is applied throughout. All SAEs utilize an expansion factor of 25, meaning the learned dictionary $\boldsymbol{D} \in \mathbb{R}^{c \times d}$ satisfies $c = 25d$, where $d$ is the dimensionality of the input activations. Each column $\boldsymbol{D}_i$ is constrained to lie on the unit $\ell_2$ ball: $\|\boldsymbol{D}_i\|_2 \leq 1$. The loss given is the standard MSE. To maintain active support coverage, a revive factor of $10^{-5}$ is added to any pre-code unit that fails to activate in a given batch, slightly increasing its pre-activation to reintroduce gradient flow. For Vanilla SAEs, we apply an adaptive $\ell_1$ penalty: if the empirical $\ell_0$ sparsity of a batch exceeds a target threshold, the $\ell_1$ regularization weight is increased to suppress overactivation. All encoder architectures consist of a one-layer linear projection followed by a ReLU activation. For the pareto results, one SAE is trained for each configuration (sparsity, models).

## B.3 Code Structure and Analysis of Dictionary

**Effective Rank of Feature Co-Activation.** To quantify the diversity of feature usage in sparse autoencoders, we compute the effective rank of the co-activation matrix $\boldsymbol{Z}^\top \boldsymbol{Z}$, where $\boldsymbol{Z} \in \mathbb{R}^{n \times p}$ contains the sparse codes across $n$ inputs. Each entry in $\boldsymbol{Z}^\top \boldsymbol{Z}$ reflects how often pairs of features are jointly active, and its spectral structure reveals how concentrated or distributed these co-activations are. The effective rank [144] is defined as:

$$\text{Erank}(\boldsymbol{Z}^\top \boldsymbol{Z}) = \exp\left(-\sum_{i=1}^{p} \tilde{\lambda}_i \log \tilde{\lambda}_i\right) \tag{2}$$

where $\tilde{\lambda}_i$ are the eigenvalues of $\boldsymbol{Z}^\top \boldsymbol{Z}$ normalized to sum to 1. This corresponds to the exponential of the Shannon entropy of the spectrum. High effective rank indicates that feature co-activations are spread across many directions (i.e., more diverse, less redundant usage), while low effective rank suggests repeated use of a few dominant feature combinations.

**Pairwise Coherence of Dictionary Features.** To further analyze the internal structure of learned dictionaries, we examine the distribution of pairwise inner products $\boldsymbol{D}^\top \boldsymbol{D}$, which captures the angular alignment between features. Figure 17 shows histograms ofof the values for dictionaries trained on the vision models. We used the SAEs from the pareto front, with a fixed inference-time sparsity of $k = 5$. We observe that dictionaries learned by standard SAEs (e.g., Vanilla, TopK) exhibit a distribution sharply peaked at zero, reflecting a strong bias toward global quasi-orthogonality. This is a direct consequence of their encoding mechanism: features are selected via a single global projection, which incentivizes mutually uncorrelated features to avoid interference.

By contrast, dictionaries learned by MP-SAE display broader distributions, including significant mass at nonzero values. This reflects an important difference in inductive bias: MP-SAE does not enforce orthogonality globally. Instead, its sequential, residual-guided encoder dynamically selects features that are orthogonal *conditioned* on prior selections. This means that the dictionary can afford to contain closely aligned features – as long as they do not co-activate for the same input.

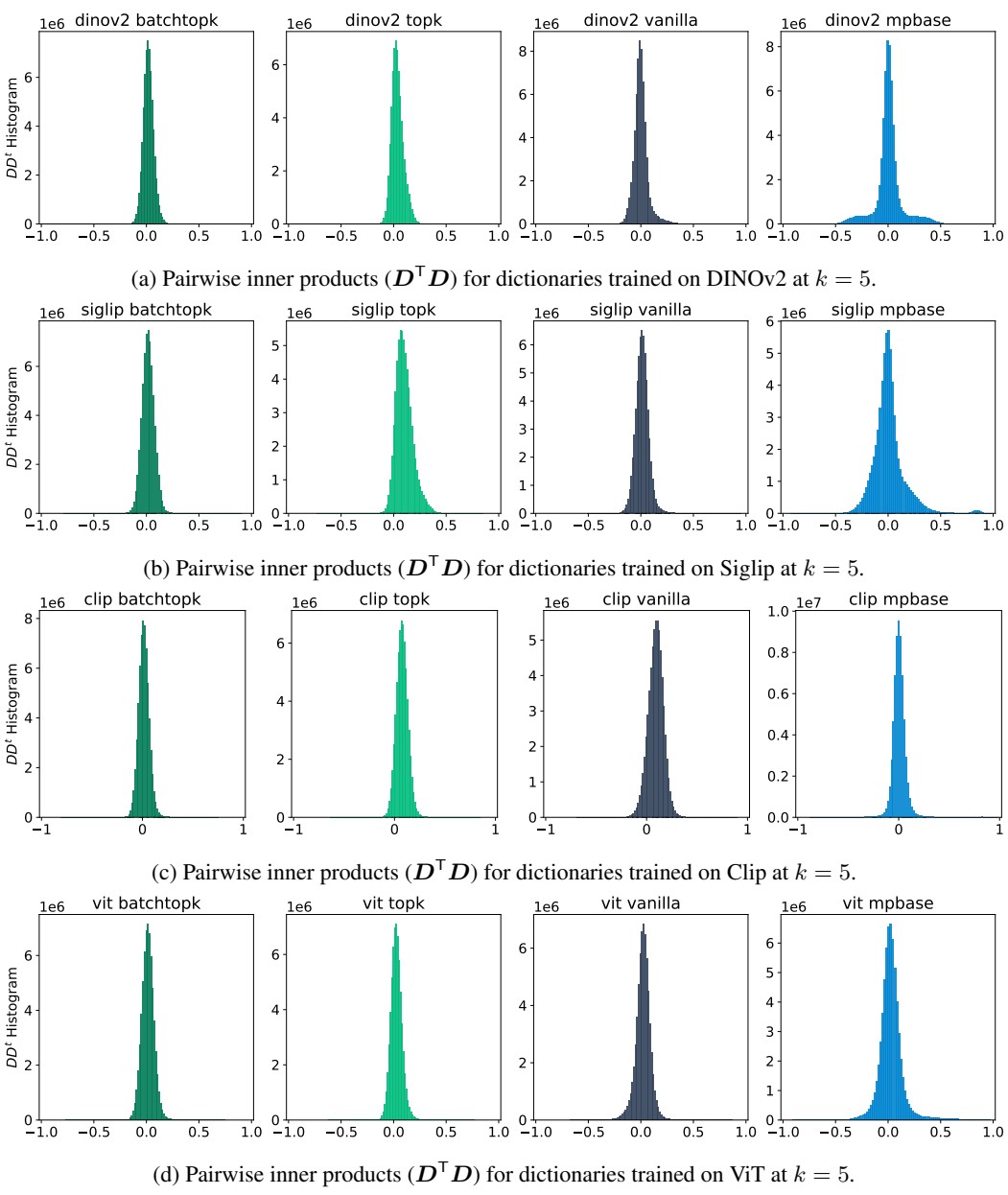

(a) Pairwise inner products ($\boldsymbol{D}^\mathsf{T}\boldsymbol{D}$) for dictionaries trained on DINOv2 at $k = 5$.

(b) Pairwise inner products ($\boldsymbol{D}^\mathsf{T}\boldsymbol{D}$) for dictionaries trained on Siglip at $k = 5$.

(c) Pairwise inner products ($\boldsymbol{D}^\mathsf{T}\boldsymbol{D}$) for dictionaries trained on Clip at $k = 5$.

(d) Pairwise inner products ($\boldsymbol{D}^\mathsf{T}\boldsymbol{D}$) for dictionaries trained on ViT at $k = 5$.

Figure 17: Distribution of dictionary coherence ($\boldsymbol{D}^\mathsf{T}\boldsymbol{D}$) reveals differing inductive biases. Standard SAEs encourage globally orthogonal features; MP-SAE tolerates correlated features and resolves interference at inference.

## B.4 Multimodal Models

**Setup.** We evaluates SAEs of 4 VLMs: CLIP, SigLIP, SigLIP2, and AIMv2. We use the final layer of the shared embedding space—that is, the common representation into which both image and caption inputs are projected. This layer reflects the aligned modality-invariant space optimized by contrastive pretraining. We train all SAEs on the full MS-COCO dataset [102], which consists of approximately 100,000 images and 500,000 associated captions. Each training example corresponds to a single embedding vector (either image or text), and models are trained jointly across both modalities using a shared dictionary (expansion factor 25). As in prior settings, we constrain each dictionary column to lie on the unit $\ell_2$ ball. All SAEs use a one-layer encoder followed by ReLU activations and are trained with mean squared error (MSE) loss. We fix the target inference-time sparsity to $\|z\|_0 = 5$. The optimization procedure mirrors that used for vision models for a total of 20 training epochs. The same revive mechanism is applied, whereby inactive units are nudged via a small additive bias ($10^{-5}$) to encourage gradient flow.

**Modality score** To quantify the extent to which learned features specialize by modality or capture shared structure, we compute the Modality Score for each feature, following the formulation of [105]. Values near 1 indicate image-specific features; values near 0 indicate text-specific features; and intermediate values reflect balanced, multimodal activation. To ensure that Modality Scores reflect relative activation patterns rather than absolute energy differences between modalities, we scale the energy of text inputs by a factor of $1/5$ before computing the above quantities. This corrects as we have 5 times more captions than images. We find that the same overall patterns emerge when we apply a modality wise normalization.

The figure 18 illustrates the modality gap and how different inference strategies shape learned representations in vision-language models. Each point on the sphere represents a direction in a shared embedding space, such as the one learned by CLIP [75], where both images and captions are aligned.

On the left, standard sparse autoencoders (SAEs) tend to learn split dictionaries [103, 104], assigning different features to each modality despite their alignment in the joint space. This results in high modality selectivity: visual and textual inputs activate disjoint sets of features.

On the right, MP-SAE progressively refines its representation by explaining modality-specific components in early steps and shifting focus to shared structure in later steps. This allows the model to discover genuinely multimodal concepts that respond to both image and text inputs, effectively bridging the modality gap and producing more balanced, semantically aligned features.

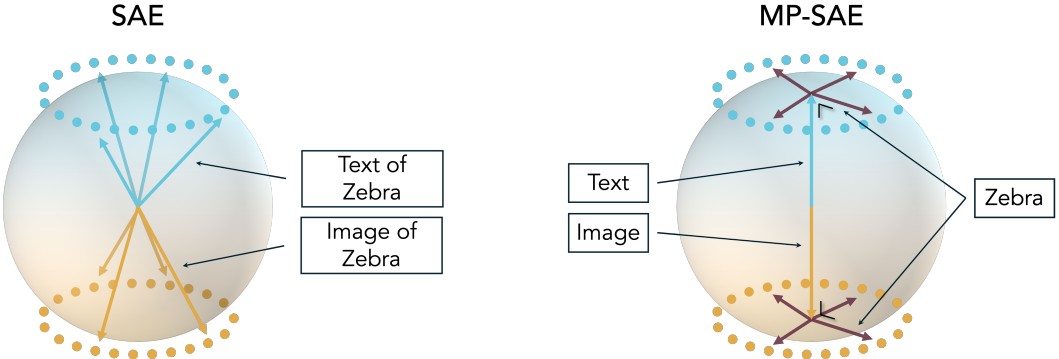

Figure 18: **Illustration of modality-selective vs. multimodal inference.** Each point represents an image or caption embedding on the unit sphere of a joint representation space. Left: standard SAEs learn split dictionaries, with features specializing in only one modality (blue = text, yellow = image), despite semantic alignment. Right: MP-SAE explains modality-specific content in early steps and aligns shared concepts in later ones. For instance, an image of a zebra and the text "a zebra in the savannah" may initially activate modal specific features, but converge toward shared features encoding the concept "zebra."

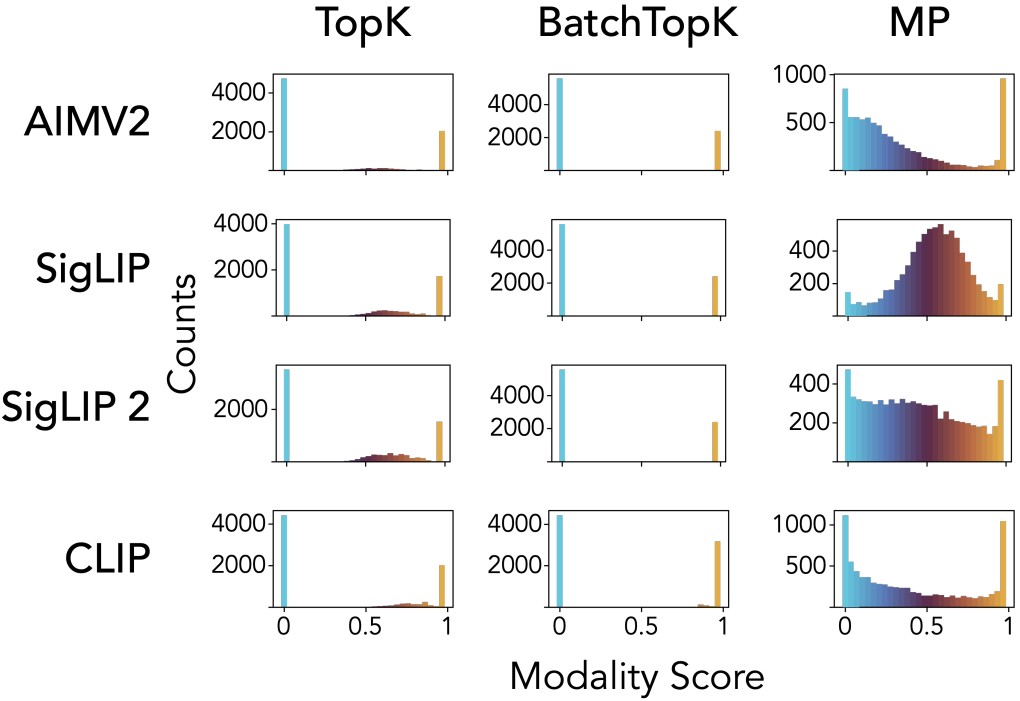

Figure 19: Distribution of modality scores for features learned by different SAEs, across four vision-language models: CLIP, SigLIP, SigLIP2, and AIMv2. Standard SAEs (e.g., TopK, BatchTopK) tend to produce bimodal distributions, with many units activating exclusively for image or text inputs, indicating a strong modality-specific specialization. In contrast, MP-SAE yields a significantly flatter distribution with substantial density near $0.5$, revealing the emergence of genuinely multimodal features that respond to both modalities. This supports the hypothesis that MP-SAE's inference allows it to progressively isolate shared semantic structure after accounting for modality-specific variance.

## B.5 Preliminary Experiments with Language Models

### B.5.1 Experimental details

To assess whether our analysis of the vision domain (see Sec. B.4) generalizes to a language modeling setting, we perform preliminary experiments using a 4-layer Transformer model [145] pretrained on the TinyStories dataset [146].[6] We train the Vanilla [14], TopK [15], and MP-SAE (see Sec. 3) on residual stream representations extracted from midway through the model, i.e., end of the second Transformer block (from amongst 4 total blocks). $p$, which refers to the width of our SAEs, i.e., the size of the sparse code $z$, is set to be $4 \times m$ for all SAEs; here, $m$ corresponds to the dimensionality of the residual stream. For TopK and MP-SAE, we set $\ell_0 = 100$ for any inputted representation; for Vanilla SAE, we search for a value of $\lambda$ to ensure after training $\ell_0$, on average, is 100. Inline with prior work [22], activations are standardized by computing a population mean and standard deviation and rescaled to be, on expectation, $\sqrt{m}$ magnitude in norm. Training is performed using Adam optimizer [147], with a constant learning rate of $10^{-3}$, $\beta_1, \beta_2 = 0.9, 0.95$, weight decay of $10^{-4}$, and gradient clipping at unit-norm magnitude for all the SAEs. Batch-size is set to be 5000 tokens, each of which is randomly sampled from samples (stories) from the train-split of TinyStories. Training goes on for approximately 40K iterations. All results in the following experiments are performed on samples drawn from the eval-split of TinyStories.

### B.5.2 Results

**Reconstruction Error with Inference $L_0$.** Similar to Fig. 8, for all SAEs, we take their parameters trained up to iteration $t$ and compare the inputted representations ($x$) with the ones

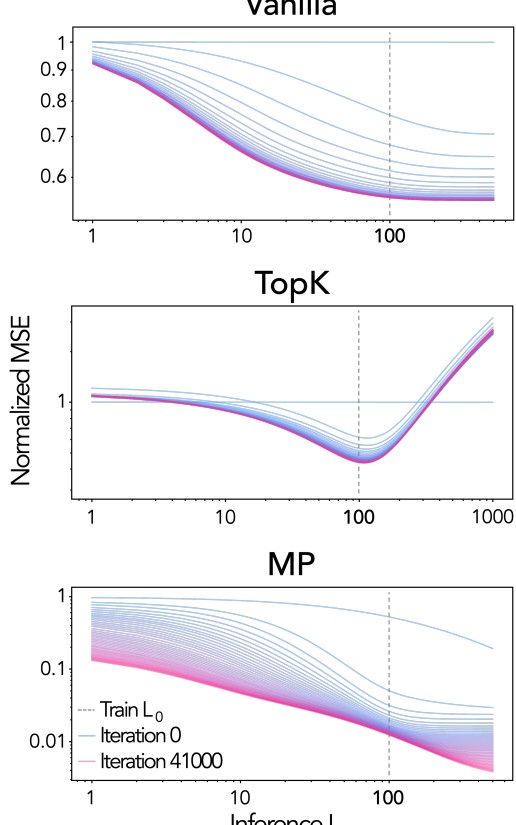

Figure 20: **Normalized MSE with increase in training iterations and inference-time sparsity.** We train Vanilla, TopK, and MP SAEs and plot the normalized MSE between their predicted vs. ground-truth representations from models trained on TinyStories. Results clearly show the flexibility of MP-SAE: increasing number of latents used for reconstruction monotonically reduce error, allowing one to use more or less latents, as desired. This property is emergent with training, however, since earlier iterations of MP-SAE do not exhibit it.

predicted by the SAE given its $k$ largest magnitude latents in the sparse code, denoted $x_{t,k}$. Specifically, we compute the **normalized MSE**, $\mathbb{E}_x \left[ ||\hat{x}_{t,k} - x||^2 / ||x||^2 \right]$, for increasing values of $t$ and $k$. We note this experiment is different from the one reported in Fig. 8 because it also assesses the effects of training, hence going beyond highlighting the inference-time flexibility of MP-SAE. Such an analysis is manageable in the current setting since models are not exceptionally big ($\sim 1$M parameters).

Results are shown in Fig. 20, where we most noticeably see a similar trend as Fig. 8: as $k$ is increased, Vanilla SAEs smoothly move towards a minimum amount of error that corresponds to the value of $k$ used for training; TopK SAEs see reduction in error up to the value of $k$ used for training, witnessing a large increase in error thereafter; and MP-SAEs again exhibit their flexibility, with error monotonically decreasing even beyond the value of $k$ used for training. It is worth noting that this flexibility of MP-SAE only emerges after enough training has occurred—in fact, earlier checkpoints exhibit a saturation or lower-bound of loss akin to the Vanilla SAEs. Finally, we emphasize again that this inference-time flexibility is a natural consequence of building on error residuals, unlike prior SAEs. Prior work hoping to elicit such a behavior had to manually train via a mixture of sparsity values [15].

---

[6]We also provide a trained MP-SAE on Gemma activations at `https://github.com/eslubana/mpsae`.

**Babel Score Analysis on Language Models.** Following the analysis in Figure 7, we compute the Babel scores for dictionaries learned on language model embeddings using MP, Vanilla, and TopK SAEs. Results are shown in Figure 21, reporting both the coherence of the full dictionary (left) and that of the concepts co-activated at inference (right).

The same trends observed in the vision models hold here as well. MP-SAE learns dictionaries with higher Babel scores globally—indicating more interference across features—but selects sets of concepts with lower mutual interference at inference time. This again reflects MP's bias toward conditional orthogonality. Conversely, Vanilla and TopK SAEs learn more globally incoherent dictionaries, but their inference-time selections often include more correlated features. Notably, the y-axis ranges in Figure 21 are similar to those from the vision experiments 7, suggesting that this behavior is not modality-specific indicating that this effect is not data dependent.

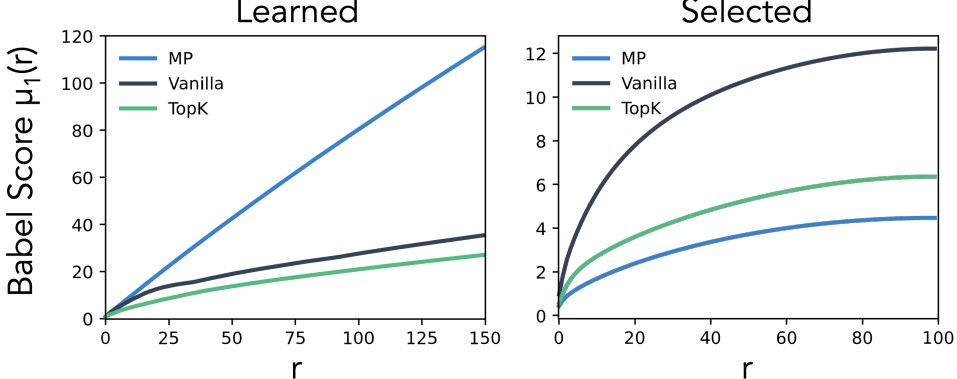

Figure 21: **Babel score analysis on language models.** We report the coherence of the full learned dictionary (left) and the subset of concepts co-activated at inference (right), across MP, Vanilla, and TopK SAEs. As in the vision setting, MP learns globally coherent dictionaries but selects less interfering concepts at inference, while Vanilla and TopK show the opposite pattern. The similarity in value ranges across modalities suggests that these effects are not data dependent.

**Effective Rank.** We compute the effective rank of the co-activation matrix $Z^\top Z$ for MP, ReLU, and TopK SAEs trained on language model embeddings.

As shown in Figure 22, MP-SAE exhibits a non-monotonic trend: the effective rank increases with sparsity, peaks at a critical point, and then declines. This suggests that MP initially promotes diverse feature use before redundancy emerges. In contrast, ReLU shows a steady decline and eventually saturates, while TopK decays slowly but drops sharply when the inference $\ell_0$ exceeds the training level. The drop in effective rank reflects increasing feature reuse and reduced diversity in the representation.

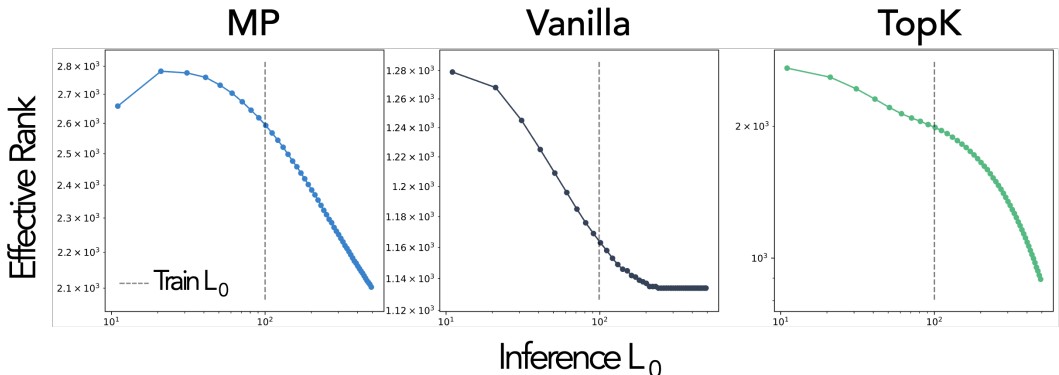

Figure 22: **Effective rank of the co-activation matrix $Z^\top Z$.** MP-SAE shows a rise-then-drop trend, indicating initial diversification followed by redundancy. ReLU steadily declines and saturates, while TopK decays slowly but drops sharply when inference sparsity exceeds training.

## C   Theoretical Guarantees for MP-SAE Inference

We restate three foundational properties of Matching Pursuit—originally established in the sparse coding literature [74]—and interpret them in the context of sparse autoencoders. These properties help elucidate the structure and dynamics of the representations learned by MP-SAE.

- **Stepwise orthogonality** (Proposition C.1): at each iteration, the residual becomes orthogonal to the feature most recently selected by the greedy inference rule. This sequential orthogonalization mechanism gives rise to a locally disentangled structure in the representation and reflects the conditional independence induced by MP-SAE inference.
- **Monotonic decrease of residual energy** (Proposition C.2): the $\ell_2$ norm of the residual decreases whenever it retains a nonzero projection onto the span of the dictionary. This guarantees that inference steps lead to progressively refined reconstructions, and enables sparsity to be adaptively tuned at inference time without retraining.
- **Asymptotic convergence** (Proposition C.3): in the limit of infinite inference steps, the reconstruction converges to the orthogonal projection of the input onto the subspace defined by the dictionary. Thus, MP-SAE asymptotically recovers all structure that is representable within its learned basis.

**Proposition C.1** (Stepwise Orthogonality of MP Residuals). *Let $r^{(t)}$ denote the residual at iteration $t$ of MP-SAE inference, and let $j^{(t-1)}$ be the index of the feature selected at step $t-1$. If the column $j^{(t-1)}$ of the dictionary $D$ satisfy $\|D_{j^{(t-1)}}\|_2 = 1$, then the residual becomes orthogonal to the previously selected feature:*

$$D_{j^{(t-1)}}^{\top} r^{(t)} = 0.$$

*Proof.* This follows from the residual update:

$$r^{(t)} = r^{(t-1)} - D_{j^{(t-1)}} z_{j^{(t)}}^{(t-1)},$$

with $z_{j^{(t)}}^{(t-1)} = D_{j^{(t-1)}}^{\top} r^{(t-1)}$. Taking the inner product with $D_{j^{(t-1)}}$ gives:

$$D_{j^{(t-1)}}^{\top} r^{(t)} = D_{j^{(t-1)}}^{\top} r^{(t-1)} - \|D_{j^{(t-1)}}\|^2 z_{j^{(t)}}^{(t-1)} = z_{j^{(t)}}^{(t-1)} - z_{j^{(t)}}^{(t-1)} = 0. \qquad \square$$

This result captures the essential inductive step of Matching Pursuit: each update removes variance along the most recently selected direction, producing a residual that is orthogonal to it. Applied iteratively, this localized orthogonality promotes the emergence of conditionally disentangled structure in MP-SAE. In contrast, other sparse autoencoders lack this stepwise orthogonality mechanism, which helps explain the trend observed in the Babel function during inference in Figure 7.

**Proposition C.2** (Monotonic Decrease of MP Residuals). *Let $r^{(t)}$ denote the residual at iteration $t$ of MP-SAE inference, and let $z_{j^{(t)}}^{(t)}$ be the nonzero coefficient selected at that step, Then the squared residual norm decreases monotonically:*

$$\|r^{(t+1)}\|_2^2 - \|r^{(t)}\|_2^2 = -\|D_{j^{(t)}} z_{j^{(t)}}^{(t)}\|_2^2 \leq 0.$$

*Proof.* From the residual update:

$$r^{(t+1)} = r^{(t)} - D_{j^{(t)}} z_{j^{(t)}}^{(t)},$$

we can rearrange to write:

$$r^{(t)} = r^{(t+1)} + D_{j^{(t)}} z_{j^{(t)}}^{(t)}.$$

Taking the squared norm of both sides:

$$\|r^{(t)}\|_2^2 = \|r^{(t+1)} + D_{j^{(t)}} z_{j^{(t)}}^{(t)}\|_2^2$$
$$= \|r^{(t+1)}\|_2^2 + 2\langle r^{(t+1)}, D_{j^{(t)}} \rangle z_{j^{(t)}}^{(t)} + \|D_{j^{(t)}} z_{j^{(t)}}^{(t)}\|_2^2.$$

By Proposition C.1, the cross term vanishes:

$$\langle \boldsymbol{r}^{(t+1)}, \boldsymbol{D}_{j^{(t)}} \rangle = 0,$$

yielding:

$$\|\boldsymbol{r}^{(t)}\|_2^2 = \|\boldsymbol{r}^{(t+1)}\|_2^2 + \|\boldsymbol{D}_{j^{(t)}} \boldsymbol{z}_{j^{(t)}}^{(t)}\|_2^2. \qquad \square$$

The monotonic decay of residual energy ensures that each inference step yields an improvement in reconstruction, as long as the residual lies within the span of the dictionary. Crucially, this property enables MP-SAE to support adaptive inference-time sparsity: the number of inference steps can be varied at test time—independently of the training setup—while still allowing the model to progressively refine its approximation. This explains the continuous decay observed in Figure 8, a guarantee not provided by other sparse autoencoders.

**Proposition C.3** (Asymptotic Convergence of MP Residuals). *Let $\hat{\boldsymbol{x}}^{(t)} = \boldsymbol{x} - \boldsymbol{r}^{(t)}$ denote the reconstruction at iteration $t$, and let $\mathbf{P}_{\boldsymbol{D}}$ be the orthogonal projector onto $\mathrm{span}(\boldsymbol{D})$. Then:*

$$\lim_{t \to \infty} \|\hat{\boldsymbol{x}}^{(t)} - \mathbf{P}_{\boldsymbol{D}} \boldsymbol{x}\|_2 = \lim_{t \to \infty} \|\mathbf{P}_{\boldsymbol{D}} \boldsymbol{r}^{(t)}\|_2 = 0.$$

This convergence result is formally established in the original Matching Pursuit paper by Mallat and Zhang [74, Theorem 1]. This result implies that MP-SAE progressively reconstructs the component of $\boldsymbol{x}$ that lies within the span of the dictionary, converging to its orthogonal projection in the limit of infinite inference steps. When the dictionary is complete (i.e., $\mathrm{rank}(\boldsymbol{D}) = m$), this guarantees convergence to the input signal $\boldsymbol{x}$.

# D  Societal Impact

Interpretability plays a critical role in the safe and trustworthy deployment of AI systems. As large-scale models are increasingly integrated into everyday technologies and used by millions of people, the risks associated with their opaque decision-making grow substantially. Without interpretability, it becomes difficult to detect biases, failures, or unintended behaviors in these powerful systems. By enabling the extraction of structured, human-interpretable features, tools like Sparse Autoencoders help researchers and practitioners understand how models encode semantics, reasoning patterns, or social attributes. This transparency is especially important in safety-critical domains such as healthcare, legal decision-making, or education, where opaque model behavior can result in harmful or unfair outcomes. Interpretability methods also support model auditing and targeted interventions, making it possible to align AI behavior with human values. However, such tools can also be misused—for example, to reverse-engineer proprietary models or infer sensitive attributes. We emphasize that our method does not amplify these existing risks.

