# OpenReview forum: "From Flat to Hierarchical: Extracting Sparse Representations with Matching Pursuit"
_NeurIPS.cc/2025/Conference — NeurIPS 2025 poster_

### Official Review · Reviewer_vRBm · 2025-06-23

**Clarity:** 3
**Significance:** 4
**Originality:** 4
**Rating:** 4
**Confidence:** 4

**Summary:**

The authors present a novel SAE architecture called Matching Pursuit Sparse Autoencoders (MP-SAEs). This architecture uses insights from sparse coding to replace the SAE encoder with a greedy algorithm to iteratively select a latent to reconstruct part of the residual. The authors claim this architecture is able to handle hierarchical relationships better than existing SAE architectures, and finds richer features than existing SAE architectures on synthetic and vision-based benchmarks.

**Questions:**

I will raise my score if the authors remove the claim that MP-SAEs handle hierarchical relationships, as this does not appear to be true. Alternatively, if the authors modify the MP-SAE architecture so it does actually handle hierarchical relationships, even when feature variance is low or none, I will also raise my score.

I will raise my score further if the authors train MP SAEs on an LLM like Gemma-2-2b, run SAEBench, and compare these results with Matryoshka SAEs. This would allow a more direct comparison with existing SOTA SAEs and make it easier to recommend MP-SAEs as the SOTA going forward. Training a MP-SAE on a real LLM would allow creating feature dashboards to give a qualitative sense of the type of features that are recovered as well.

I also have the following questions on the work:

- Conceptually I don't see how the MP-SAE actually encourages learning the parent latent rather than a parent + child (feature absorption) latent. E.g. if a there's a parent concept P and a child concept C, why wouldn't the MP-SAE learn a latent for C + P and a latent for P, and activate C + P when the child is active and P when the parent is active? In the code provided for the paper, if I reduce the feature firing variance then the MP-SAE does often learn the absorption solution rather than correctly separating the components of the hierarchy. Having parent and child features fire with high independent variance makes them look like independent features to a tied SAE, rather than the SAE actually handling hierarchy.
- The work mentions that MP-SAEs can capture non-linear features. Given the dictionary is still just linear vectors, why do the authors think that MP-SAEs are capturing non-linear concepts? Have the authors observed any specific non-linear concepts that are recovered by MP-SAEs, e.g. things like days of the week being embedded in a circle?

**Ethical Concerns:**

["NO or VERY MINOR ethics concerns only"]

**Final Justification:**

My main issue with this work was that the paper claimed MP-SAEs solve hierarchy in general, as I do not believe this is true. The authors justified this by copying the Matryoshka SAEs code and benchmark verbatim, but subtly modifying the benchmark to add extra variance to feature firings. Without this modification, MP-SAEs fail to deal with hierarchy and fail at that benchmark. Furthermore, despite modifying the benchmark to favor MP-SAEs, the authors then used the results of this modified benchmark to claim that MP-SAEs outperform Matryoshka SAEs at solving hierarchy.

The authors have agreed to the following, so I will raise my score, as I specified earlier in the discussion:

- Explain that they have changed the Matryoshka SAEs benchmark by adding extra firing variance
- Explain why they think this is justified
- Explain that MP-SAEs fail to handle hierarchy in the unmodified version Matryoshka SAEs benchmark with (near) perfectly correlated firing magnitudes, since MP-SAEs fail to solve hierarchy if the features have strongly correlated firing magnitudes.

**Limitations:**

yes

**Paper Formatting Concerns:**

no concerns

**Quality:**

2

**Strengths And Weaknesses:**

## Strengths

This paper introduces a novel SAE architecture that is fundamentally different than the simple linear encoder that is typically used currently, and this is a huge contribution to the field, especially given the promising initial results in the paper. The fact that this architecture is directly inspired by matching pursuit from sparse coding field is also exciting. This architecture is a great contribution to the field and opens up a lot of avenues of future work around selecting which latents to use explicitly to reconstruct given inputs.

## Weaknesses

While I feel this work is important, there are a number of serious weaknesses.

The most serious weakness is that the main claim of this work, that MP-SAEs solve hierarchy, is not true. This claim is supported based on toy model experiments, but these toy model experiments use a high independent variance for feature firing magnitudes. This high variance makes it impractical for the SAE to learn the absorption solution, which is mixing parent + child latents together, as the high variance means the merged solution will struggle to reconstruct this variance - basically this variance makes it look to the SAE like the features are actually independent. I find that if I reduce the variance in the toy model used in this paper, the MP-SAE no longer handles hierarchy and indeed frequently learns mixtures of parent + child features. Furthermore, it appears that this work uses the toy model setup from the Matryoshka SAEs work in their code nearly verbatim, but has modified the firing variance so it is much higher than in the original work, indicating the authors are aware that the default setting of 0 or low variance causes their solution to fail. There is no reason to assume that hierarchical features in reality have such high independent variance in their firings, and that their firing magnitudes are uncorrelated with each other, so I do not expect MP-SAEs to solve hierarchy in general.

Another weakness of this work is that no evaluations are done on LLMs, which makes it hard to compare the performance of these SAEs against existing SOTA SAEs in a real setting. Furthermore, none of the benchmarks performed investigate if the latents learned are actually interpretable (e.g. using something like a sparse-probing benchmark or even auto-interp).

Ideally the authors should train MP-SAEs on something like Gemma-2-2b and run the evaluations from SAEBench [1], ideally comparing to SOTA SAEs like Gemma Scope [3]. SAEBench also includes a feature absorption metric which evaluates how well the SAE disambiguate hierarchical features, so if MP-SAEs do in-fact handle hierarchy correctly they should achieve stellar performance on that benchmark, similar to Matryoshka SAEs. SAEBench also includes sparse probing, concept disambiguation, and auto-interp benchmarks that can be used to gauge how interpretable the learned latents are. This would also allow directly comparing performance to Matryoshka SAEs, as these also are designed to address feature hierarchy issues [4].

This is minor, but the paper motivates conditional hierarchy based on the paper "The geometry of categorical and hierarchical concepts in large language models", but my understanding is that this paper has been debunked [2].

This method is likely much less efficient than standard SAEs as the encoder requires a for-loop rather than a single forward pass to find activating features.

References
1. Karvonen, Adam, et al. "Saebench: A comprehensive benchmark for sparse autoencoders in language model interpretability." _arXiv preprint arXiv:2503.09532_ (2025).
2. Golechha, Satvik, et al. "Intricacies of Feature Geometry in Large Language Models." _The Fourth Blogpost Track at ICLR 2025_.
3. Lieberum, Tom, et al. "Gemma scope: Open sparse autoencoders everywhere all at once on gemma 2." _arXiv preprint arXiv:2408.05147_ (2024).
4. Bussmann, Bart, et al. "Learning multi-level features with matryoshka sparse autoencoders." _arXiv preprint arXiv:2503.17547_ (2025).

---

> ### Author Rebuttal · Authors · 2025-07-31
>
> We thank the reviewer for their feedback and are particularly excited that they consider this work **“a huge contribution to the field” that “opens up many avenues for future work.”**
>
> >_(Paraphrased) Authors deviated from Matryoshka’s toy setup: MP‑SAEs separate parent and child features only under high feature‑variance; with low variance, they collapse into parent+child mixtures_
>
> We thank the reviewer for raising the issue of feature variance, as it exposes **a strong and implicit assumption in the Matryoshka toy setup** that is not discussed in the original paper: **parent and child norms are perfectly correlated**. In this setup, each input is defined as $x = p + c$ with a fixed proportional norm $\|p\| = \lambda \|c\|$. This structure systematically favors absorption (i.e., learning a single parent+child latent) because the signal-to-signal overlap is exact and firing magnitudes are perfectly correlated. In such a scenario, it is mathematically optimal for any $\ell_0$-penalized method (like MP‑SAE) to encode both with a single feature, as the absorbed solution is constant and highly compressible.
> Our variant of the toy setup preserves the same structure ($x = p + c$) but **relaxes the norm constraint**, sampling
> $$p \sim \mathcal{N}(\mu_1, \sigma_1), \quad c \sim \mathcal{N}(\mu_2, \sigma_2),$$
> so that children still appear only with parents but the magnitudes are no longer perfectly proportional. This introduces realistic variance, reflecting that in real data, children rarely co‑occur with identical parent magnitudes. We strongly believe this setup better reflects realistic hierarchical data and highlights MP‑SAE’s robustness beyond over‑simplified scenarios.
> We acknowledge that high variance with equal means ($ \mu_2=\mu_1$) can yield edge cases where a strong child and weak parent make $p+c \approx c$, making the parent appear almost absent. To address the reviewer’s point, we have now evaluated MP‑SAE over a grid of child means and variances, with the parent fixed as $p \sim \mathcal{N}(1,(1/4)^2)$. For each configuration, we computed the cosine similarity between learned features and the ground‑truth parent across 15 runs. Lower similarity indicates successful separation; high similarity or NaNs indicate absorption or failure to recover either component.
> ### MP
> |Std\Mean|0.1|0.5|1.0|
> |:--|--:|--:|--:|
> |1/400|3.6e-03|7.3e-02|1.3e-03|
> |1/40|5.0e-03|6.2e-02|1.2e-03|
> |1/8|-|4.3e-02|1.1e-03|
> |1/4|-|-|1.1e-03|
>
> ### Matryoshka
> |Std\Mean|0.1|0.5|1.0|
> |:--|--:|--:|--:|
> |1/400|nan|4.8e-02|2.1e-01|
> |1/40|nan|9.0e-02|1.5e-01|
> |1/8|-|9.6e-02|2.1e-01|
> |1/4|-|-|2.2e-01|
>
> ### Vanilla
> |Std\Mean|0.1|0.5|1.0|
> |:--|--:|--:|--:|
> |1/400|nan|7.0e-01|4.5e-01|
> |1/40|nan|7.0e-01|4.5e-01|
> |1/8|-|7.0e-01|4.7e-01|
> |1/4|-|-|4.7e-01|
>
> Overall, across the board, we see that MP-SAE reliably recovers the parent and child as distinct features, even when the child’s firing magnitude is very small and exhibits low variance. In contrast, other methods consistently fail in these regimes, either merging parent and child into a single atom or failing to recover either component (as reflected by NaNs in the results). This confirms that feature absorption observed in prior work arises from **toy benchmarks assuming perfect parent‑child correlation**!
>
> >_the authors should train MP-SAEs on something like Gemma-2-2b and run the evaluations from SAEBench [...] to make it easier to recommend MP-SAEs as the SOTA going forward_
>
> Thank you for the comment! Given this and some of the other comments, we believe there is a misunderstanding on the goal of our paper and we would hence like to clarify this first.
>
> **Goals of our work:** Please note **the goal of our work was not to develop a “SOTA” SAE.** We in fact strongly think such an SAE does not exist, and by no means do we believe MP-SAE is that universal tool (as we explicitly noted in the limitations of our work, and are further formalizing in a follow-up paper now). Instead, we note the goal of our work was to highlight limitations of the standard working hypothesis that underlies the design of SAEs, i.e., the linear representation hypothesis (LRH). We wanted to assess that when neural network representations encode concepts in a manner that is outside the scope of LRH, are SAEs, which are motivated via LRH, able to capture such concepts. To this end, we aimed at providing a formalization of (our interpretation of) LRH and how concepts exhibiting hierarchical structure are outside that scope. Thus, our main goal with proposing MP-SAE was to offer a baseline to compare standard SAEs with. With real data experiments (e.g., our large-scale vision experiments), we wanted to see if we can produce interesting findings (such as our results on multi-modal representations) when using a tool which has a different inductive bias than standardly used interpretability tools (such as linear probes, which by definition will find concepts that are solely linearly accessible),
>
> **On empirical evaluations with SAEBench:** We would like to emphasize our experiments are already fairly wide in their scope: we evaluated MP-SAE against several standard SAE architectures on **large-scale vision models, multimodal vision-language models, smaller language models, and synthetic tasks!** While we certainly agree with the reviewer that analyzing MP-SAEs trained on LLMs to enable a comparison with other SAEBench results would be interesting, we strongly emphasize such an evaluation warrants its own paper and is not the goal of our work. We are nevertheless happy to add minimal experiments for MP-SAEs trained on Gemma-2B, e.g., reporting the learning curves and feature visualizations from SAELens, in the final version of the paper.
>
> >_Furthermore, none of the benchmarks performed investigate if the latents learned are actually interpretable (e.g. using something like a sparse-probing benchmark or even auto-interp)._
>
> We emphasize our analysis of multimodal models, where we show the surprising finding of representations decomposing across different modalities (see Figure 17 in the Appendix), involved analyzing what inputs (both text and visual tokens) activate the latents. This involved producing several feature visualization maps (in fact Figure 8b in the main paper emerged out of this analysis), which we are happy to add to the appendix. **We will also upload the general tooling we developed for extracting these features to our Github!** We would have been happy to release these results now, but are unable to do so because of NeurIPS guidelines.
>
> >_...but my understanding is that this paper has been debunked._
>
> Great question! We feel “debunked” is too strong. The blog post shows that in high-dimensional spaces, random vectors can satisfy conditional orthogonality, which we agree with. **However, the reverse still holds: hierarchical factors of the data distribution satisfy this geometrical condition (see the updated paper).** Thus, seeking conditionally orthogonal directions to explain the data remains meaningful (as shown in Figs. 8b, 17)—we emphasize the phrase “explain the data” here, since random vectors that satisfy conditional orthogonality will not enable good reconstruction. More broadly, this applies to all SAEs: even vanilla SAEs rely on quasi-orthogonal directions, and while random vectors can appear orthogonal, only those that reconstruct the data provide meaningful explanations.
>
> >_This method is likely much less efficient than standard SAEs…_
>
> Fair point! We believe efficiency is generally not a crucial factor for interpretability tools, since the training cost is a one-time expenditure and if it can offer you a useful lens into the model, that cost can be worth it. That said, we emphasize that across all our experiments, **we found MP-SAE converges substantially faster than alternative architectures.** For example, we found MP-SAEs can achieve the same loss as a TopK SAE in 1/10th the training iterations (~95% of the wall clock time).
>
> >_Conceptually I don't see how the MP-SAE actually encourages learning the parent latent rather than a parent + child…_
>
> Apologies if this was unclear! In MP-SAE, residual updates naturally separate parent and child concepts. If $P$ is selected first, the residual becomes $r^1 = x - P = C$, so the next step selects the feature aligned with $C$ rather than $P+C$, MP-SAE will thus naturally learn a feature aligned with $C$ rather than the absorbed $P+C$ direction.
>
> >_…Given the dictionary is still just linear vectors, why do the authors think that MP-SAEs are capturing non-linear concepts?_
>
> Thank you for the question! We discuss this in Section 3(iii) but are happy to clarify. **The key distinction is between non-linear and non-linearly accessible concepts**, where the selection depends non-linearly on the input. In Matching Pursuit, concepts selected at $ t>1$ depend on $ D^\top r^{(t-1)} $, with $ r^{(t-1)} = x - D \cdot top_1 (D^\top x)$, which is non-linear in $ x $. By contrast, standard SAEs select all features based only on $W_{\text{enc}}x$, which is linear. Thus, MP enables non-linear selection even though the reconstruction $\hat{x} = Dz$ remains a linear combination. We will clarify this in the final paper.
>
> >_Have the authors observed any specific non-linear concepts that are recovered by MP-SAEs…?_
>
> A specific example of a non-linearly accessible concept appears in the multimodal features extracted by MP-SAE (though not recovered by standard SAEs), as shown in our main results and detailed in Appendix B.4. Figure 17 provides an illustration of how such concepts are encoded, highlighting how MP-SAE can recover multimodal features that are inaccessible to standard SAEs. We will revise the manuscript to emphasize more clearly that this is an example of non-linearly accessible concepts recovered by MP-SAEs.
>
> We hope our response addresses the reviewer’s concerns and encourages them to consider raising their score. We remain available for any further questions!

---

> > ### Comment · Reviewer_vRBm · 2025-08-01
> >
> > I want to emphasize, I think MP-SAEs are a cool idea, but **they do nothing to deal with hierarchy**. My main issue is that this paper's main claim is therefore false, and can easily be shown to be false. **If this false claim is kept in the paper, other researchers will read this, get excited, and proceed to waste large amounts of their time trying to use MP-SAEs as a solution to absorption.** The examples in the rebuttal claiming the opposite are flawed and intentionally misleading, as I will explain below.
> >
> > > In this setup, each input is defined as $x = p + c$ with a fixed proportional norm $\|p\| = \lambda \|c\|$. This structure systematically favors absorption (i.e., learning a single parent+child latent) because the signal-to-signal overlap is exact and firing magnitudes are perfectly correlated. In such a scenario, it is mathematically optimal for any $\ell_0$-penalized method (like MP‑SAE) to encode both with a single feature, as the absorbed solution is constant and highly compressible.
> >
> > **This is the entire reason that solving hierarchy is difficult!** This is **why** SAEs engage in absorption. If you claim you have an SAE architecture that solves hierarchy, your solution must handle this case. If there is not strong correlation in firing patterns between parent and child features, then solving hierarchy is trivial. Any tied SAE (setting encoder to be the inverse of the decoder) will solve this. Indeed, the only reason that MP-SAEs even appear to solve hierarchy is because the encoder and decoder are functionally tied. I expect tied TopK/BatchTopK/JumpReLU /L1 SAEs to also trivially handle hierarchy if you relax the challenge.
> >
> > > it exposes a strong and implicit assumption in the Matryoshka toy setup .. parent and child norms are perfectly correlated
> >
> > The fact that Matryoshka SAEs still solve absorption *even if* the child and parent norms are perfectly correlated is precisely *why* Matryoshka SAEs are so impressive. As you said, when the parent and child norms are highly correlated, it's extremely difficult for a naive SAE implementation to differentiate them. Being able to handle this case is not a *weakness* - this is *the most challenging version of hierarchy for SAEs*. Increasing firing variance makes the task *easier*, and Matryoshka SAEs also handle that without any issue. If MP-SAEs cannot handle correlated hierarchical features then they do not solve hierarchy.
> >
> > > ...with the parent fixed as $p \sim \mathcal{N}(1,(1/4)^2)$.
> >
> > This is *still setting a large independent firing variance for the parent feature*. This is not doing anything to address the core problem that MP-SAEs only appear to solve absorption due to large independent firing variance. If you decrease that variance then MP-SAEs fail.
> >
> > > This confirms that feature absorption observed in prior work arises from toy benchmarks assuming perfect parent‑child correlation!
> >
> > Absorption happens in **reality** due to parent-child features having highly correlated firings. It is easy to show in toy models as well that this doesn't even need to be perfect, having a small independent firing variance will still cause absorption. Creating artificial toy models where this is not the case, then claiming that you've solved absorption is dishonest. Again, any tied SAE variant will solve absorption under your assumptions. Matryoshka SAEs will still solve absorption as well.
> >
> > > we have now evaluated MP‑SAE over a grid of child means and variances
> >
> > *Every single one of these means is much larger than the mean of the parent feature*. This is designed to break Matryoshka SAEs while also not being realistic. Matryoshka SAEs rely on the realistic assumption that the parent feature has larger expected MSE than child features, and so will be learned in an earlier level of the SAE. Empirically this is a valid assumption as well, as Matryoshka SAEs do solve absorption in LLMs. You are breaking this assumption by intentionally making all child features fire with much higher magnitude than the parent. If you set $\mu_c \approx 1.0$ or lower and you'll see that Matryoshka SAEs handle this setting without issue, no matter the variance. **It is underhanded to cherry-pick unrealistic examples then claim that Matryoshka SAEs only solve absorption in "toy benchmarks" with "perfect parent-child correlation".**
> >
> > > ... a comparison with other SAEBench results ... is not the goal of our work.
> >
> > When I first saw this paper I was very excited because if it is true that MP-SAEs deal with hierarchy without the feature hedging issues the Matryoshka SAEs have this would be a major breakthrough for the field. I since trained a number of MP-SAEs on Gemma-2-2b and ran SAEBench on these SAEs myself, only to find they fail at solving absorption. This is not surprising, since, as discussed above, MP-SAEs do nothing to deal with hierarchical features. Matryoshka SAEs, on the other hand, do solve absorption, both in *hard* toy models *and* in real LLM SAEs as demonstrated by SAEBench.

---

> ### Author Response · Authors · 2025-08-02
> **Response to follow-up (1/3)**
>
> We thank the reviewer for their continued engagement. While we respond to technical comments below, we emphasize from the outset that we are highly disheartened by personal comments made on our research integrity by the reviewer: arguing that we are being “intentionally misleading”, “dishonest”, and “underhanded” is both unprofessional and a blockade towards meaningful communication. We hope the reviewer will consider engaging with us in a fair and respectful way, without presuming or projecting malicious intentions on our end. We respond to specific technical comments next.
>
> ----
> ----
>
> ### **Clarifications on the term “hierarchy” and our toy task’s generative process**
>
> Several comments made by the reviewer revolve around the generative process we used in our toy experiment, arguing (if we understand the comments correctly) it lacks hierarchical structure or obscures said structure by inducing “large independent firing variance for the parent feature”. We believe these comments may be emerging out of a misunderstanding of our toy task’s generative process and, hence, would like to first map it out in greater detail. Based upon this discussion, we respond to the reviewer’s comments and justify our disagreements with them. We note that we genuinely believe we did *not* do a great job of clarifying our generative process in the paper or our last comment, and hence any misunderstandings (if they happened) were our fault. We will make sure the final version of the paper has a clearer discussion of the generative process.
>
> Please note that to enable further precision, we will use new notations below.
>
> **(Def. 1: Hierarchy.)** We say a generative process defined over variables $p$ (parent), $c$ (child) is hierarchical if $P(p | c) = 1$, while $P(c | p) < 1$. That is, if the child is sampled, it is guaranteed the parent was sampled; however, if the parent is sampled, the child may or may not be sampled.
>
> **(Def. 2: Toy generative process.)** Consider two predefined vectors $d_p, d_c \in R^d$ that represent a “parent” vs. a “child” concept, where $d_p \perp d_c$. Let $\delta(s)$ denote an indicator variable that equals 1 depending on the truth value of a statement $s$, $B(\lambda)$ the Bernoulli distribution, and $N(\mu, \sigma)$ the Gaussian distribution. Then, observations $x$ are generated as follows: $x = p * d_p + c * d_c$, where $p \sim \mathcal{N}(1, 1/16)$ and $c \sim \mathcal{N}(\mu, \sigma) * \delta(p > 0) * B(\lambda)$.
>
> - **The toy generative process satisfies Def 1:** It is easy to see this claim is true. Specifically, an observation $x$ can have the parent concept ($d_p$) without any signature of the child, i.e, one can have $\langle x, d_p \rangle > 0$ while $\langle x, d_c \rangle = 0$. This happens when $c = 0$, which occurs with probability at least $1-\lambda$ because of the Bernoulli variable ($\lambda = 0.2$ across our experiments). However, in the reverse direction one can see that if the child concept is present in an observation, i.e., $\langle x, d_c \rangle > 0$, then the parent must be present as well, i.e., $\langle x, d_p \rangle > 0$. This happens because of the indicator variable $\delta(p > 0)$. Overall, we have: $P(p > 0 | c > 0) = 1$ and $P(c > 0 | p > 0) < 1$, hence satisfying Def. 1.
>
> -----
> ----
>
> ### **Response to specific comments**
>
> The reviewer commented our chosen hyperparameters in the generative process disincentivize feature absorption by inducing minimal correlation between parent and child (i.e., they “fire independently”) and not offering sufficient contribution of the parent to an observation. The reviewer then argued that any SAE should work well in this setup (especially if it has tied encoder–decoder weights). We disagree with these claims and clarify below why.
>
> **(Continued below.)**

---

> ### Author Response · Authors · 2025-08-02
> **Response to follow-up (2/3)**
>
> > Comment: “every single one of these means is much larger than the mean of the parent”.
>
> We note for *all* of our experiments reported in the first response, mean of the child, i.e., $\mu$, satisfies the constraint that $\mu \leq 1$. More importantly, **the average contribution of the parent is always greater in norm than that of the child in our experiments**. Specifically, define $r = \frac{E_{p}[||p * d_p||^{2}]}{E_{p, c|p}[||p * d_p + c * d_c||^{2}]}$ as the expected norm contribution of the child $d_c$ to the observation $x$. If $r > 0.5$, the parent is expected to be the dominating contributor to an observation. Assuming the parameterization of Def. 2, we can evaluate $r$ analytically as follows: $r = \frac{ \frac{17}{16} }{ \frac{17}{16} + \lambda * (\mu^{2} + \sigma^{2} )}$. To vary the contribution of $d_c$ to $x$, we experimented (reported in our first response to the reviewer) with combinations of $\mu \in {0.1, 0.5, 1.0}$ and $\sigma \in {1/400, 1/40, 1/8, 1/4}$. For these values, we get $r \in [0.83, 0.998]$ (when $\mu, \sigma$ are largest value combinations, $r=0.83$; when they are their smallest combinations, $r=0.998$). Thus, the parent is the dominating part of the signal observed in $x$ for almost all our experiments, and we see MP performing consistently well in all these combinations. On account of both these arguments, we thus disagree with the reviewer’s comment that “every single one of these means is much larger than the mean of the parent”.
>
> > Comments on “independent firing”.
>
> We note statistical dependence (“correlation”), and hence Hierarchy, is present between $p, c$ in all our experiments. The reviewer argued that we have independent firing rates for the parents vs. child concepts. Referring back to our generative process in Def. 2, we note the firing rates (i.e., $p$ and $c$) follow the constraint that $P(p > 0 | c > 0) = 1$. That is, we have $P(p > 0, c > 0) = P(p > 0)$, which **necessarily implies the firing rates are statistically dependent**. We are thus unsure of the reviewer’s argument that firing rates are independent in our generative process.
>
>
> > Comment: “If you set $\mu_c \approx 1.0$ or lower and you'll see that Matryoshka SAEs handle this setting without issue, no matter the variance.”
>
> Building on the above notes, we stress that for all our experiments $\mu_c$ (now denoted $\mu$) is in fact less than 1. As reported in our grid experiments in the previous response to reviewer, we consistently see **MP-SAE performs better or on-par with Matryoshka**. We are hence uncertain of the reviewer’s claim.
>
> > Comment: “Any tied SAE (setting encoder to be the inverse of the decoder) will solve this.”
>
> While we do not follow the logic based upon which the reviewer made this claim, we nevertheless ran the experiment and **found the claim to be not true**. Specifically, we trained a Vanilla SAE with tied weights and found its performance to be rather poor when compared to MP-SAE, as reported below.
>
> |Std\Mean|0.1|0.5|1.0|
> |:---|---:|---:|---:|
> |1/400|nan|2.8e-01|3.2e-01|
> |1/40|nan|3.0e-01|3.2e-01|
> |1/8|-|2.8e-01|3.2e-01|
> |1/4|-|-|3.2e-01|
>
> > Comment: “If you claim you have an SAE architecture that solves hierarchy, your solution must handle this case."
>
> We believe this comment mandates a very specific and synthetic notion of what it means to be a hierarchy: that $c = \alpha \cdot p$. In this case, an observation will be defined as $x = p * (d_p + \alpha \cdot B(\lambda) d_c)$. As we noted in our response before, MP-SAEs, since they are trying to solve the sparse coding problem, do not work well in such a case (0.4 average cosine similarity between extract and GT features). Matryoshka, somehow, does achieve a nontrivial performance (0.2 average cosine similarity between extract and GT features) and we do find that impressive. However, inducing even slight amounts of variance in $p$ and $c$, e.g., of the order $0.01$, leads to **MP-SAEs performing substantially better** (0.003 average cosine similarity) than Matryoshka, which in fact generally fails to extract the parent and child concepts!
>    - As a meta-point, we note we disagree with the premise that a child and parent having proportional attributes implies hierarchy. For example, consider the class of red objects (parent). Apples are one such object (child). Now, you can have objects that are relatively a lot more red (e.g., stop signs) vs. barely so. Meanwhile, you can have apples that are large, small, and satisfy several other constraints (including a mixture of colors). That is, even though there is a hierarchical structure (i.e., an ontology) in which we can organize apples, there can be sufficient variation in the individual values themselves.
>
> **(Continued below...)**

---

> ### Author Response · Authors · 2025-08-02
> **Response to follow-up (3/3)**
>
> > Comment: “I since trained a number of MP-SAEs on Gemma-2-2b and ran SAEBench on these SAEs myself, only to find they fail at solving absorption.”
>
> We sincerely appreciate the reviewer’s initiative here. We are however unsure if there is a fair way to respond to the comment: the reviewer’s experiment is an independently conducted investigation that itself has undergone no vetting by the community or the authors. Importantly, the comment, at least in its phrasing, pits us against a rather recently released method (Matryoshka SAEs) on a very specific benchmark (SAEBench), while ignoring the broad spectrum of results we have offered in the paper. For example, in our experiments with multimodal models, MP-SAEs’ ability to create highly structured dictionaries that distinguish between modalities while sharing higher-level concepts is clearly a signature of MP-SAEs avoiding feature absorption in a realistic task. We note standard SAEs fail at this behavior!
>
>    - We would also like to add that we are aware of at least two organizations that have successfully employed MP-SAEs for extracting tree-structured representations from genomic language models. In the spirit of double blind review, we are unable to offer more details, but this is also why we believe this comment is difficult for us to address: we have evidence for the success of MP-SAEs for extracting hierarchical representations, based on experiments we are unable to share, and the reviewer has evidence against this, that they are unable to share. We do not believe the discussion on this comment can hence be had in a meaningful manner, due to which we implore the reviewer to scope this discussion around the experiments we have conducted and reported in the paper / rebuttals. We are happy to answer more questions on those fronts!
>
> ----
> ----
>
> **Summary:** We are sincerely thankful to the reviewer for their continued engagement. While we were disheartened by their presumed ill-intent on our end, we hope they will find our responses and the corresponding reasoning provided to be meaningful and worthy of a thoughtful conversation. We are happy to answer any further questions!

---

> > ### Comment · Reviewer_vRBm · 2025-08-02
> >
> > I thank the authors for their continued engagement and further explanations. Just to clarify a few points:
> >
> > 1. The toy model code for this paper is copy/pasted from the Matryoshka SAES work, down to idiosyncratic hyperparameter choices like setting nonstandard Adam beta values. Your code acknowledges this as well.
> > 2. However, if you use the verbatim toy model experiment that the Matryoshka SAEs work uses, MP-SAEs do not perform well and do not handle hierarchy.
> > 3. The Matyroshka SAE toy model code was thus modified so that firing variance of features in the toy model is much higher (0.25) than the original Matryoshka SAEs work (0 or 0.05) to make the parent and child features more uncorrelated.
> > 4. Only then do MP-SAEs work.
> > 5. This modified toy model experiment is the justification provided in the paper for why MP-SAEs handle hierarchy, and to justify claims that MP-SAEs work better than Matryoshka SAEs.
> >
> > If any of the above statements are not correct please let me know. Fundamentally, I feel that taking the Matryoshka SAEs hierarchical task verbatim and then diluting it until MP-SAEs pass that task is not fair.
> >
> > Furthermore, given the above, I find claims like the following dubious:
> >
> > > We came across Matryoshka SAEs relatively late in the project (their paper was released less than two months before the NeurIPS deadline).
> >
> > The Matryoshka SAEs code was published in November 2024, and you copy/pasted their code for your experiments. It is thus not credible that the authors were unaware of Matryoshka SAEs or the version of hierarchy defined in the Matryoshka SAEs toy model.
> >
> > ---
> >
> > > Building on the above notes, we stress that for all our experiments $\mu_c$ (now denoted $\mu$) is in fact less than 1. As reported in our grid experiments in the previous response to reviewer, we consistently see MP-SAE performs better or on-par with Matryoshka. We are hence uncertain of the reviewer’s claim.
> >
> > It sounds like I misunderstood what the tables you shared were indicating. If that is the case I deeply apologize for my claim that it was intentionally misleading by breaking the underlying assumptions of Matryoshka SAEs. I interpreted the column title "std/mean" with values like "1/400" as saying "std = 1, mean = 400". It sounds like I was confused about what your experiments are saying, and I apologize for not looking closer. If the values along the top of the table are means, then your experiment seems valid and I'm surprised that Matryoshka SAEs (and especially tied SAEs) do not seem to handle this. If I have time, I'll try to see if I can reproduce what you're doing here since this is surprising to me. Still, I feel setting parent variance to 0.25 does not address the core issue around MP-SAEs requires features with high independent firing variance to work.
> >
> > ---
> >
> > > We would also like to add that we are aware of at least two organizations that have successfully employed MP-SAEs for extracting tree-structured representations from genomic language models. In the spirit of double blind review, we are unable to offer more details, but this is also why we believe this comment is difficult for us to address: we have evidence for the success of MP-SAEs for extracting hierarchical representations, based on experiments we are unable to share, and the reviewer has evidence against this, that they are unable to share.
> >
> > I am also unsure how to interpret this. Maybe hierarchical features from genome LLMs do not have correlated firings to the same degree that LLMs trained on natural language do?

---

> ### Author Response · Authors · 2025-08-02
> **New response (1/3)**
>
> We again thank the reviewer for their continued engagement. We are glad to see misunderstandings on our toy task’s generative process and our previously reported experiments are beginning to clear up!
>
> ### **Summarizing points that we believe have been addressed**
>
> For completeness, before we respond to the reviewer’s latest batch of comments, we would like to summarize the precise points that we hope have been addressed.
>
> - **Reviewer comment: “that MP-SAEs solve hierarchy, is not true”.** As clarified in Defs. 1, 2 in our last response, our toy setup satisfies a reasonable definition of what it means for a generative process to be hierarchical and MP-SAEs perform very well on them (better than all other SAEs we compared to). That is, *MP-SAEs indeed disentangle concepts structured in a parent-child relationship*. We showed experiments with realistic settings to further back this claim, e.g., via analysis of multimodal models.
>
> - **Reviewer comment: our setup intentionally uses a large mean to break assumptions of Matryoshka SAEs.** As clarified in the previous response, this is *not true*. All mean and variance values are well below the upper limits the reviewer argued to be reasonable, and we showed analytically the average norm contribution of a child to an observation is anywhere between 0.2–17%, depending on the experimental setting (see Response to original rebuttals for hyperparameters).
>
> - **Reviewer comment: “Any tied SAE (setting encoder to be the inverse of the decoder) will solve this.”** As we reported previously, *this claim does not seem to hold* in any of our experiments.
>
> - **Reviewer comment: “you'll see that Matryoshka SAEs handle this setting without issue, no matter the variance.”** Building on our point above, we note we were unable to get Matryoshka SAEs to work well in several settings the reviewer reported they could work. Given that MP-SAEs work very well in most cases we experimented with, at the very least, this suggests that the two methods may have complementary benefits, if not supersede each other.
>
> We believe our last point above is where a lot of the contention may be emerging from: there are possible implicit assumptions that MP-SAEs and Matryoshka have made due to which they work well in some specific setups vs. not so much in others. For example, we have found MP-SAEs to be very meaningful with vision and vision-language models (we also emphasize again that the bulk of our experiments were focused on these domains since they were our main interest in the project), but perhaps Matryoshka has inductive biases more meaningful for language. We are happy to add a discussion in the paper to emphasize that this point is possible, hopefully offering the reader some nuance, and prompting subsequent work comparing the two SAEs.
>
> **(Continued below...)**

---

> ### Author Response · Authors · 2025-08-02
> **New response (2/3)**
>
> ### **Response to new comments**
>
> Having summarized points that we hope help clarify some of the past questions / comments, we now respond to the next batch of comments.
>
> > If any of the above statements are not correct please let me know.
>
> We believe the reviewer’s latest comments primarily revolve around how much we relied on or were inspired by Matryoshka’s codebase in the design of MP-SAEs. We hence first clarify the true order of events in our development of MP-SAEs, and then contextualize other comments with respect to that.
>
> **Order of events in the development of MP-SAEs, plus when we started experimenting with Matryoshka:** We note that the list of events offered by the reviewer is **incorrect**. We started working on MP-SAEs back in late January, primarily focusing on **raw MNIST images** as the domain of study (MNIST is still a commonly used benchmark in standard dictionary learning literature). We then moved to working with two more toy domains, including a **hierarchical mixture of sinusoids** task inspired by the Mercedes frame (again, a common domain in dictionary learning literature) and a primitive version of our mixture of Gaussians generative process that relates to our final toy experiment reported in the paper. Most of this work happened in March, concluding towards end-of-April with our evaluations on **large-scale Vision**, **large-scale multimodal models**, and **transformers trained on Tinystories** (i.e., small  language models), which showed our observed phenomenology from toy experiments transferred to realistic domains.
>
> Then, **on April 20th**, one of the co-authors, who was not involved in the project until now, suggested that if we were to write a paper on SAEs and hierarchical concepts, we would certainly be asked to experiment with Matryoshka. Since NeurIPS deadline (15th May) was fast approaching, we hence started focusing on Matryoshka’s toy experiments codebase on **April 26th**: given that this was late in the game, we focused on reappropriating their code instead of writing something from scratch; we acknowledge this multiple times in the paper and our anonymous github repo.
>
> In our first attempt, we did fail to get MP-SAEs to work off-the-shelf with the toy setup, which was **surprising since we had at this point significant evidence towards the success of MP-SAEs on real plus toy data experiments**. We then realized the assumption of perfect correlation between parent and child is causing this challenge: MP-SAEs start to work very well if we increase the variance to 0.0625 (we emphasize the reviewer likely misinterpreted our comments before: **we set variance to $0.25^2 = 0.0625$**; the standard deviation is set to $0.25$). This trend continued to hold in several experiments we reported in the grids reported in our last comment: nonzero, but very low, variance suffices to get MP-SAEs to work. Given that we did not find the perfect correlation assumption in Matryoshka’s toy codebase to be well-justified, we ended up reporting our experiments with an increased variance setting. This seemed perfectly reasonable to us, and since we cited Matryoshka multiple times, including in our appendix and anonymous github repo, we believed we have conducted everything in a fair spirit. We also ended up not including other toy experiments (MNIST / sinuosids) because of space constraints (though we are willing to note a workshop version of our work already has them, and we are also happy to include these experiments in the final version of the paper). Furthermore, Matryoshka-based toy experiments, given that they had precedence in the community, felt like the right toy experiment to include.
>
> **To summarize:** We did not try to intentionally break Matryoshka’s codebase to make MP-SAEs look better---we had solid results across at least 6 domains of different levels of realism (MNIST, mixture of Sinusoids, mixture of Gaussians, large-scale vision models, large-scale multimodal models, and toy language models based on Tinystories). We noticed an unreasonable assumption in Matryoshka’s toy experiment, changed it, and found Matryoshka to be quite brittle beyond it. We accordingly reported these experiments, but only with the assumption that the new setup is reasonable. Since then, in our responses to reviewer comments we have sweeped the hyperparameters from the code (refer to result grids in first rebuttals response), finding our claims to work well for both large and fairly low mean / variance settings (though not for ones with zero variance).
>
> **(Continued below...)**

---

> ### Author Response · Authors · 2025-08-02
> **New response (3/3)**
>
> Having offered our order of events in the design of MP-SAE and rebutted reviewer's version thereof, we now respond to critical comments we found in the reviewer's latest response.
>
> > The Matyroshka SAE toy model code was thus modified so that firing variance of features in the toy model is much higher (0.25) than the original Matryoshka SAEs work (0 or 0.05) to make the parent and child features more uncorrelated.
>
> We wanted to respond to this comment again in order to ensure the response is highlighted: the **variance in our experiments is $0.0625$**. The reviewer’s quoted number is likely the standard deviation, which indeed equals $0.25$.
>
> > Fundamentally, I feel that taking the Matryoshka SAEs hierarchical task verbatim and then diluting it until MP-SAEs pass that task is not fair.
>
> We hope the reviewer can understand a set of malicious intention actions to “dilute” the setup “until MP-SAEs pass” was never performed: we had significant evidence for MP-SAEs’ success. **We only changed the setup in Matryoshka because its assumption of perfect correlation did not make sense to us**. We are happy to offer a detailed note about this in the paper, but again sincerely request the reviewer to not assume malice on our end or comment in accordance with that.
>
> > The Matryoshka SAEs code was published in November 2024, and you copy/pasted their code for your experiments. It is thus not credible that the authors were unaware of Matryoshka SAEs or the version of hierarchy defined in the Matryoshka SAEs toy model.
>
> We again refer the reviewer to our discussed order of events in the comments above. **We did not read the Matryoshka paper until at least April 20th, and did not experiment with it until April 26th.** Final experiments, as reported, were performed between May 10–15. Thus, while we did borrow the final structure of the hierarchical task from Matryoshka SAEs, it was very late into the project. Even our co-author who made us aware of Matryoshka SAEs had not heard of the paper until at least late February, and definitely had not experimented with it until April. We also note the standards of deeming a “codebase” as contemporary work seem unreasonable to us: it is not justified to expect that a codebase would proliferate across the community, let alone to other sister fields (to which most of the co-authors belong; we will avoid saying more in the interest of double blind reviewing).
>
> > I am also unsure how to interpret this. Maybe hierarchical features from genome LLMs do not have correlated firings to the same degree that LLMs trained on natural language do?
>
> We would prefer to not speculate much without analyzing both experiments sufficiently. We note hierarchical structures are extremely common in biological settings (e.g., phylogenetic trees) and have fairly high correlations between parent and child nodes. At the very least, these experiments are suggestive of MP-SAEs’ utility in a practical scenario.
>
> -----
> -----
>
> **Summary:** We hope the discussion above clarifies the order of events that led up to our work. If required, we’d be happy to offer precise documentation of dates. However, we do want to note that at this point the discussion is losing the focus on science and methods offered by our work, and circling around **assumed** intentions that led to our results. Nevertheless, please let us know if you have any further questions and we'd be happy to respond!

---

> > ### Comment · Reviewer_vRBm · 2025-08-02
> >
> > Thank you again for your continued engagement on these questions. Just to follow up on an earlier point:
> >
> > > we trained a Vanilla SAE with tied weights and found its performance to be rather poor when compared to MP-SAE
> >
> > I investigated further whether tied SAEs will similarly pass your benchmark, and I find they do indeed achieve perfect performance, matching MP-SAEs on your toy model. This indicates that the only reason MP-SAEs appear to handle hierarchy at all is, in-fact, due their being functionally tied, and not due to anything related to the MP-SAE architecture. My code is below:
> >
> > ```python
> > class TiedBatchTopKSAE(BatchTopKSAE):
> >     def __init__(self, d_model, n_latents, n_steps, target_l0, lr=3e-2):
> >         nn.Module.__init__(self)
> >
> >         self.d_model = d_model
> >         self.n_latents = n_latents
> >         self.target_l0 = target_l0
> >         self.threshold_momentum = 0.9
> >
> >         self.W_dec = nn.Parameter(0.25 * torch.randn(d_model, n_latents))
> >         self.b_dec = nn.Parameter(torch.zeros(d_model))
> >
> >         self.optimizer = torch.optim.Adam(self.parameters(), lr=lr, betas=(0.5, 0.9375))
> >         self.scaler = torch.amp.GradScaler("cuda")
> >         self.scheduler = get_wsd_scheduler(
> >             self.optimizer,
> >             n_steps=n_steps,
> >             n_warmup_steps=100,
> >             percent_cooldown=0.2,
> >             end_lr_factor=0.1,
> >         )
> >         self.register_buffer(
> >             "threshold", torch.tensor(float("nan"), device=self.device)
> >         )
> >
> >     @property
> >     def b_enc(self):
> >         return self.b_dec.new_zeros(self.n_latents, requires_grad=False)
> >
> >     @property
> >     def W_enc(self):
> >         return self.W_dec.T
> >
> >     def get_acts(self, x):
> >         return super().get_acts(x - self.b_dec)
> >
> >     def step(self, x, return_metrics=False):
> >         return super().step(x - self.b_dec, return_metrics)
> > ```
> >
> > This code is just an extension of your `BatchTopKSAE` code, but set up to subtract the decoder bias on all inputs, and tie the encoder and decoder together.
> >
> > If you insert this code in the bottom of your `sae.py` file, and replace `topk_sae = BatchTopKSAE(...)` with `topk_sae = TiedBatchTopKSAE(...)` in your toy model experiment notebook, you will see the result is perfect performance on your toy model, matching the MP-SAE. I cannot share the resulting plot here unfortunately, but if you run this code you will see it is true.
> >
> > ## Why tied SAEs solve hierarchy with high variance
> >
> > Tied SAEs cannot engage in absorption in the same way as untied SAEs, since they cannot change the encoder to not fire the parent latent when the child is active. They cannot learn the absorption solution (f1 = C + P, f2 = P), because f1 cannot account for this high variance. Thus the SAE must learn f1 = C, f2 = P, the correct solution.
> >
> > However, if there's low variance, then the tied SAE can easily learn the absorption solution f1 = C + P, f2 = P. When a child latent fires, then f1 is able to perfectly reconstruct it, and (assuming it's a TopK or JumpReLU SAE), the activation function keeps f2 from firing at all. This is what happens with MP-SAEs as well - if there's low variance, then the MP-SAE will first pick f1=C + P and perfectly represent both the parent and child latents together, and still get perfect reconstruction.

---

> > ### Comment · Reviewer_vRBm · 2025-08-02
> >
> > Just to summarize:
> >
> > My position is that MP-SAEs are a cool idea and an interesting addition to the field.
> >
> > **My problem is with the claims this paper makes around MP-SAEs ability to handle hierarchical features**. The paper goes even farther, and claims that MP-SAEs outperform Matryoshka SAEs at handling hierarchical features. However, this is problematic for the following 4 reasons:
> >
> > 1.  This paper uses the evaluation code from Matryoshka SAEs, but changes the evaluation criteria that were used to evaluate Matryoshka SAEs by increasing the feature firing variance.
> > 2.  If this change was not made, MP-SAEs fail to handle hierarchical features outright. That MP-SAEs fail to handle hierarchical features without large independent firing variance is never mentioned in the paper.
> > 3.  The fact that the evaluation criteria from Matryoshka SAEs is being changed is never mentioned in the paper, despite claiming to outperform Matryoshka SAEs.
> > 4. This change in the evaluation criteria is not justified in the paper.
> >
> > This would be like a paper claiming to get a SOTA score on MMLU, but when you dig into their code, it turns out they modified MMLU in ways that make their model score higher. I will take the authors at their word that this was not done with dishonest intent and is just an oversight.
> >
> > I further believe that the only reason that MP-SAEs appear to solve hierarchy is simply because it is functionally a tied SAE. As I demonstrated in my previous comment, tied SAEs have no problem with the hierarchical features toy model used in this paper due to the high independent firing variance between parent and child.
> >
> > As I said in my initial review, **if the claim that MP-SAEs solve hierarchy is dropped, I will raise my score.**  At the very least, the points above 4 points should be explicitly and prominently talked about in the paper.
> >
> > Otherwise, researchers will read this paper, think that MP-SAEs solve hierarchical features in general, and proceed to waste large amounts of their time trying to use MP-SAEs as a solution to absorption. This is not fair to other researchers.
> >
> > The limitations in MP-SAEs ability to handle correlated hierarchical features may be fine in models for genomics or vision - I don't have experience with training SAEs for those fields - but this will definitely not handle hierarchical features in LLMs.
> >
> > Again, I think MP-SAEs are a nice addition to the field, I just wish the authors would not also claim that MP-SAEs solve hierarchy. Or if they want to keep that claim, then at least add a serious discussion explaining that the evaluation used is different from what Matryoshka SAEs use and why, and that *MP-SAEs fail to handle hierarchical features if the features have correlated firing magnitudes*.

---

> ### Author Response · Authors · 2025-08-03
> **Response to Comments on Tied-Weights SAEs (1/2)**
>
> We again thank the reviewer for their continued initiative. Before we respond to their comment on BatchTopK with tied weights, we would like to clarify the chronology of events here.
>
> ### **Chronology of discussion around tied-weights SAEs:**
>
> - **Reviewer states their claim:** The reviewer claimed that any SAE with *tied weights* should perform well in our setup because of the ‘large’ variance; more specifically, the reviewer stated “I expect tied TopK/BatchTopK/JumpReLU /L1 SAEs to also trivially handle hierarchy if you relax the challenge”.
>   - To be quantitatively precise, we note that the parent activation variance was $0.0625$, with a mean of $1.0$ in our experiments---we hence disagree the variance is “large”. Child mean and variance were chosen such that the child’s norm contribution to an observation is bounded between $0.2–17$%, which we also disagree is large.
>
> - **We performed tied-weights ReLU SAE experiment:** To evaluate the validity of reviewer’s claim, we ran experiments with tied-weights vanilla (i.e., L1) SAEs. *We found the reviewer’s claim does not hold true*: vanilla SAEs with tied weights have much worse disentanglement of parent and child concepts.
>
> - **Reviewer updated their statement to BatchTopK and JumpReLU:** The reviewer responded with a BatchTopK variant that they claimed works well, i.e., achieves similar performance as MP-SAEs. Reviewer also updated their earlier claim with a qualifier phrase stating that their claim assumes BatchTopK / JumpReLU SAEs are used: more specifically, the new statement by the reviewer is “When a child latent fires, then f1 is able to perfectly reconstruct it, and (*assuming it's a TopK or JumpReLU SAE*), the activation function keeps f2 from firing at all”.
>
> - **Reviewer conclusion:** Based on point 3, the reviewer claimed MP-SAEs are only able to disentangle parent and child concepts *because* of their weight-tied structure.
>
> As we show next, **we strongly disagree with the reviewer’s conclusion**, and to an extent with point 3 itself. More broadly, we want to emphasize to the reviewer that we are unsure if performing one experiment can suffice to outrightly reject or accept a hypothesis, **especially when contrary evidence has been offered**. That is, outrightly claiming that MP-SAEs work “because of tied-weights” because of a singular experiment the reviewer conducted, when we provided experiments of similar nature that do not support the claim (ones with tied-ReLU SAEs), makes the reviewer argument insufficient in our opinion. We also note that the reviewer’s subsequent updating of their statement in response to our experiments countering their claim, i.e., adding the qualifier “assuming BatchTopK / JumpReLU” to their claim, does not invoke confidence on our end. Nevertheless, we provide our replication of BatchTopK experiments next.
>
> ---------
>
> **(Continued below...)**

---

> ### Author Response · Authors · 2025-08-03
> **Response to Comments on Tied-Weights SAEs (2/2)**
>
> ### **Replicating the BatchTopK experiment**
>
> We used the reviewer’s offered code to assess whether BatchTopK with tied weights (called tied-BatchTopK hereafter) yields similar performance to MP, as claimed by the reviewer. Since reviewer suggested this should especially happen in the “most generous setting” that we used in our submitted version (i.e., Figure 4), we reconducted that experiment to compare tied-BatchTopK and MP-SAEs. In line with those experiments, we again report “Flat MSE” vs. “Hierarchical MSE”, i.e., error at capturing ground truth similarity of concepts within a given level vs. across different levels of the hierarchy. We also sweep correlation between concepts at the same level of the hierarchy, which, originally, was our attempt in the paper to intentionally make MP-SAEs fail. Results are as follows.
>
> **Flat MSE**
> |Correlation|TiedBatchTopK|MP|
> |----|------:|------:|
> |0.0|1.0e-04|2.3e-07|
> |0.1|1.3e-03|4.9e-07|
> |0.2|6.2e-03|3.4e-07|
> |0.3|1.9e-02|3.3e-07|
> |0.4|5.7e-02|5.5e-07|
> |0.5|1.4e-01|8.7e-03|
> |0.6|3.0e-01|5.5e-02|
> |0.7|4.5e-01|2.6e-01|
> |0.8|6.1e-01|2.7e-01|
> |0.9|8.6e-01|3.4e-01|
>
> **Hierarchical MSE**
> |Correlation|TiedBatchTopK|MP|
> |----|------:|------:|
> |0.0|1.2e-05|6.4e-07|
> |0.1|1.8e-05|6.2e-07|
> |0.2|6.8e-05|6.3e-07|
> |0.3|1.8e-04|7.3e-07|
> |0.4|3.7e-04|1.4e-06|
> |0.5|2.0e-03|3.9e-03|
> |0.6|5.7e-03|6.2e-03|
> |0.7|3.3e-03|1.0e-02|
> |0.8|1.0e-02|3.4e-03|
> |0.9|3.5e-02|8.5e-06|
>
>
> While we indeed see that, *to an extent*, tied-BatchTopK is able to disentangle the parent and child concepts, we find its error (both flat and hierarchical MSE) is *orders of magnitude worse* than MP-SAEs’. When visualizing the correlation matrices, we do see the *hierarchical structure is reflected in the concepts extracted by tied-BatchTopK, but there is very clear and noticeable interference* between concepts both within a level and across them. **We will happily include these results in the final version of the paper.**
>
> **Additional note on tied-weights SAEs:** More broadly, we note tied-weights SAEs are generally not used in literature because their limited expressivity constrains their reconstruction abilities. For example, prompted by reviewer comment, we trained tied-weights BatchTopK SAEs on TinyStories and found their normalized MSE to be twice that of MP-SAEs. Moreover, their learned dictionaries are less structured (Babel score of ~30, compared to ~100 of MP-SAEs). In this sense, one can also see MP-SAEs as an interesting protocol for scaling up tied-weights SAEs!
>
> ---------
>
> ### **Our Conclusion on Tied-Weights SAEs**
>
> We have put in an honest effort to assess reviewer’s claim that any tied-weight SAE should perform equivalent to MP-SAEs in our experimental setup. **As we understand the claim, it is based on an intuitive argument—not a formal proof.** Hence, given our empirical evidence against the claim, we are unwilling to concede to it. That is, **we disagree** that tying the encoder–decoder weights is the reason MP-SAEs are able to disentangle hierarchical concepts in our toy experiments (regardless of variance). Our experiments with vanilla SAEs back our claim, and our tied-BatchTopK experiments add more nuance to the statement, but do not suffice to support the reviewer’s conclusion. Moreover, our large-scale experiments in vision showcase MP-SAEs’ expressivity, and toy experiments with TinyStories we conducted in response to the reviewer’s comment do as well.
>
> Overall, **we are very happy to include a discussion of these results in the paper,** clarifying that we tried a hypothesis (i.e., tied-weights structure) different to our core hypothesis in the paper (i.e., MP-SAEs’ ability to learn conditionally orthogonal dictionaries) to explain why MP-SAEs perform well at extracting hierarchical concepts. We hope this helps.

---

> ### Author Response · Authors · 2025-08-03
> **Concluding remarks (1/2)**
>
> We again thank the reviewer for their engagement and initiative. We believe the discussion has brought up several interesting ideas, a subset of which we agree with and a subset that we do not. As we have promised throughout our responses, we will be updating the paper to reflect these points. However, to be precise, we take this opportunity to qualify how we will edit the paper to address the reviewer's comments. We also note that we do not intend to respond in a quoted statement form for the reviewer’s last comment: there are several points there we disagree with (e.g., that “researchers reading this paper will waste large amounts of time”), but believe these disagreements primarily emerge from differences in definitions and different empirical priors based upon domains of experience. We hence do not believe the conversation can move further unless agreement is achieved on these fundamentals, so we will make this our last response in the current stage of discussion. We believe we have addressed all comments raised by the reviewer, and, regardless of whether they increase their score or whether the paper is accepted / rejected, we are happy with our stances and planned paper edits (as stated below) in response to reviewer’s comments.
>
> - **Definition of hierarchy and what it means to “solve” it.** We believe an implicit, but crucial point of contention has been what it means to “solve hierarchy”. Our definition, as formalized in our responses (i.e., $P(p|c) = 1$, but $P(c|p) < 1$), is met by our broad spectrum of experimental settings. MP-SAEs perform well in these settings (while also outperforming other SAEs). However, in the extreme, this definition corresponds to $P(p|c) = 1$ and $P(c|p) = 1$. We believe this latter case maps on to the reviewer's definition of what “solving hierarchy” means: that is, if one claims their SAE is able to disentangle hierarchical concepts, it should work well for this condition. We happily concede MP-SAEs perform poorly in this case: under perfect correlation, MP-SAEs do not work well. However, we disagree with the strong version of the reviewer's claim, i.e., that MP-SAEs only work under large variance (i.e., uncorrelated parent-child firing). Our experiments contain several large correlation settings in which MP-SAEs perform well, just not the perfect correlation ones. We also do not believe the perfect correlation setting is particularly reasonable, but this is arguably just a reflection of the domain priors in which we vs. the reviewer find ourselves more experienced in.
>    - **Planned edits:** We will formally state our definition of what it means for a generative process to be hierarchical, and we will state that we do *not* believe the scenario of $P(p|c) = P(c|p) = 1$, i.e., scenario of perfect correlation, is particularly reasonable. We will qualify the statement with the note that this latter case is what recent contemporary work has sought to address, highlighting again Matryoshka SAEs and their ability to handle this case. Crucially, we will add a note that it is possible that MP-SAEs and Matryoshka SAEs are complementary tools for handling scenarios with hierarchical concepts. We believe we have offered significant formal and empirical evidence to this end, and will add the new experiments conducted during discussion with the reviewer to the final version of the paper: in these experiments, we observed under non-zero, but nevertheless large, correlation, MP-SAEs succeed and often outperform Matryoshka;  however, when the correlation approaches 1, we will empirically show and explicitly emphasize that there are scenarios where Matryoshka outperforms MP-SAEs.
>
> - **Discussion of alternative hypotheses to explain MP-SAEs’ success.** The reviewer brought up the hypothesis that tied-weights structure, i.e., same encoder–decoder weights, leads to the success of MP-SAEs. Based on our significant empirical evidence, we are currently unable to accept this hypothesis, but do find the improved results for tied-BatchTopK to be interesting. We will thus make the following edits to the paper.
>   - **Planned edits:** We will add experiments conducted in response to the reviewer’s comments on tied-weights SAEs to the final version of the paper, clarifying that tied-ReLU SAEs do not show much signs of disentanglement between parent and child concepts, but tied-BatchTopK SAEs do. Both, however, severely underperform MP-SAEs. We will provide precise quantitative results (as reported before in our response), and their corresponding visualizations to support our claims.
>
> **(Continued below...)**

---

> ### Author Response · Authors · 2025-08-03
> **Concluding remarks (2/2)**
>
> - **Discussion and experiments ablating firing variances.** This point relates to the first one, but just to emphasize again, we happily concede the experiment reported in Figure 4 used unnecessarily large parent firing variance, inducing reduced correlation between parent concept and observations: specifically, the norm contribution of the parent to an observation was $\sim 83$%. We have since run experiments, as reported in our responses, to increase this contribution to $\sim 99.8$%, yielding a setting where observations and parents are highly correlated. In this case, MP-SAEs continue to outperform other methods. We will edit the paper as follows to report these results.
>    - **Planned edits:** We will add the full spectrum of results for different mean / variance hyperparameters, as reported in our responses, to the appendix. We will also replace Figure 4 with the case where the parent contribution to an observation is $99.8$%, allowing a reader further confidence in our claims. We will nevertheless report inter-concept similarity visualizations similar to Figure 4 for all other settings we have experimented with in the appendix.
>
>
> - **Addition of other domains.** The reviewer expressed concern that we may have intentionally changed the code for toy experiments from paper on Matryoshka SAEs to ensure MP-SAEs outperform Matryoshka. Despite the reviewer’s insistence on this (and aggressive, disrespectful comments that called us “dishonest”, “intentionally misleading”, and “underhanded”), as we have stated multiple times, this was not the case: we had a multitude of results demonstrating the ability of MP-SAEs for extracting hierarchical concepts, including in alternative toy domains (MNIST and mixture of Sinuoids inspired by the concept of Mercedes Frame from dictionary learning) and realistic domains (large-scale vision and multimodal models, plus small scale TinyStories models). We only started experimenting with the toy Matryoshka experiments after the above analyses were done, and only changed its generative process because of its, in our opinion, unreasonable assumption of perfect correlation between parent and child where we expect any solution to the sparse coding to necessarily fail. Nevertheless, we will edit the paper as follows to clarify our stance.
>   - **Planned edits:** We will add experiments on further toy domains (MNIST and mixture of Sinusoids) that were conducted before we analyzed the toy domain borrowed from paper on Matryoshka SAEs. Though we already note at several points that we borrowed a toy domain from Matryoshka, we will emphasize this a lot more strongly in both the paper and the github repo. We will clarify with details on why we modified the generative process of Matryoshka, and as mentioned before, show that MP-SAEs fail to disentangle parent and child concepts if Matryoshka’s perfect correlation assumption is preserved. We will emphasize that Matryoshka succeeds in this case.
>
> ------
> ------
>
> **Summary:** We again thank the reviewer for their feedback. We will be including several clarifications on points the reviewer has very thoroughly engaged with us, as noted above. While our current plan is to focus on merely the above points, if the reviewer found any of our other responses to be interesting, we request they let us know and we would be happy to accommodate them in the paper! We do however emphasize that we do not intend to change our narrative: we believe our claims are true, despite the reviewer claiming otherwise, and we intend to stick to that.

---

> > ### Comment · Reviewer_vRBm · 2025-08-03
> >
> > Thank you for the continued engagement with this thread. I want to clarify a few statements that show there is a misunderstanding about my claims.
> >
> > **Definition of hierarchy and what it means to “solve” it.**
> > We have the same definition of "hierarchy" here. I also define hierarchy as $P(c|p) = 1, P(p|c) < 1$, the same as the authors. I never claimed anything around $P(c|p) = 1, P(p|c) = 1$, I don't know where that misunderstanding came from.  **My disagreement is that the authors add an additional constraint, that when c fires, its magnitude must be highly uncorrelated with p**. For example, according to the authors, when c fires, if $|c| = \lambda |p|$  where $\lambda$ is some constant, then this does not count as hierarchy. This constraint is not present in any other work I know of dealing with hierarchy, and was not present in the Matryoshka SAEs code where the authors took their benchmark from. If this constraint is not added, the MP-SAEs no longer can be said to solve hierarchy.
> >
> > **Tied SAEs**
> >
> > > We used the reviewer’s offered code to assess whether BatchTopK with tied weights (called tied-BatchTopK hereafter) yields similar performance to MP
> >
> > My explicit, testable claim here is that when I run the code for the tied BatchTopK SAE I provided in the authors own experiment used to generate Figure 4a in the paper, it gives an identical result to the MP-SAE result in Figure 4a. This plot is used in the paper to justify the claim that MP-SAEs solve hierarchy. If that plot is sufficient to claim the MP-SAEs handle hierarchy under the author's definition, then it should be sufficient to claim that the tied SAE I provided also handles hierarchy. Do the authors agree or disagree that running the tied SAE code I provided in their experiment generates identical results to the decoder cosine sim plot in Figure 4a in the paper?
> >
> > I also cannot validate the claims the authors make in the tables provided as I have not seen any code for these experiments. The code the authors provided only generate Figure 4a in their paper, and the tied SAE code I provided generates identical plots. Can the authors share the code they are running to create these tables?

---

> > > ### Comment · Reviewer_vRBm · 2025-08-03
> > >
> > > > Planned edits: We will add experiments on further toy domains (MNIST and mixture of Sinusoids) that were conducted before we analyzed the toy domain borrowed from paper on Matryoshka SAEs. Though we already note at several points that we borrowed a toy domain from Matryoshka, we will emphasize this a lot more strongly in both the paper and the github repo. We will clarify with details on why we modified the generative process of Matryoshka, and as mentioned before, show that MP-SAEs fail to disentangle parent and child concepts if Matryoshka’s perfect correlation assumption is preserved. We will emphasize that Matryoshka succeeds in this case.
> > >
> > > If the authors acknowledge the points above, I am happy to raise my score. Specifically this entails:
> > >
> > > - That the toy model from Matryoshka SAEs has been modified
> > > - justification for this modification
> > > - and that MP-SAEs fail to work in the original, unmodified version of the Matryoshka SAEs benchmark.
> > >
> > > This means acknowledging that MP-SAEs cannot deal with hierarchy if parent and child have correlated firing magnitudes *when the child fires*.
> > >
> > > Just to check - by "Matryoshka’s perfect correlation assumption" the authors mean that Matryoshka SAEs can still  disentangle hierarchical features even if there is no variance in firing magnitude, as in $|c| = \lambda |p|$, **not** $P(c|p) = 1, P(p|c) = 1$, correct? This means explicitly noting that MP-SAEs fail to work if the firing magnitudes of parent and child features are correlated **when the child fires**. It does NOT mean that the parent can only fire if the child fires. I agree that differentiating p and c when $P(c|p) = 1, P(p|c) = 1$ is impossible and would be silly to note in the paper.

---

> > > > ### Author Response · Authors · 2025-08-03
> > > > **Response**
> > > >
> > > > > “[...] according to the authors, when c fires, if $|c| = \lambda |p|$ where $\lambda$ is some constant, then this does not count as hierarchy [...]”
> > > >
> > > > It seems we did get confused in the very last batch of comments by the reviewer, believing the reviewer was suggesting P(c|p) = 1; we apologize for the confusion and are glad to see the reviewer agrees with this scenario’s triviality. We respond to the specific experimental setting reviewer sought for us to address, i.e., $|c| = \lambda |p|$, next.
> > > >
> > > > **On $|c| = \lambda |p|$:** We believe this is a very strong assumption, and we do not see why it should be related to hierarchy (see our Def. 1). Our formulation does not impose any constraint on firing magnitude. The only requirement for hierarchy is that a child can be active only if its parent is active. For example, the feature *apple* can fire only when the feature *red object* fires (assuming all apples are red). *However, considering that the parent’s firing magnitude reflects the color intensity and the child’s firing magnitude reflects the apple’s size, we see no reason why these two magnitudes should be perfectly correlated.* Importantly, if we assumed that larger apples are also redder, we may even question whether it makes sense to disentangle the color feature (p) from the apple feature (c), as they are perfectly correlated and could simply be represented by a single feature (p + c). We understand this is exactly what feature absorption is, but we don’t see any logical reason why one should deem the child to be a “child” in the sense of a hierarchical definition. The child’s imperfect firing correlation with the parent is what provides any semblance of it being a child.
> > > >
> > > >
> > > > >”[...] Do the authors agree or disagree that running the tied SAE code I provided in their experiment generates identical results to the decoder cosine sim plot in Figure 4a in the paper?”
> > > >
> > > > We cannot provide a definitive answer to the reviewer’s question, as the results depend on both the initialization (multiple runs generally yield different results because sparse recovery always depends on the quality of the initial dictionary) and the level of cosine similarity (correlation). This is why, for different levels of correlation, we reported the median over 10 runs (different initializations) in our tables above, which we assure the reviewer were computed using the code provided by the reviewer. For zero cosine similarity, we found TiedBatchTopK performs better than Matryoshka SAEs and the other SAEs we tested—this includes a similar (though still slightly worse) visualization diagram compared to MP-SAEs; as we stated in our response before, we do acknowledge TiedBatchTopK’s success to this extent. Meanwhile, MP-SAE still achieves the best performance on both our hierarchical and flat MSE. As correlation increases, TiedBatchTopK is no longer able to recover the intra-level correlation.
> > > >
> > > > [1] Arora et al. “Simple, Efficient, and Neural Algorithms for Sparse Coding” PMLR 2015
> > > >
> > > > >”I also cannot validate the claims the authors make in the tables provided as I have not seen any code for these experiments. The code the authors provided only generates Figure 4a in their paper, and the tied SAE code I provided generates identical plots. Can the authors share the code they are running to create these tables?”
> > > >
> > > > Please note the results in the tables can easily be obtained by looping the code used to generate Figure 4a over the different correlation levels (as provided in our code) and repeating it for 10 runs.
> > > >
> > > > As it comes to sharing the precise code, we note that we are unwilling to share links that break NeurIPS rebuttals guidelines. While we can probably figure out a minimalistic version of our notebook code to allow the reviewer to execute these experiments and share via OpenReview (similar to what the reviewer did), we genuinely feel we have devoted too much energy to these rebuttals and thus do not find ourselves inclined to simplify the code and figure out a way to share it. If our code is wrong or the results have some issue when they are released, we will happily rescind the paper if that is what it takes. However, if there is no trust in our comments or experiments, then we are unsure if openreview is the best platform for this conversation anymore.
> > > >
> > > >
> > > > > If the authors acknowledge the points above, I am happy to raise my score. Specifically this entails:
> > > >
> > > > As we stated before, we promise to update the paper in accordance with our list of points. This includes saying MP-SAEs do not perform well when the parent-child firings are “**perfectly** correlated”.

---

> > > > > ### Comment · Reviewer_vRBm · 2025-08-04
> > > > >
> > > > > > On $|c| = \lambda |p|$: We believe this is a very strong assumption, and we do not see why it should be related to hierarchy
> > > > >
> > > > > This is completely inverted. I am not claiming that ONLY IF $|c| = \lambda |p|$ does it count is hierarchy. My claim is that if you say you solve hierarchy in general, you cannot ignore this case. This case is functionally identical to $|c| = \lambda |p| + \epsilon$ where $\epsilon$ is some small magnitude noise. It sounds like the authors are in agreement that MP-SAEs cannot handle hierarchy in this situation, MP-SAEs will only work if $\epsilon$ is larger than some threshold. It is a very strong assumption to assume that all hierarchical features have a large $\epsilon$. Why is it that anytime there are hierarchical feature firings, their magnitudes must always have some large random variance relative to each other? Surely this is a strong assumption and cannot be assumed in general, especially when no other methods for dealing with hierarchy in the literature make this assumption.
> > > > >
> > > > > >  This includes saying MP-SAEs do not perform well when the parent-child firings are “perfectly correlated”
> > > > >
> > > > > I think its more correct to say that  MP-SAEs do not perform well when parent-child firings are **"strongly correlated"**.
> > > > >
> > > > > >  we genuinely feel we have devoted too much energy to these rebuttals
> > > > >
> > > > > Yes I also agree this has taken far too much of my time
> > > > >
> > > > > ---
> > > > >
> > > > > **Decision**
> > > > >
> > > > > My main issue with this work was that the paper claimed MP-SAEs solve hierarchy in general, as I do not believe this is true. The authors justified this by copying the Matryoshka SAEs code and benchmark verbatim, but subtly modifying the benchmark to add extra variance to feature firings. Without this modification, MP-SAEs fail to deal with hierarchy and fail at that benchmark. Furthermore, despite modifying the benchmark to favor MP-SAEs, the authors then used the results of this modified benchmark to claim that MP-SAEs outperform Matryoshka SAEs at solving hierarchy.
> > > > >
> > > > > The authors have agreed to the following, so I will raise my score, as I specified earlier in the discussion:
> > > > >
> > > > > - Explain that they have changed the Matryoshka SAEs benchmark by adding extra firing variance
> > > > > - Explain why they think this is justified
> > > > > - Explain that MP-SAEs fail to handle hierarchy in the unmodified version Matryoshka SAEs benchmark with (near) perfect firing magnitudes, since MP-SAEs fail to solve hierarchy if the features have strongly correlated firing magnitudes.

---

> > > > > > ### Author Response · Authors · 2025-08-04
> > > > > >
> > > > > > Thank you. We promise updates we agreed to, including the final list of 3 points the reviewer jotted, will be in the paper.

---

### Official Review · Reviewer_wmMK · 2025-06-25

**Clarity:** 3
**Significance:** 3
**Originality:** 2
**Rating:** 5
**Confidence:** 3

**Summary:**

This paper considers the problem of designing a sparse autoencoder whose success does not depend on the traditional linear representation hypothesis. Specifically, the paper proposes replacing the global orthogonality condition of the linear representation hypothesis with a conditional orthogonality condition, that allows for non-orthogonality of some pairs of dictionary elements. The paper provides an algorithm to learn this type of dictionary via a matching pursuit algorithm.

On both synthetic and real data, the paper shows that their approach learns dictionaries that better reconstruct the data, produces representations that are more diverse, and learns features on multi-modal data that are less aligned with modality.

**Questions:**

- Can you give an explicit description of the paper's dictionary learning algorithm?

- Can you provide a formal link between conditional orthogonality and the matching pursuit algorithm?

- Can you explain what is novel about the MP-SAE algorithm relative to the standard matching pursuit algorithm?

- Can you give an explicit example of a non-linear concept that your algorithm will recover but standard SAEs will not?

**Ethical Concerns:**

["NO or VERY MINOR ethics concerns only"]

**Final Justification:**

After the discussion with the authors, my view on this paper has improved, and I am now in favor of acceptance. With the planned edits/clarifications promised by the authors, I think this will be a useful addition to the sparse auto-encoder/model interpretation literature.

**Limitations:**

Yes

**Quality:**

3

**Strengths And Weaknesses:**

Strengths:

- This paper addresses a timely problem. SAEs for concept discovery in large models is a fashionable problem, but the core underlying assumptions of these methods have recently come under scrutiny. This paper proposes a change in assumptions that allows for more flexible dictionaries.

- The paper provides a thorough comparison on a variety of models and metrics. The paper investigates their proposed conditional orthogonality assumption and finds some evidence of it occurring empirically.

Weaknesses:

- The paper provides details on inference in the model, but not on the training procedure. It's not clear what the loss function is for fitting the dictionary. If it's some MSE reconstruction loss on the reconstructed x's, then it's not clear how backpropping through the argmax's are handled.

- It is not clear how the concept of conditional orthogonality is formally related to the matching pursuit algorithm. From Proposition 3.1, we know that the updated residual is orthogonal to the last selected dictionary element, and so we would expect the next selected dictionary element to be among the most orthogonal remaining elements. However, it's not clear how this places any restrictions on the dictionary elements themselves. Also, I don't see where the hierarchy in the definition of conditional orthogonality is arising.

- At a high-level, I'm struggling to see what is the difference between the MP-SAE algorithm, and the standard matching pursuit algorithm. Is the difference in how the dictionary is learned?

- The matching pursuit algorithm can be applied to any learned dictionary, including those learned by other SAEs. It would be interesting to disentangle the effects of the dictionary learning procedure and the inference method by applying the matching pursuit algorithm to the other methods' learned dictionaries. This would help us understand if in Figure 5, better reconstruction is due to a superior dictionary or due to a superior inference/reconstruction algorithm.

- The paper talks about their approach being able to accommodate non-linear concepts, but nothing concrete is shown in the paper. Perhaps a worked out toy example would be appropriate, where the matching pursuit algorithm can find the non-linear concept but the standard TopK VAE is unable to represent it.

- I'm not entirely sure what's going on in Figure 7. I understand that the top row corresponds to the Babel score on the entire dictionary. But the bottom row seems to be over the dictionary elements selected for a single data point or possibly averaged over multiple data points. There's not really enough information to figure it out.

- Although the paper compares against the Matryoshka SAE on a synthetic task, there's no comparison against it on the real-world data. At a high level, the Matryoshka approach is trying to solve a similar problem, so it seems important to compare against it on real data as well.

- While the quantitative results show improvements for the proposed method over previous SAE methods on various metrics, it's still unclear what all of this enables. An application that shows the matching pursuit approach uncovers concepts that the other methods are unable to recover would go a long way towards selling this paper.

---

> ### Author Rebuttal · Authors · 2025-07-30
>
> We thank the reviewer for their careful reading and constructive feedback. We are pleased that they recognized our **"thorough comparison on a variety of models and metrics"**, and that our method **"learns dictionaries that better reconstruct the data, produces representations that are more diverse, and learns features on multi-modal data that are less aligned with modality"**. We also thank them for their questions and comments.
>
> ---
>
> >_The paper provides details on inference in the model, but not on the training procedure. [...] what the loss function is for fitting the dictionary [...] how backpropping through the argmax's are handled._
>
> Thank you for highlighting this point. We agree that the training procedure was not sufficiently detailed in the current version and will include a dedicated section in the appendix to clarify it in the final version of the paper. In brief, we note the dictionary is trained using a standard objective for training SAEs: **mean squared error (MSE).** We do not use any auxiliary objectives or regularization terms compared to other SAEs. While the Matching Pursuit algorithm relies on a non-differentiable Top-1 selection (i.e., argmax) to identify the most correlated component, **we follow the same strategy as in TopK-based SAEs and use a Straight-Through Estimator (STE)** to enable gradient flow. Gradients are propagated only through the selected features, consistent with the approach used in TopK-SAE [1].
>
> In practice, this Top-1 selection can be implemented in a differentiable manner as follows:
> ```python
> def diff_top1(input):
>     values, indices = torch.max(input, dim=1, keepdim=True)
>     z = torch.zeros_like(input)
>     z.scatter_(1, indices, values)
>     return z
> ```
>
> >_Can you provide a formal link between conditional orthogonality and the matching pursuit algorithm? [...] it's not clear how this places any restrictions on the dictionary elements themselves._
>
> Thank you for this question! **As another reviewer also noted the same thing, we realize now our original explanation of the connection to conditional orthogonality was not sufficiently clear, and we will revise the paper to address this directly.**
>
> In brief, we note a formal link between conditional orthogonality and the Matching Pursuit (MP) algorithm arises in the context of block-incoherent dictionaries, as discussed by Peotta and Vandergheynst [2]. In such dictionaries, atoms are highly correlated within groups but nearly orthogonal across groups. MP, by design, tends to select atoms that best reduce the residual at each step; in block-incoherent settings, this typically leads to selecting atoms from distinct groups. As a result, the selected atoms are approximately orthogonal across groups, effectively approximating conditional orthogonality in practice. Conditional orthogonality corresponds to the idealized case of perfect block-incoherence, where atoms are orthogonal to all atoms outside their group. We will revise Section 3 to clarify this interpretation and explicitly highlight this connection.
>
> **Conditional orthogonality is not imposed as a constraint on the dictionary $D$** , but rather emerges as a property of the inference trajectory. At each step, MP selects the atom most aligned with the current residual, which is, by construction, orthogonal to the span of the previously selected atom. This results in a set of features that explain complementary structure in context. We will make this distinction explicit in the revised version.
>
>
>
> >_Also, I don't see where the hierarchy in the definition of conditional orthogonality is arising._
>
> **The connection between hierarchy and conditional orthogonality is based on the findings of [3]**, who show that in the representation spaces of LLMs, hierarchically related concepts (e.g., plant ⇒ animal, mammal ⇒ bird) tend to lie in orthogonal subspaces. Conditional orthogonality naturally follows from this: concepts at different levels of a hierarchy should be orthogonal to each other.
>
> >_Can you give an explicit description of the paper's dictionary learning algorithm? […] what is novel about the MP-SAE algorithm relative to the standard matching pursuit algorithm?_
>
> Thank you for raising this important point. To clarify, **our work does not introduce a new sparse inference algorithm but rather embeds Matching Pursuit (MP) within a trainable sparse autoencoder.** Similar ideas have appeared in image restoration (e.g., Learned MP [4]), but to our knowledge, this is the first application to sparse dictionary learning with an analysis of its representational properties. While Section 3 describes the algorithm and its motivation, we recognize that the dictionary learning aspect could be emphasized more clearly and will revise it in the final version.
>
> Standard MP is a sparse approximation algorithm: given a fixed dictionary $D$, it approximates a sparse code $z$ for an input $x$ by solving
> $$
> \min_z \|x - D z\|_2^2 \quad \text{s.t.} \quad \|z\|_0 \leq k.
> $$
> This is purely an inference method and does not involve learning the dictionary.
>
> MP-SAE, by contrast, unrolls MP within a sparse autoencoder architecture and performs dictionary learning, where the dictionary $D$ is learned across a dataset. The encoder is fixed and implemented via the MP algorithm, while the decoder corresponds to the learnable dictionary $D$. The training objective becomes:
>
> $$
> \min_D \sum_i \|x^i - D z^i\|_2^2 \quad \text{where} \quad z^i = \text{MP}(x^i, D).
> $$
>
> Thus, in our architecture, the *forward pass* performs sparse inference using MP, and the *backward pass* updates the dictionary $D$ based on the reconstruction loss.
>
> >_It would be interesting to disentangle the effects of the dictionary learning procedure and the inference method […] This would help us understand if in Figure 5, better reconstruction is due to a superior dictionary or due to a superior inference/reconstruction algorithm._
>
> This is a great question! In fact, to explore the separation between inference and dictionary learning, we applied MP’s inference process to dictionaries learned by other SAEs during the early stages of this project. **We consistently observed that MP inference improved reconstruction relative to the original shallow encoders. However, these combinations still underperformed compared to MP-SAE, where the dictionary is learned jointly with MP inference!**
>
> Arguably, this is expected since the inference and dictionary learning protocol in SAEs are tightly coupled: the structure of the learned dictionary depends on the inference method used during training. As discussed in [4] and supported by our Babel score metric, the encoder implicitly constrains the space of learnable dictionaries. By avoiding the inductive biases of shallow encoders—such as the quasi-orthogonality assumption—MP-SAE is able to learn more coherent and expressive dictionaries. Thus while it is difficult to fully disentangle the two components, we hope this explanation complements Figure 5.
>
> >_The paper talks about their approach being able to accommodate non-linear concepts, but nothing concrete is shown in the paper [...] Can you give an explicit example of a non-linear concept that your algorithm will recover but standard SAEs will not?_
>
> Thank you for this valuable comment. **An explicit example of a non-linearly accessible concept appears in the multimodal features extracted by MP-SAE but not recovered by standard SAEs**, as shown in our main results and detailed in Appendix B.4. Figure 17 provides an illustration of how such concepts are encoded, highlighting how MP-SAE can recover multimodal features that are inaccessible to standard SAEs. **We will revise the manuscript to emphasize clearly that this is an example of non-linearly accessible concepts uniquely captured by our approach**
>
> >_I'm not entirely sure what's going on in Figure 7._
>
> Thank you for this remark! We agree that Figure 7 needs clearer explanation and will revise it in the manuscript. The top row reports the Babel score across the entire dictionary, reflecting the overall coherence of learned features, independent of how they are used. The bottom row focuses on the features actually selected during inference for each input: we compute the Babel score on this active subset for each data point and then average across the dataset. This highlights the coherence of the features that are used to represent data points.
>
> >_[...] At a high level, the Matryoshka approach is trying to solve a similar problem, so it seems important to compare against it on real data as well._
>
> Fair point! This was indeed an oversight on our part. We encountered Matryoshka SAEs relatively late in the project (the corresponding paper was released only two months before the NeurIPS deadline), and due to time constraints, we could only perform toy experiments comparing MP-SAEs with Matryoshka SAEs. These initial experiments support our claim that MP-SAEs are better suited to extracting hierarchical concepts in controlled settings where the feature structure is known. Since Matryoshka SAEs also aim for hierarchical disentanglement, we fully agree that they should be included in our real-data experiments. **We are now running these additional baselines and will incorporate the results in the final version.**
>
> [1] https://arxiv.org/abs/2406.04093
>
> [2] https://ieeexplore.ieee.org/document/4291867
>
> [3] https://arxiv.org/abs/2406.01506
>
> [4] https://arxiv.org/abs/2503.01822
>
> ---
>
> **Summary:** We thank the reviewer for their feedback, which prompted us to clarify the training procedure, formalize the link between Matching Pursuit and conditional orthogonality, and distinguish MP‑SAE from standard MP. Their comments also led to clearer explanations on hierarchy, examples of non‑linear concepts, clarification of Figure 7, and additional comparisons to Matryoshka SAEs. We hope our responses address these points, and if so, we hope they will encourage the reviewer to consider raising their score!

---

> > ### Comment · Reviewer_wmMK · 2025-08-04
> >
> > Thanks to the authors for their clarifications. Based on these responses and the other reviews, I'm inclined to raise my score. A few points:
> >
> > 1. I think it would improve the quality of the work to include the experiments where the MP inference algorithm is run on the representations produced by other autoencoders, to show that tying MP inference to the learning process improves reconstruction.
> >
> > 2. I'm not sure how the multimodality section proves that the MP-SAE approach learns non-linear concepts. It does show that modality is not the primary source of concepts, but in what way does it show that the resulting concepts are non-linear?
> >
> > 3. I think a qualitative example demonstrating that something like Figure 17 actually does occur in the multimodal setting would be helpful. Does MP-SAE actually learn a zebra concept that spans both text and images? The quantitative results show that other SAEs will not, but it'd be nice to see the positive result for MP-SAEs.
> >
> > 4. I think at a high-level, it's important for the paper to clarify what is theoretically justified, what is empirically observed, and what is heuristic. The initial framing of the paper suggests (at least in my reading) that the concepts will be investigated with more theoretical rigor than actually is done. For example, there is no theoretical guarantee that the learned dictionary will satisfy conditional orthogonality (even in a toy setting); Prop 3.1 maybe suggests the possibility, but falls far short of an analysis of the learning dynamics. Also, there is no theoretical guarantee that the dictionary will encode non-linearly accessible concepts;  again, the equation at line  256-257 suggests this might happen, but I wouldn't say that based on that equation that "MP-SAE thus naturally enables the discovery of higher-order concepts that are conditionally dependent on previously explained structure." At best, this is a heuristic justification with some empirical evidence .

---

> > > ### Author Response · Authors · 2025-08-04
> > >
> > > We thank the reviewer for their thoughtful feedback and for considering raising their score. We will ensure that these points are addressed and reflected in the final version of the paper, as detailed below:
> > >
> > > 1. We will run MP inference on our large vision models using other autoencoder dictionaries to validate that tying MP inference to the learning process improves reconstruction. We will add the equivalent of Figure 5 (with MP inference) in the appendix.
> > >
> > > 2. We will clarify that MP-SAE learns non-linearly accessible multimodal concepts, as other SAEs, being linear extractors, fail to recover such concepts.
> > >
> > > 3. We have observed that MP-SAE actually learns multiple multimodal concepts (such as “zebra”) as shown in Section 4 and Figure 18. We will also add a link to a tool we developed to better visualize qualitative examples of extracted features.
> > >
> > > 4. We will revise the manuscript to clearly distinguish between empirical observations and theoretical justifications. The reviewer’s questions and comments highlighted that this distinction was not clearly explained, and we are grateful to the reviewer for bringing this to our attention.

---

### Official Review · Reviewer_ZQvY · 2025-06-27

**Clarity:** 4
**Significance:** 3
**Originality:** 3
**Rating:** 5
**Confidence:** 4

**Summary:**

The authors present a formalized version of the linear representation hypothesis, which enables them to develop a sparse auto-encoder (SAE) algorithm based on a pursuit matching algorithm. This algorithm is initially evaluated on synthetic data and subsequently applied to real-world data, demonstrating the superiority of the representations it produces compared to existing methods.

**Questions:**

In the derivation of the representation encoding cost, the authors mention the inclusion of a homeostasis mechanism to facilitate learning in inactive units. Could you explain the hit function used and provide evidence of its effectiveness, such as demonstrating that each activity can be activated with roughly equal probability on average (see for example doi:10.1162/neco.2010.05-08-795 )?

During the evaluation using synthetic data, the authors leverage the known dimensionality of the ground truth. Have there been observations of changes in the quality of the representation when using higher or lower dimensions? Additionally, has the effect of dimensionality on the two datasets been assessed?

In the final experiment, the authors demonstrate that their modality index is more evenly distributed between vision and language in their algorithm. Furthermore, it seems that the reconstruction errors are improved. How are these two results interconnected?

**Ethical Concerns:**

["NO or VERY MINOR ethics concerns only"]

**Final Justification:**

The responses to my comments and to those of other reviewers convinced me to raise my score to accept that contribution.

**Limitations:**

One of the limitations, which could also be viewed as a potential strength, of this paper is its interpretation in the context of neurobiology. Given that the algorithm is presented as a hierarchical neural network, it may bear similarities to biological neural networks, particularly in terms of the sparsity of activity observed in neurophysiological experiments. Could the authors draw connections between their algorithm and other hierarchical sparse coding algorithms that aim to establish links with biological systems such as the visual cortex?

**Paper Formatting Concerns:**

I have no concerns regarding the formatting of this paper. However, it might be useful to number all equations for easier reference.

**Quality:**

3

**Strengths And Weaknesses:**

The paper is clearly and effectively presented, allowing for a thorough evaluation of its quality. One of its key strengths is the formalization of the problem, along with the derivation of the sparse autoencoder algorithm and the relevance of the results presented.

While the pursuit matching algorithm appears to yield better results, particularly in its ability to extract more expressive features than standard SAEs, it would have been beneficial to include examples of these features for the various SAE to further support the findings.

---

> ### Author Rebuttal · Authors · 2025-07-30
>
> We thank the reviewer for their thoughtful and constructive feedback, as well as for bringing a neurobiology perspective to our work! We are happy that the reviewer thinks that our work **“is clearly and effectively presented”**. We are particularly encouraged by the recognition that **“the formalization of the problem, along with the derivation of SAEs and the relevance of the results presented”** are considered as key strengths. We also appreciated the analogy drawn by the reviewer between our proposed MP-SAEs and neurobiological systems—particularly the observation that MP-SAEs, when viewed as a hierarchical neural network, may bear similarities to biological sparse coding mechanisms observed in the visual cortex. This perspective is interesting and aligned with some of the broader motivations behind our work.
>
> ---
>
> ## Comments
>
> >_While the pursuit matching algorithm appears to yield better results, particularly in its ability to extract more expressive features than standard SAEs, it would have been beneficial to include examples of these features for the various SAE to further support the findings._
>
>
> Thank you for the suggestion! We agree that including examples of the extracted features across different SAEs would have helped support our claim that MP-SAE, with its inductive bias towards hierarchical concepts, infers more expressive and structured features. We note we have already run several experiments on this front and found MP-SAE selects features that are more global in nature early-on in its inference process, while later-on it selects features that are more fine-grained and local. **As an example, we highlight Fig. 8(b), where we show that in Vision-language models, features selected by MP-SAE early-on distinguish the input modality, but later features correspond to shared concepts between the modalities (such as an object).** We have performed several further experiments (including feature visualization plots) to corroborate these claims, finding, e.g., in Tinystories models features related to topics, frequent lexical patterns like character names, or syntactical concepts are selected early-on, but concepts for punctuations are fairly common later in inference for MP. Since NeurIPS does not allow uploading of new results this year, we are unable to share these visualizations at the moment, but **promise to include them in the final version of the paper!**
>
>
> >_In the derivation of the representation encoding cost, the authors mention the inclusion of a homeostasis mechanism to facilitate learning in inactive units. Could you explain the hit function used and provide evidence of its effectiveness, such as demonstrating that each activity can be activated with roughly equal probability on average (see for example doi:10.1162/neco.2010.05-08-795 )?_
>
>
> Thank you for raising the important issue of “inactive units” (or dead neurons). **We would like to clarify that our paper does not include any notion of a “homeostasis mechanism” nor employ a “hit function.”** However, we are happy to provide more context on this point.
> The concern highlighted by the reviewer refers to a known issue in sparse coding models: when neurons compete to represent data, some may learn faster or become more selective, leading to imbalanced representations. In extreme cases, this can result in some dictionary elements being underutilized or inactive. The paper referenced by the reviewer addresses this by proposing a “cooperative homeostasis” mechanism to equalize the activation probability across neurons.
>
>
> Our paper does not include such a mechanism, and to our knowledge, this is not a common practice in the SAE literature. More importantly, we believe that enforcing uniform activation would be inappropriate in this context, as some concepts naturally occur more frequently or are more salient. Imposing uniformity could therefore distort the true structure of the representations. That said, regarding inactive units, some methods (like TopK [1]) instead rely on auxiliary losses to mitigate dead neurons, but these losses do not enforce that every neuron is activated with roughly equal probability on average.
>
>
> [1] Gao et al. “Scaling and evaluating sparse autoencoders” arXiv:2406.04093 (2024)
>
> ---
>
> ## Questions
>
> >_During the evaluation using synthetic data, the authors leverage the known dimensionality of the ground truth. Have there been observations of changes in the quality of the representation when using higher or lower dimensions? Additionally, has the effect of dimensionality on the two datasets been assessed?_
>
> We assume by “dimensionality”, the reviewer is referring to the sparsity level of the ground truth sparse codes (i.e., the ℓ0​ norm of the ground-truth codes). We certainly agree that, in real-world scenarios, the dimensionality of the ground truth will be unknown. This, however, makes us furthermore excited about MP-SAEs: in our experiments with the synthetic domain the reviewer is referring to, **we found that MP-SAE to be the only method evaluated that did not require prior knowledge of this sparsity level at training time!** This is a key advantage of our approach, as it avoids the need to tune sparsity hyperparameters based on ground-truth labels.
>
> In the case that by “dimensionality” the reviewer means “dictionary size”, we note that we indeed studied the effect of dictionary dimensionality in a setting with pretrained large-scale vision models (Figure 5). Here, we varied the number of concepts from p=125 to p=1250 and observed that as the dictionary grows, the performance gap between MP-SAE and other state-of-the-art SAEs widens. This indicates that MP-SAE is able to effectively leverage increased representational capacity, highlighting its scalability and expressivity. In our synthetic experiments, however, we match the size of the dictionary to that of the ground truth in order to enable direct one-to-one comparisons and evaluate recovery quality in a controlled setting.
>
> We hope we understood the question correctly and are happy to clarify further should any part of our response remain unclear!
>
> >_In the final experiment, the authors demonstrate that their modality index is more evenly distributed between vision and language in their algorithm. Furthermore, it seems that the reconstruction errors are improved. How are these two results interconnected?_
>
> **Thank you ! This question has been a point of discussion among the authors!**
>
> While we do not claim a direct causal link between reconstruction error and modality scores, our working hypothesis is that both emerge from MP-SAE’s bias toward collecting concepts orthogonal to each other align well with the geometry of the VLM where the multimodal information is orthogonal to the modality information.
>
> Specifically, MP-SAE quickly identifies the dominant modality direction (e.g., “image” or “text”) early in inference. Its residual-based selection then emphasizes features that explain remaining variance -> often shared, cross-modal structure! In contrast to classical SAEs, which tend to learn modality specific atoms that do not generalize across modalities.
>
> This hypothesis aligns with recent findings by Papadimitriou et al. (2025), who show that VLM embeddings contain sparse, semantically meaningful directions that are often unimodal, but aligned across modalities. These “bridge” features span shared subspaces and co-activate on paired image-text inputs. **We’ll clarify this interpretation in the final version and appreciate the opportunity to expand on this point.**
>
>
> >_Could the authors draw connections between their algorithm and other hierarchical sparse coding algorithms that aim to establish links with biological systems such as the visual cortex?_
>
>
> Great suggestion! **The MP inference process (greedy selection + residual inhibition) closely mirrors mechanisms proposed in cortical sparse coding**, particularly lateral inhibition in early visual areas. We will clarify this connection in the introduction and related work, situating MP-SAE within the lineage of biologically inspired sparse inference models.
>
>
> Thank you for carefully reviewing the formatting as well. **We really appreciate you pointing out that some equations are missing numbers**, and we will definitely address this in the final version. We remain available for any further clarifications or additional revisions the reviewer may have.

---

> ### Comment · Reviewer_ZQvY · 2025-08-04
> **about “dimensionality”**
>
> yes, by  “dimensionality”, I meant “dictionary size”, and thanks for your precision that you checked for that. can you further develop how your model did not require knowing the ℓ0​ norm of the ground-truth codes?

---

> ### Author Response · Authors · 2025-08-04
> **about ℓ0​ norm of the ground-truth codes**
>
> Thank you for your comment. We are glad our response clarified your question regarding the "dictionary size".
>
> Regarding the ℓ₀ norm, or the sparsity of the ground-truth codes, our MP-SAE can operate without any prior knowledge of this value, as demonstrated in the synthetic experiments.
>
> * **Vanilla and Matryoshka SAEs:** These models are trained with an L1 regularization term $\lambda \\| z \\|_1$.
>
>   During training, \$\lambda\$ is adaptively tuned to target a desired sparsity \$\ell\_0\$:
>
>      * If $\\|z\\|_0 < \ell_0$ target, then $\lambda \leftarrow \lambda \cdot (1-\epsilon)$
>      * If $\\|z\\|_0 > \ell_0$ target, then $ \lambda \leftarrow \lambda \cdot (1+\epsilon)$
>
> * **BatchTopK SAE:** Here, $k = \ell_0 \text{ target} \times n$, where $n$ is the batch size, enforcing a fixed number of active components per batch.
>
> * **MP-SAE:** In contrast, Matching Pursuit leverages the residual at each iteration to decide whether to add a new feature. The algorithm continues selecting features until either:
>
>   1. the residual $\\| r\\|_2=\\| x - \hat{x}\\|_2$ falls below a threshold, or
>   2. the support of the sparse code $z$ stabilizes (if the threshold is unreachable).
>
> **This approach eliminates the need for any prior ℓ₀ target, relying instead on a residual threshold that directly reflects the reconstruction error.**
>
> In the other experiments, we unrolled Matching Pursuit for a fixed $k$ iterations rather than using the residual-based stopping criterion, but the ability to operate without a predefined ℓ₀ target is an inherent advantage of MP-SAE.
>
> We hope this answers your question, and we remain happy to clarify or provide additional details if any other questions arise!

---

### Official Review · Reviewer_9fBU · 2025-06-29

**Clarity:** 4
**Significance:** 3
**Originality:** 3
**Rating:** 5
**Confidence:** 2

**Summary:**

The paper introduces a new conditional orthogonality inductive bias for SAEs, enabling recovery of hierarchical and non-linear features instead of forcing a single flat linear basis. The authors utilize this idea and propose MP-SAE architecture and evaluate it on diverse set of datasets, showing that the method strongly outperforms baselines. Empirical findings of the paper provide further evidence to the intuition that representation spaces are better viewed as nested subspaces, thus justifying further relaxation to empirical support to relaxing the overly restrictive LRH.

**Questions:**

Please, see Weaknesses

**Ethical Concerns:**

["NO or VERY MINOR ethics concerns only"]

**Final Justification:**

After reading extensive discussion of authors with other reviewers, and their willingness to further elaborate and acknoledge certain assumptions in the manuscript, I decided to maintain my score still leaning towards acceptance.

**Limitations:**

yes

**Quality:**

4

**Strengths And Weaknesses:**

## Strengths
- The paper is very well written and includes neat and informative illustrations.
- The authors provide a broad discussion of SAEs (which is very useful for an outside reader), carefully formalizing the new constraint and explaining its intuition
- Conditional orthogonality and its instationed in MP-SAE is well motivated, connected to existing literature, and thoroughly analyzed
- MP-SAE is evaluated on diverse datasets including both synthetic benchmarks, and natural data (coco) with a sota vision models (CLIP, DINOv2), where the proposed method strongly outperformed the baselines
- The analysis is further extended to vision-language models and toy language models in the Appendix
- Authors discuss theoretical guaruantees of the MP-SAE infernce in Appendix

## Weaknesses
- It would be great to see more results for the LLMs, e.g. 0.5-3b models, to strenghten the generalization of the approach across the modalities
- Did authors observed any failure modes of proposed MP-SAE under highly entangled representations when compared to Vanilla SAE?

---

> ### Author Rebuttal · Authors · 2025-07-30
>
> We thank the reviewer for their positive and thoughtful feedback. We particularly appreciate the acknowledgement that the paper **“is very well written and includes neat and informative illustrations.”** We’re also glad that the **“broad discussion of SAEs”** was found to be **“very useful for an outside reader,”** and that the reviewer considers our treatment of **“conditional orthogonality and its instantiation in MP-SAE”** to be **“well motivated, connected to existing literature, and thoroughly analyzed.”** We also thank the reviewer for raising several interesting points, which we address below.
>
> ---
>
> >_It would be great to see more results for the LLMs, e.g. 0.5-3b models, to strengthen the generalization of the approach across the modalities._
>
> Thank you for this suggestion! While **we agree that including results on larger language models (e.g., 0.5–3B) would help further validate the utility of MP-SAEs, we emphasize our goal in this paper was broader**: we wanted to contextualize limitations of existing SAEs with respect to representational structures that lie outside the scope of the linear representation hypothesis, i.e., hierarchical concepts. We thus proposed and used MP-SAE as an architecture that has a baked-in prior for modeling hierarchical structures, and could hence help analyze how SAEs that do capture hierarchical concepts differ from ones that do not. To this end, we conducted experiments comparing different SAEs in Sections 3, 4 on **large scale vision models**, **multimodal models**, **smaller language models**, and **synthetic toy problems**. Across these rather disparate domains, we saw an almost perfect consistency of results: one gets better expressivity, flexibility, and ability to model hierarchical concepts with MP-SAE, but not with standard SAEs. Given the breadth of our evaluation, we do believe our current validation suffices to justify our claims. However, the broader utility of MP-SAE, e.g., in finding hierarchical concepts in LLMs, which we are quite excited about, warrants a more thorough study that is best left for a future paper (we do however note we are already pursuing this!).
>
> ---
>
> >_Did authors observe any failure modes of proposed MP-SAE under highly entangled representations when compared to Vanilla SAE?_
>
> Great question—we were also curious about the robustness of MP-SAEs when the data contains entangled / correlated concepts! To study this systematically, we had reported in Figure 14 (appendix) an extended version of our synthetic testbed results from the paper, where we compare how different SAEs behave when the data contains highly entangled features. Vanilla SAE results are shown in the first column of the figure, and MP-SAE results in the last. In brief, **we see MP-SAEs break down entangled levels into hierarchical stages, capturing both coarse and fine structure, while Vanilla SAE flattens the hierarchy, focusing only on one level of it.** Neither method perfectly recovers all ground-truth features in this challenging setup, but they fail in notably different ways, as discussed below.
>
> - For MP-SAE, we observe a recurring and characteristic behavior: the model tends to allocate one feature aligned with the parent direction, as well as one representative feature per group of children. For example, in the first MP-SAE run (top-left panel), the parent is represented by the second-to-last component, while child groups are captured by components at indices 2, 6, and 11, each corresponding to a ground-truth child feature. Notably, MP-SAE does not assign a unique feature to every child. Instead, it learns additional components that interpolate between the representative child and the others within the group. When intra-level correlations are high, this results in a two-stage decomposition:
>  1. a general direction aligned with one child that captures the shared structure of the group, and
>  2. refinement features that describe deviations from this common direction.
>
>     We observe a similar behavior at the parent level as well. When parent features are highly entangled, MP-SAE tends to represent the shared parent structure using one dominant direction, followed by secondary components that refine this direction to better match the individual parents. Thus, this coarse-to-fine decomposition emerges at multiple levels of the hierarchy, not just within child groups.
>
>
>     This hierarchical organization allows MP-SAE to partially resolve entangled structure by capturing both group-level similarities and within-group variations. As a result, the model typically selects a larger number of active components (visible in the support of the sparse code), reflecting the increased representational complexity required.
>
> - In contrast, Vanilla SAE is able to reconstruct the children accurately, despite their high correlation, but consistently fails to capture the parent features. This is evident in the very low or absent activation of parent components in the sparse codes. The model appears to allocate its full capacity to fitting the entangled children, effectively ignoring higher-level structure.
>
> We hope this clarifies the distinct failure modes of the two approaches and addresses the reviewer’s question. We will add this discussion to the appendix to make these observations more accessible to readers.

---

> > ### Comment · Reviewer_9fBU · 2025-08-05
> > **A Reply to the Rebuttal**
> >
> > After reading extensive discussion of authors with other reviewers, and their willingness to further elaborate and acknoledge certain assumptions in the manuscript, I decided to maintain my score still leaning towards acceptance.

---

### Official Review · Reviewer_twXJ · 2025-07-02

**Clarity:** 3
**Significance:** 3
**Originality:** 3
**Rating:** 5
**Confidence:** 4

**Summary:**

This paper proposes the incorporation of an existing sparse coding concept (matching pursuit) into sparse autoencoder architectures in order to more effectively represent meaningful, hierarchical features. The authors analyze the weaknesses of existing methods, and claim that matching pursuit sparse autoencoders (MP-SAEs) are able to more effectively represent features that existing methods fail to capture, specifically features that are conditionally dependent, hierarchical, nested, or nonlinear.

**Questions:**

1. Similar to your relaxing of the linear representation hypothesis into conditional orthogonality, other papers (such as https://doi.org/10.48550/arXiv.2306.07304) attempt to formalize SAEs in a different way. Are MP-SAEs able to fit into the formalizations given here? If not, why or why not are MP-SAEs still a comparable method to traditional SAE architectures?
2. When considering MP-SAEs as a method, their benefits seem clear in the synthetic examples provided. However, are there clear use cases in real-world data that MP-SAEs would be better suited for?

**Ethical Concerns:**

["NO or VERY MINOR ethics concerns only"]

**Final Justification:**

The authors clarified many of the positioning weaknesses with respect to other works in this area and committed to comparing against other baselines. I also followed the long discussion with the other reviewer. I would like to leave my score at a 5. It's a great paper.

**Limitations:**

yes

**Quality:**

4

**Strengths And Weaknesses:**

Strengths:

* Paper presents a thorough basis for why the proposed method matters, highlighting gaps in existing methods
* Thorough discussion of existing and proposed formalizations is given
* The structure of the paper is largely clear and concise.
* Effective, well-scoped argument about MP-SAEs as an alternative architecture, for use in cases where the true features are outside of the global orthogonality regime/linear representation hypothesis.
* Solid advancement, proposing an original (though inspired quite heavily by the existing matching pursuit method), useful architecture.

Weaknesses:

* The argument that MP-SAEs are a valid operationalization of conditional orthogonality is based on a claim that no evidence is provided for
* Sections 3 and 4 have considerable overlap, particularly when discussing experimental results that validate claimed properties of MP-SAEs.
* Inconsistent benchmarking against existing models. Figure 4 compares MP-SAEs to Matryoshka SAEs, but Matryoshka SAEs are omitted from figures 5-8. Matryoshka SAEs also seem to be the most comparable SAE architecture, so this omission hurts the strength of claims.

---

> ### Author Rebuttal · Authors · 2025-07-30
>
> We thank the reviewer for their thoughtful and encouraging feedback! We are glad they found our method to be a **“solid advancement”** and appreciated our **“effective, well-scoped argument about MP-SAEs as an alternative architecture.”** We are especially pleased that the reviewer highlighted our **“thorough discussion of existing and proposed formalizations”** and the clarity of our motivation for relaxing the linear representation hypothesis. Below, we address specific comments.
>
> ---
>
> > _Q1: [...] The argument that MP-SAEs are a valid operationalization of conditional orthogonality is based on a claim that no evidence is provided for_
>
> Thank you for raising this point! To clarify the mechanism via which MP operationalizes conditional orthogonality, we refer the reviewer to Proposition 3.1: therein, we show **MP’s inference dynamics follow an orthogonality constraint whereby each inference step uses a concept (i.e., dictionary atom) that results in a residual orthogonal to said concept** (see Figure 3-left for an intuitive visualization). While this constraint certainly does not guarantee conditional orthogonality, i.e., it does not guarantee that subsequent concepts used for explaining the residual will be orthogonal to the previously selected ones, in practice we find the local constraint leads to an emergent organization of the dictionary such that inference with MP indeed satisfies conditional orthogonality.
>
> Specifically, we find MP-SAE selects concepts that are orthogonal to previously selected ones (quantified via the Babel score in Fig. 7). Similar results have previously been reported in dictionary learning literature, from which we sought our motivation: e.g., Peotta and Vandergheynst [1] show that when atoms are highly correlated within groups but nearly orthogonal across them (a.k.a., block incoherent dictionaries), MP tends to select atoms from distinct groups, effectively satisfying conditional orthogonality.
>
> **We nevertheless agree with your comment that the precise mechanism via which MP is operationalizing conditional orthogonality can be better clarified**. We promise to revise Section 3 and add the discussion above to address this concern in the final version of the manuscript!
>
> [1] Peotta and Vandergheynst “Matching Pursuit With Block Incoherent Dictionaries” IEEE Transactions on Signal Processing (2007)
>
> ---
>
> >_Q2 [...] Sections 3 and 4 have considerable overlap, particularly when discussing experimental results that validate claimed properties of MP-SAEs._
>
> We agree. However, as you noted, the goal of Section 3 was to formalize MP-SAEs and use a toy setting that helps establish their ability to extract hierarchical concepts. Meanwhile, in Section 4 our goal was to scale up our experiments to natural domains and demonstrate the validity of our claims in these significantly more complicated settings. Merging the two sections, which we did attempt, induces a back-and-forth between the precise experimental setup being investigated, resulting in a convoluted writing flow. Nevertheless, **we promise to try to smoothen the transitions between the sections and minimize redundancy** therein: e.g., we can try to keep the formalization of MP-SAE in Section 3 and move all experiments to Section 4, with a subsection each on synthetic vs. natural domains.
>
> ---
>
> >_Q3: Inconsistent benchmarking against existing models. Figure 4 compares MP-SAEs to Matryoshka SAEs, but Matryoshka SAEs are omitted from figures 5-8. Matryoshka SAEs also seem to be the most comparable SAE architecture, so this omission hurts the strength of claims._
>
> **Fair point! This was indeed an oversight on our part.** We came across Matryoshka SAEs relatively late in the project (their paper was released less than two months before the NeurIPS deadline). Because our large-scale experiments focus on vision, whereas theirs are on language, we could not directly adopt their code for comparison given the limited time before the deadline. We were, however, able to perform toy experiments, which corroborate that MP-SAEs are better suited for extracting hierarchical concepts in controlled regimes where feature structure is known. Nonetheless, since Matryoshka SAEs also target hierarchical disentanglement, we agree they merit inclusion in our real-data experiments. **All these additional baselines will be includes in the final version**.
>
> ---
>
> >_Q4: Similar to your relaxing of the linear representation hypothesis into conditional orthogonality, other papers (such as https://doi.org/10.48550/arXiv.2306.07304) attempt to formalize SAEs in a different way. Are MP-SAEs able to fit into the formalizations given here? If not, why or why not are MP-SAEs still a comparable method to traditional SAE architectures?_
>
> Great question! As such, **MP-SAEs are fully compatible with the formalism described in Fel et al.** and directly comparable to traditional SAEs, differing only in the inductive bias of the encoder. Specifically, traditional SAEs typically use a shallow encoder, e.g., a single linear layer followed by a non-linearity, to produce a sparse code with (say) K active latents. In contrast, MP-SAEs, based on the Matching Pursuit procedure, unroll the same encoding process over K steps by greedily selecting atoms based on correlation with the residual at each step. This eventually still leads to a K-sparse code, building one concept at a time.
>
> Overall, given the reviewer’s prompt, we will use Fel et al.’s formalization to add the following note that clarifies the connection between different SAEs:
>
>
> \begin{aligned}
> (Z^\star, D^\star) = \arg\min_{Z, D} \| A - Z D^\top \| \\\\
>  \text{Subject to:} \\\\
>  Z_i \in \{ e_1, \ldots, e_k \} \quad &\text{(\textbf{ACE})}, \\\\
>  D^\top D = \mathbf{I} \quad &\text{(\textbf{ICE})}, \\\\
>  Z \geq 0,\ D \geq 0 \quad &\text{(\textbf{CRAFT})}, \\\\
>  Z = \Psi_{\theta(A),\  \|Z\|_0 \leq K} \quad &\text{(\textbf{SAE})}, \\\\
>  Z = MP_K(A;\ D) \quad &\text{(\textbf{MP-SAE})}
> \end{aligned}
>
> \begin{aligned}
> MP_K(A; D) = \sum_{k=1}^K \operatorname{top1}({r}^{(k-1)} D^\top), \quad
>     {r}^{(0)} = A, \quad
>     {r}^{(k)} = {r}^{(k-1)} - \operatorname{top1}({r}^{(k-1)} D^\top) D
> \end{aligned}
>
>
> ---
>
> >_Q5: When considering MP-SAEs as a method, their benefits seem clear in the synthetic examples provided. However, are there clear use cases in real-world data that MP-SAEs would be better suited for?_
>
> Great question! First, we would like to note that our goal in this paper was to contextualize popular SAE architectures with respect to recent developments identifying hierarchical, nonlinear, and multidimensional representations in neural networks. Such representations lie outside the scope of the linear representation hypothesis (LRH)—the assumed model of conceptual organization in neural networks that motivates SAEs. Specifically focusing on hierarchical concepts, we show how standard SAE architectures fail to identify representations corresponding to them. Thus, **our goal with proposing MP-SAE was to show how, if one knows the structure they are trying to capture, it is possible to design useful SAE architectures that scale easily** (as shown by our large-scale vision experiments), perform very well across a broad spectrum of metrics (as shown in Section 4), and show surprising results (e.g., an explanation for the modality gap in vision-language models).
>
> That said, **hierarchical concepts are commonplace in several natural domains**: e.g., LLMs show a very structured, hierarchical organization of high-level concepts like human emotions [1], which can matter for monitoring what a model is modeling about the user [2]; scene graphs are often used to develop hierarchical, compositional representations of inter-object relations in vision [3]; and several scientific domains exhibit tree-structured concepts, e.g., phylogenetic trees, that may be extractable from language models trained on data such as genomic structures [4]. We strongly expect MP-SAEs will find utility in these domains, warranting a thorough study that is best left for future work.
>
> [1] Zhao et al. “Emergence of Hierarchical Emotion Organization in Large Language Models” arXiv:2507.10599 (2025)
>
> [2] Chen et al. “Designing a Dashboard for Transparency and Control of Conversational AI” arXiv:2406.07882 (2024)
>
> [3] Johnson et al. “Image Retrieval using Scene Graphs”, IEEE Conference on Computer Vision and Pattern Recognition (CVPR) (2015)
>
> [4] Brixi et al. “Genome modeling and design across all domains of life with Evo 2” bioRxiv (2025)

---

### Decision · Program_Chairs · 2025-09-17

**Decision:**

Accept (poster)

**Comment:**

The paper introduces the idea of integrating the matching pursuit sparse coding framework into sparse autoencoder architectures, aiming to improve the representation of meaningful and hierarchical features. It effectively highlights the shortcomings of existing methods while proposing a novel solution. The main concern, as noted by reviewers, is the adjustment of benchmark parameters, which should be carefully addressed in the final version. In addition, there are several other reviewer comments regarding clarity and presentation that the authors should incorporate. Despite these limitations, the work makes a worthwhile contribution to the interpretability literature.